

# Bethe vectors and recurrence relations for twisted Yangian based models

**Vidas Regelskis**

Department of Physics, Astronomy and Mathematics,
University of Hertfordshire, Hatfield AL10 9AB, UK
Institute of Theoretical Physics and Astronomy, Vilnius University,
Saulėtekio av. 3, Vilnius 10257, Lithuania

vidas.regelskis@gmail.com

## Abstract

We study Olshanski twisted Yangian based models, known as one-dimensional "soliton non-preserving" open spin chains, by means of algebraic Bethe Ansatz. The even case, when the bulk symmetry is $\mathfrak{gl}_{2n}$ and the boundary symmetry is $\mathfrak{sp}_{2n}$ or $\mathfrak{so}_{2n}$, was studied in [12]. In the present work, we focus on the odd case, when the bulk symmetry is $\mathfrak{gl}_{2n+1}$ and the boundary symmetry is $\mathfrak{so}_{2n+1}$. We explicitly construct Bethe vectors and present a more symmetric form of the trace formula. We use the composite model approach and $Y(\mathfrak{gl}_n)$-type recurrence relations to obtain recurrence relations for twisted Yangian based Bethe vectors, for both even and odd cases.

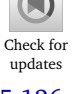

# 1   Introduction

Twisted Yangian based models, known as one-dimensional "soliton non-preserving" open spin chains, were first investigated by means of analytic Bethe Ansatz techniques [1–3,9] and more recently in [4]. Such models are known to play a role in Yang-Mills theories, where twisted Yangians emerge in the context of integrable boundary overlaps [7,15] and open fishchains [17].

    A crucial step in understanding twisted Yangian based models is finding explicit expressions of Bethe vectors. In the case when the bulk symmetry is $\mathfrak{gl}_{2n}$ and the boundary symmetry is $\mathfrak{sp}_{2n}$ or $\mathfrak{so}_{2n}$, this was achieved in [12] using algebraic Bethe anstaz techniques put forward in [8, 29]. These techniques apply to the cases, when the $R$-matrix intertwining monodromy matrices of the model can be written in a six-vertex block-form. The monodromy matrix is then also written in a block-form, in terms of matrix operators $A$, $B$, $C$, and $D$, that are matrix analogues of the conventional creation, annihilation and diagonal operators. Exchange relations between these matrix operators turn out to be reminiscent of those of the standard six-vertex model. Such techniques have been used to study $\mathfrak{so}_{2n}$- and $\mathfrak{sp}_{2n}$-symmetric spin chains in [13, 14, 16, 28, 30]. A more general framework of such techniques has recently been proposed in [11].

    In the present paper we extend the results of [12] to the odd case, when the bulk symmetry is $\mathfrak{gl}_{2n+1}$ and the boundary symmetry is $\mathfrak{so}_{2n+1}$. This extension is based on a simple observation that the generating matrix of the odd twisted Yangian $Y^+(\mathfrak{gl}_{2n+1})$ can be decomposed into four overlapping $(n+1) \times (n+1)$-dimensional matrix operators satisfying the same exchange relations as those of $Y^+(\mathfrak{gl}_{2n+2})$ thus allowing us to employ the same algebraic Bethe Ansatz approach. However, the overlapping introduces a new challenge since the middle entry of the generating matrix is now included in both $A$ and $B$ matrix operators leading to an uncertainty in the $AB$ exchange relation. This issue is resolved in the technical Lemma 3.8 stating action of the middle entry on Bethe vectors. Computing this action requires knowledge of recurrence relations for Bethe vectors. We use the composite model techniques together with the $Y(\mathfrak{gl}_n)$-type recurrence relations found in [20] to obtain the $Y^{\pm}(\mathfrak{gl}_{2n})$- and $Y^+(\mathfrak{gl}_{2n+1})$-type recurrence relations. The main results of this paper are presented in Theorem 3.9 and Propositions 4.4 and 4.6.

The first main result, Theorem 3.9, states that Bethe vectors, defined by formula (66), are eigenvectors of the transfer matrix, defined by formula (68), provided Bethe equations (77) and (78) hold. This Theorem is an extension of Theorems 4.3 and 4.4 in [12] to the odd case. Commutativity of transfer matrices is shown in Appendix A.2. We also found a more symmetric form of the trace formula for Bethe vectors derived in [12]. The new formula is presented in Proposition 3.12. Its main ingredient is the so-called "master" creation operator, defined by formula (79). Low rank examples of the "master" creation operator are presented in Example 3.11.

The second main result, Propositions 4.4 and 4.6, present recurrence relation for $Y^{\pm}(\mathfrak{gl}_{2n})$- and $Y^{+}(\mathfrak{gl}_{2n+1})$-based Bethe vectors, respectively. Schematically, they are of the form

$$\Psi^{(m_1,...,m_n)} = \sum_{1 \leq i \leq n} s_{i,2n-i+1} \Psi^{(m_1,...,m_{i-1},m_i-2,...,m_{n-1}-2,m_n-1)}$$
$$+ \sum_{1 \leq i < j \leq n} (s_{i,2n-j+1} + s_{j,2n-i+1}) \Psi^{(m_1,...,m_{i-1},m_i-1,...,m_{j-1}-1,m_j-2,...,m_{n-1}-2,m_n-1)}, \quad (1)$$

in the even case and

$$\Psi^{(m_1,...,m_n)} = \sum_{1 \leq i \leq n} s_{i,n+1} \Psi^{(m_1,...,m_{i-1},m_i-1,...,m_{n-1}-1,m_n-1)}$$
$$+ \sum_{1 \leq i < n} (s_{i,n+2} + s_{n,n+i+2}) \Psi^{(m_1,...,m_{i-1},m_i-1,...,m_{n-1}-1,m_n-2)}$$
$$+ \sum_{1 \leq i \leq n} s_{i,2n-i+2} \Psi^{(m_1,...,m_{i-1},m_i-2,...,m_{n-1}-2,m_n-2)}$$
$$+ \sum_{1 \leq i < j < n} (s_{i,2n-j+2} + s_{j,2n-i+2}) \Psi^{(m_1,...,m_{i-1},m_i-1,...,m_{j-1}-1,m_j-2,...,m_{n-1}-2,m_n-2)}, \quad (2)$$

in the odd case. Here $m_i$'s indicate excitation numbers associated with the $i$-th simple root of the boundary symmetry algebra, $s_{ij}$'s represent generating series of the twisted Yangian, and all scalar factors and spectral parameter dependencies are omitted. These relations are compatible with the weight grading of twisted Yangian (see Appendix A.1). Repeated application of relations (1) and (2) allows us to express Bethe vectors $\Psi^{(m_1,...,m_n)}$ in terms of those with no level-$n$ excitations, i.e. with $m_n = 0$. The latter Bethe vectors obey $Y(\mathfrak{gl}_n)$-type recurrence relations of the form [20]

$$\Psi^{(m_1,...,m_{n-1},0)} = \sum_{1 \leq i < n} s_{i,n} \Psi^{(m_1,...,m_{i-1},m_i-1,...,m_{n-1}-1,0)}, \quad (3)$$

the explicit form of which is recalled in Appendix A.3. This feature is explained in Remark 3.3. Recurrence relations (1) and (2) are rather complex, especially in the odd case. However, low rank cases, explicitly stated in Examples 4.5 and 4.7, are manageable for practical computations. Moreover, the known results of $Y(\mathfrak{gl}_n)$-based models [19–21, 23] can be employed after the first step of nesting.

The paper is organised as follows. In Section 2 we introduce notation used throughout the paper and recall the necessary algebraic properties of twisted Yangians. In Section 3 we present the algebraic Bethe anstaz: Bethe vectors, their eigenvalues and the corresponding Bethe equations. We consider both even and odd cases simultaneously giving a coherent framework needed for obtaining recurrence relations. In Section 4 we obtain recurrence relations and present a proof of the technical Lemma 3.8. In Appendix A we recall weight grading of $Y^{\pm}(\mathfrak{gl}_N)$, a recurrence relation for $Y(\mathfrak{gl}_n)$-based Bethe vectors, and provide a proof of commutativity of transfer matrices.

## 2 Definitions and preliminaries

Throughout the manuscript the middle alphabet letters $i, j, k, \ldots$ will be used to denote integer numbers, letters $u, v, w, \ldots$ will denote either complex numbers or formal parameters, and letters $a$ and $b$ (often decorated with additional indices) will be used to label vector spaces.

### 2.1 Lie algebras

Choose $N \geq 2$. Let $\mathfrak{gl}_N$ denote the general linear Lie algebra and let $e_{ij}$ with $1 \leq i, j \leq N$ be the standard basis elements of $\mathfrak{gl}_N$ satisfying

$$[e_{ij}, e_{kl}] = \delta_{jk} e_{il} - \delta_{il} e_{kj}. \tag{4}$$

The orthogonal Lie algebra $\mathfrak{so}_N$ and the symplectic Lie algebra $\mathfrak{sp}_N$ can be regarded as subalgebras of $\mathfrak{gl}_N$ as follows. For any $1 \leq i, j \leq N$ set $\theta_{ij} := \theta_i \theta_j$ with $\theta_i := 1$ in the orthogonal case and $\theta_i := \delta_{i>N/2} - \delta_{i \leq N/2}$ in the symplectic case. Introduce elements $f_{ij} := e_{ij} - \theta_{ij} e_{\bar{j}\bar{i}}$ with $\bar{i} := N - i + 1$ and $\bar{j} := N - j + 1$. These elements satisfy the relations

$$[f_{ij}, f_{kl}] = \delta_{jk} f_{il} - \delta_{il} f_{kj} + \theta_{ij} (\delta_{j\bar{l}} f_{k\bar{i}} - \delta_{i\bar{k}} f_{\bar{j}l}), \tag{5}$$

$$f_{ij} + \theta_{ij} f_{\bar{j}\bar{i}} = 0, \tag{6}$$

which in fact are the defining relations of $\mathfrak{so}_N$ and $\mathfrak{sp}_N$. It will be convenient to denote both algebras by $\mathfrak{g}_N$. Write $N = 2n$ or $N = 2n + 1$. In this work we will focus on the following chain of Lie algebras

$$\mathfrak{gl}_N \supset \mathfrak{g}_N \supset \mathfrak{gl}_n \supset \mathfrak{gl}_{n-1} \supset \cdots \supset \mathfrak{gl}_2,$$

where $\mathfrak{gl}_n$, $\mathfrak{gl}_{n-1}$, $\ldots$, $\mathfrak{gl}_2$ are subalgebras of $\mathfrak{g}_N$ generated by $f_{ij}$ with $1 \leq i, j \leq k$ and $k = n, n-1, \ldots, 2$, respectively.

### 2.2 Matrix operators

For any $k \in \mathbb{N}$ let $E_{ij}^{(k)} \in \mathrm{End}(\mathbb{C}^k)$ with $1 \leq i, j \leq k$ denote the standard matrix units with entries in $\mathbb{C}$ and let $E_i^{(k)} \in \mathbb{C}^k$ with $1 \leq i \leq k$ denote the standard basis vectors of $\mathbb{C}^k$ so that $E_{ij}^{(k)} E_l^{(k)} = \delta_{jl} E_i^{(k)}$. We will frequently use the barred index notation

$$E_{\bar{i}\bar{j}}^{(k)} := E_{k-i+1, k-j+1}^{(k)}, \quad E_{\bar{i}}^{(k)} := E_{k-i+1}^{(k)}. \tag{7}$$

Introduce matrix operators

$$I^{(k,k)} := \sum_{i,j} E_{ii}^{(k)} \otimes E_{jj}^{(k)}, \qquad P^{(k,k)} := \sum_{i,j} E_{ij}^{(k)} \otimes E_{ji}^{(k)}, \qquad Q^{(k,k)} := \sum_{i,j} E_{ij}^{(k)} \otimes E_{\bar{i}\bar{j}}^{(k)}, \tag{8}$$

where the tensor product is defined over $\mathbb{C}$. We will always assume that the summation is over all admissible values, if not stated otherwise. Note that the operator $Q^{(k,k)}$ is an idempotent operator, $(Q^{(k,k)})^2 = k Q^{(k,k)}$, obtained by partially transforming the permutation operator $P^{(k,k)}$ with the transposition $\omega : E_{ij}^{(k)} \mapsto E_{\bar{j}\bar{i}}^{(k)}$, that is, $Q^{(k,k)} = (\mathrm{id} \otimes \omega)(P^{(k,k)}) = (\omega \otimes \mathrm{id})(P^{(k,k)})$.

Next, we introduce a matrix-valued rational function

$$R^{(k,k)}(u) := I^{(k,k)} - u^{-1} P^{(k,k)}, \tag{9}$$

called the *Yang's R-matrix*. It is a solution of the quantum Yang-Baxter equation in $\mathbb{C}^k \otimes \mathbb{C}^k \otimes \mathbb{C}^k$:

$$R_{12}^{(k,k)}(u-v) R_{13}^{(k,k)}(u-z) R_{23}^{(k,k)}(v-z) = R_{23}^{(k,k)}(v-z) R_{13}^{(k,k)}(u-z) R_{12}^{(k,k)}(u-v). \tag{10}$$

Here the subscript notation indicates the tensor spaces the matrix operators act on. We will use such a subscript notation throughout the manuscript. We will also make use the partially $\omega$-transposed $R$-matrix

$$\widehat{R}^{(k,k)}(u) := (\mathrm{id} \otimes \omega)(R^{(k,k)}(u)) = I^{(k,k)} - u^{-1}Q^{(k,k)}, \tag{11}$$

satisfying a transposed version of (10):

$$R_{12}^{(k,k)}(u-v)\widehat{R}_{23}^{(k,k)}(v-z)\widehat{R}_{13}^{(k,k)}(u-z) = \widehat{R}_{13}^{(k,k)}(u-z)\widehat{R}_{23}^{(k,k)}(v-z)R_{12}^{(k,k)}(u-v). \tag{12}$$

## 2.3 Twisted Yangian $Y^{\pm}(\mathfrak{gl}_N)$

We briefly recall the necessary details of the "$\rho$-shifted" twisted Yangian $Y^{\pm}(\mathfrak{gl}_N)$ adhering closely to [3,12] (see also [27] and Chapters 2 and 4 in [26]); here the upper (resp. lower) sign in $\pm$ corresponds to the orthogonal (resp. symplectic) case. The parameter $\rho \in \mathbb{C}$ is introduced to accommodate applications to Yang-Mills theories and condensed matter systems, where $\rho$ plays a role of a boundary parameter, and integrable overlaps, where $\rho$ appears as an integer parameter in the nesting procedure.

Twisted Yangian $Y^{\pm}(\mathfrak{gl}_N)$ is a unital associative $\mathbb{C}$-algebra with generators $s_{ij}[r]$ where $1 \le i,j \le N$ and $r \in \mathbb{N}$. The defining relations, written in terms of the generating series $s_{ij}(u) := \delta_{ij} + \sum_{r \ge 1} s_{ij}[r]u^{-r}$, where $u$ is a formal variable, are

$$
\begin{aligned}
[s_{ij}(u), s_{kl}(v)] = {} & \frac{1}{u-v}\Big(s_{kj}(u)s_{il}(v) - s_{kj}(v)s_{il}(u)\Big) \\
& - \frac{1}{u-\tilde{v}}\Big(\theta_{j\bar{k}}s_{i\bar{k}}(u)s_{\bar{j}l}(v) - \theta_{i\bar{l}}s_{k\bar{l}}(v)s_{\bar{i}j}(u)\Big) \\
& + \frac{1}{(u-v)(u-\tilde{v})}\theta_{i\bar{j}}\Big(s_{k\bar{i}}(u)s_{\bar{j}l}(v) - s_{k\bar{i}}(v)s_{\bar{j}l}(u)\Big),
\end{aligned} \tag{13}
$$

and

$$\theta_{ij}s_{\bar{j}\bar{i}}(\tilde{u}) = s_{ij}(u) \pm \frac{s_{ij}(u) - s_{ij}(\tilde{u})}{u - \tilde{u}}. \tag{14}$$

Here $\bar{\imath} = N - i + 1$, $\bar{\jmath} = N - j + 1$, etc., and $\tilde{u} := -u - \rho$, $\tilde{v} := -v - \rho$. These relations can be cast in a matrix form as follows. Combine the series $s_{ij}(u)$ into the generating matrix

$$S^{(N)}(u) := \sum_{i,j} E_{ij}^{(N)} \otimes s_{ij}(u). \tag{15}$$

The defining relations (13) and (14) are then equivalent to the twisted reflection equation

$$R_{12}^{(N,N)}(u-v)S_1^{(N)}(u)\widehat{R}_{12}^{(N,N)}(\tilde{v}-u)S_2^{(N)}(v) = S_2^{(N)}(v)\widehat{R}_{12}^{(N,N)}(\tilde{v}-u)S_1^{(N)}(u)R_{12}^{(N,N)}(u-v), \tag{16}$$

and the symmetry relation

$$\omega(S^{(N)}(\tilde{u})) = S^{(N)}(u) \pm \frac{S^{(N)}(u) - S^{(N)}(\tilde{u})}{u - \tilde{u}}. \tag{17}$$

## 2.4 Block decomposition

Set $\hat{n} := n$ when $N = 2n$ and $\hat{n} := n + 1$ when $N = 2n + 1$. Then define $\hat{n} \times \hat{n}$ dimensional matrix operators

$$
\begin{aligned}
A^{(\hat{n})}(u) &= \sum_{i,j} E_{ij}^{(\hat{n})} \otimes s_{ij}(u), & B^{(\hat{n})}(u) &= \sum_{i,j} E_{ij}^{(\hat{n})} \otimes s_{i,n+j}(u), \\
C^{(\hat{n})}(u) &= \sum_{i,j} E_{ij}^{(\hat{n})} \otimes s_{n+i,j}(u), & D^{(\hat{n})}(u) &= \sum_{i,j} E_{ij}^{(\hat{n})} \otimes s_{n+i,n+j}(u).
\end{aligned} \tag{18}
$$

These operators are matrix analogues of the conventional $a$, $b$, $c$ and $d$ operators of the six-vertex type algebraic Bethe Ansatz. The exchange relations that we will need are [12]:

$$A_b^{(\hat{n})}(v)B_a^{(\hat{n})}(u) = R_{ab}^{(\hat{n},\hat{n})}(u-v)B_a^{(\hat{n})}(u)\widehat{R}_{ab}^{(\hat{n},\hat{n})}(\tilde{v}-u)A_b^{(\hat{n})}(v)$$
$$+ \frac{P_{ab}^{(\hat{n},\hat{n})}B_a^{(\hat{n})}(v)\widehat{R}_{ab}^{(\hat{n},\hat{n})}(\tilde{v}-u)A_b^{(\hat{n})}(u)}{u-v} \mp \frac{B_b^{(\hat{n})}(v)Q_{ab}^{(\hat{n},\hat{n})}D_a^{(\hat{n})}(u)}{u-\tilde{v}}, \qquad (19)$$

$$R_{ab}^{(\hat{n},\hat{n})}(u-v)B_a^{(\hat{n})}(u)\widehat{R}_{ab}^{(\hat{n},\hat{n})}(\tilde{v}-u)B_b^{(\hat{n})}(v) = B_b^{(\hat{n})}(v)\widehat{R}_{ab}^{(\hat{n},\hat{n})}(\tilde{v}-u)B_a^{(\hat{n})}(u)R_{ab}^{(\hat{n},\hat{n})}(u-v), \qquad (20)$$

$$R_{ab}^{(\hat{n},\hat{n})}(u-v)A_a^{(\hat{n})}(u)A_b^{(\hat{n})}(v) - A_b^{(\hat{n})}(v)A_a^{(\hat{n})}(u)R_{ab}^{(\hat{n},\hat{n})}(u-v)$$
$$= \mp \frac{R_{ab}^{(\hat{n},\hat{n})}(u-v)B_a^{(\hat{n})}(u)Q_{ab}^{(\hat{n},\hat{n})}C_b^{(\hat{n})}(v) - B_b^{(\hat{n})}(v)Q_{ab}^{(\hat{n},\hat{n})}C_a^{(\hat{n})}(u)R_{ab}^{(\hat{n},\hat{n})}(u-v)}{u-\tilde{v}}, \qquad (21)$$

$$C_a^{(\hat{n})}(u)A_b^{(\hat{n})}(v) = A_b^{(\hat{n})}(v)\widehat{R}_{ab}^{(\hat{n},\hat{n})}(\tilde{v}-u)C_a^{(\hat{n})}(u)R_{ab}^{(\hat{n},\hat{n})}(u-v)$$
$$+ \frac{P_{ab}^{(\hat{n},\hat{n})}A_a^{(\hat{n})}(u)\widehat{R}_{ab}^{(\hat{n},\hat{n})}(\tilde{v}-u)C_b^{(\hat{n})}(v)}{u-v} \mp \frac{D_a^{(\hat{n})}(u)Q_{ab}^{(\hat{n},\hat{n})}C_b^{(\hat{n})}(v)}{u-\tilde{v}}, \qquad (22)$$

and

$$\widehat{D}^{(\hat{n})}(\tilde{u}) = A^{(\hat{n})}(u) \pm \frac{A^{(\hat{n})}(u)-A^{(\hat{n})}(\tilde{u})}{u-\tilde{u}}, \qquad \pm\widehat{B}^{(\hat{n})}(\tilde{u}) = B^{(\hat{n})}(u) \pm \frac{B^{(\hat{n})}(u)-B^{(\hat{n})}(\tilde{u})}{u-\tilde{u}}. \qquad (23)$$

Here indices $a$ and $b$ label two distinct copies of $\mathrm{End}(\mathbb{C}^{\hat{n}})$, and $\widehat{D}^{(\hat{n})}(\tilde{u})$, $\widehat{B}^{(\hat{n})}(\tilde{u})$ are $\omega$-transposed matrices. Taking matrix coefficients of (19)–(23) one obtains relations among generating series that coincide with those given by the defining relations (13) and (14).

*Remark* 2.1. In the $\hat{n} = n+1$ case all four operators in (18) are "overlapping". For example, when $N = 3$, we have $\hat{n} = n+1 = 2$ giving

$$A^{(\hat{n})}(u) = \begin{pmatrix} s_{11}(u) & s_{12}(u) \\ s_{21}(u) & s_{22}(u) \end{pmatrix}, \qquad B^{(\hat{n})}(u) = \begin{pmatrix} s_{12}(u) & s_{13}(u) \\ s_{22}(u) & s_{23}(u) \end{pmatrix},$$
$$C^{(\hat{n})}(u) = \begin{pmatrix} s_{21}(u) & s_{22}(u) \\ s_{31}(u) & s_{32}(u) \end{pmatrix}, \qquad D^{(\hat{n})}(u) = \begin{pmatrix} s_{22}(u) & s_{23}(u) \\ s_{32}(u) & s_{33}(u) \end{pmatrix}.$$

We will mostly be interested in the $A$ and $B$ operators. The $A$ operator will be used to construct a transfer matrix of the spin chain and the $B$ operator will be used to construct creation operators. Both $A$ and $B$ operators include generating series $s_{i\hat{n}}(u)$ with $1 \le i \le n$ associated with the short root of $\mathfrak{so}_{2n+1}$. These series will be used to construct level-$n$ creation operator and should only be considered as elements of the $B$ operator. Likewise, the "middle" generating series $s_{\hat{n}\hat{n}}(u)$ is also included in both $A$ and $B$ operators (and $C$ and $D$), but should only be considered as an element of the $A$ operator. These issues will be resolved by restricting to the upper-left $(n-1) \times (n-1)$-dimensional submatrix of the $A$ operator (such a restriction is compatible with the $AB$ exchange relation, see Lemma 3.5) and by explicitly computing the action of $s_{\hat{n}\hat{n}}(u)$ on level-$n$ Bethe vectors (see Lemma 3.8).

## 3 Bethe Ansatz

### 3.1 Quantum space

We study spin chains with the full quantum space given by

$$L^{(n)} := L(\boldsymbol{\lambda}^{(1)}) \otimes \cdots \otimes L(\boldsymbol{\lambda}^{(\ell)}) \otimes M(\boldsymbol{\mu}), \qquad (24)$$

where $\ell \in \mathbb{N}$ is the length of the chain, each $L(\boldsymbol{\lambda}^{(i)})$ and $M(\boldsymbol{\mu})$ are finite-dimensional irreducible highest-weight representations of $\mathfrak{gl}_N$ and $\mathfrak{g}_N$, respectively, and the $N$-tuples $\boldsymbol{\lambda}^{(1)}$ and $\boldsymbol{\mu}$ are their highest weights. We will say that $L^{(n)}$ is a *level-n quantum space*.

The space $L^{(n)}$ can be equipped with a structure of a left $Y^\pm(\mathfrak{g}_N)$-module as follows. Introduce Lax operators

$$\mathcal{L}^{(N)}(u) := \sum_{i,j} E_{ij}^{(N)} \otimes (\delta_{ij} - u^{-1} e_{ji}), \tag{25}$$

$$\mathcal{M}^{(N)}(u) := \sum_{i,j} E_{ij}^{(N)} \otimes (\delta_{ij} - u^{-1} f_{ji}). \tag{26}$$

Choose an $\ell$-tuple $\boldsymbol{c} = (c_1, \dots, c_\ell)$ of distinct complex parameters. Then for any $\xi \in L^{(n)}$ the action of $Y^\pm(\mathfrak{gl}_N)$ is given by

$$S_a^{(N)}(u) \cdot \xi = \overrightarrow{\prod_i} \mathcal{L}_{ai}^{(N)}(u - c_i) \, \mathcal{M}_{a,\ell+1}^{(N)}(u + (\rho \pm 1)/2) \overleftarrow{\prod_i} \widehat{\mathcal{L}}_{ai}^{(N)}(\tilde{u} - c_i) \cdot \xi, \tag{27}$$

where the subscript $a$ labels the matrix space of $S^{(N)}$ and the subscripts $i = 1, \dots, \ell$ and $\ell + 1$ label the individual tensorands of the space $L^{(n)}$, which we call *bulk* and *boundary* quantum spaces. The bulk spaces are *evaluation representations* of $Y(\mathfrak{gl}_N)$ and the boundary space is an *evaluation representation* of $Y^\pm(\mathfrak{gl}_N)$. Moreover, since $L^{(n)}$ is finite-dimensional, the formal variable $u$ can be evaluated to any complex number, not equal to any $c_i$, $\tilde{c}_i$, and $-(\rho \pm 1)/2$.

Let $1_{\boldsymbol{\lambda}^{(i)}}$ and $1_{\boldsymbol{\mu}}$ denote highest-weight vectors of $L(\boldsymbol{\lambda}^{(i)})$ and $M(\boldsymbol{\mu})$, respectively. Set

$$\eta := 1_{\boldsymbol{\lambda}^{(1)}} \otimes \cdots \otimes 1_{\boldsymbol{\lambda}^{(\ell)}} \otimes 1_{\boldsymbol{\mu}}. \tag{28}$$

Then $s_{ij}(u) \cdot \eta = 0$ if $i > j$ and $s_{ii}(u) \cdot \eta = \mu_i(u)\eta$ where

$$\mu_i(u) := \frac{u + (\rho \pm 1)/2 - \mu_i}{u + (\rho \pm 1)/2} \prod_{j \leq \ell} \frac{u - c_j - \lambda_i^{(j)}}{u - c_i} \cdot \frac{\tilde{u} - c_j - \lambda_i^{(j)}}{\tilde{u} - c_i}. \tag{29}$$

Note that $\mu_{N-i+1} = -\mu_i$ and $\mu_{\hat{n}} = 0$ when $\hat{n} = n + 1$.

An important property of $L^{(n)}$ is that the subspace $(L^{(n)})^0 \subset L^{(n)}$, annihilated by $s_{ij}(u)$ with $i > n$, $j \leq \hat{n}$ and $i > j$, is isomorphic to an $(\ell + 1)$-fold tensor product of irreducible $\mathfrak{gl}_n$ representations. Its subspace $(L^{(n)})^1 \subset (L^{(n)})^0$, annihilated by $s_{ni}(u)$ with $i < n$, is isomorphic to an $(\ell + 1)$-fold tensor product of irreducible $\mathfrak{gl}_{n-1}$ representations. This can be continued to give the following chain of (sub)spaces

$$L^{(n)} \supset (L^{(n)})^0 \supset (L^{(n)})^1 \supset \cdots \supset (L^{(n)})^{n-1}, \tag{30}$$

where $(L^{(n)})^0, (L^{(n)})^1, \dots, (L^{(n)})^{n-1}$ are isomorphic to $(\ell+1)$-fold tensor products of irreducible finite-dimensional $\mathfrak{gl}_n, \mathfrak{gl}_{n-1}, \dots, \mathfrak{gl}_2$ representations, respectively. This property ensures that nested algebraic Bethe Ansatz techniques can be applied.

## 3.2 Nested quantum spaces

Choose an $n$-tuple $\boldsymbol{m} := (m_1, \dots, m_n)$ of non-negative integers, the excitation (magnon) numbers. For each $m_k$ assign an $m_k$-tuple $\boldsymbol{u}^{(k)} := (u_1^{(k)}, \dots, u_{m_k}^{(k)})$ of complex parameters (off-shell Bethe roots) and an $m_k$-tuple $\boldsymbol{a}^k := (a_1^k, \dots, a_{m_k}^k)$ of labels, except that for $m_n$ we assign two $m_n$-tuples of labels, $\dot{\boldsymbol{a}} := (\dot{a}_1, \dots, \dot{a}_{m_n})$ and $\ddot{\boldsymbol{a}} := (\ddot{a}_1, \dots, \ddot{a}_{m_n})$. We will often use the following shorthand notation:

$$\boldsymbol{u}^{(k\dots l)} := (\boldsymbol{u}^{(k)}, \boldsymbol{u}^{(k+1)}, \dots, \boldsymbol{u}^{(l)}). \tag{31}$$

We will assume that $\boldsymbol{u}^{(k\dots k)} = \boldsymbol{u}^k$ and that $\boldsymbol{u}^{(k\dots l)}$ is an empty tuple if $k > l$ so that, for instance,

$$f(\boldsymbol{u}^{(1\dots k)}, \boldsymbol{u}^{(k\dots l)}) = f(\boldsymbol{u}^{(1\dots k)}),$$

for any function or operator $f$ when $k \geq l$. For any tuples $\boldsymbol{u}$ and $\boldsymbol{v}$ of complex parameters we set

$$f^{\pm}(u_i, v_j) := \frac{u_i - v_j \pm 1}{u_i - v_j}, \qquad f^{\pm}(\boldsymbol{u}, \boldsymbol{v}) := \prod_{i,j} f^{\pm}(u_i, v_j), \qquad \frac{1}{\boldsymbol{u} - \boldsymbol{v}} := \prod_{i,j} \frac{1}{u_i - v_j}, \quad (32)$$

where the products are over all admissible indices $i$ and $j$.

Let $V_{a_i^k}^{(k)}$ denote a copy of $\mathbb{C}^k$ labelled by "$a_i^k$" and let $W_{\boldsymbol{a}^k}^{(k)}$ be defined by

$$W_{\boldsymbol{a}^k}^{(k)} := V_{a_1^k}^{(k)} \otimes \cdots \otimes V_{a_{m_k}^k}^{(k)} \cong (\mathbb{C}^k)^{\otimes m_k}. \quad (33)$$

Labels $a_i^k$ will be used to trace the action of matrix operators. We illustrate this property with an example. Let $\xi = \xi_{a_1^k} \otimes \cdots \otimes \xi_{a_{m_k}^k} \in W_{\boldsymbol{a}^k}^{(k)}$ and let $M_{a_j^k}^{(k)} \in \mathrm{End}(V_{a_j^k}^{(k)})$ be a matrix operator acting in the space labelled $a_j^k$. Then

$$M_{a_j^k}^{(k)} \xi = \xi_{a_1^k} \otimes \cdots \otimes \xi_{a_{j-1}^k} \otimes \left( M_{a_j^k}^{(k)} \xi_{a_j^k} \right) \otimes \xi_{a_{j+1}^k} \otimes \cdots \otimes \xi_{a_{m_k}^k}.$$

Let $V_{\dot{a}_i}^{(\hat{n})}, V_{\ddot{a}_i}^{(\hat{n})} \cong \mathbb{C}^{\hat{n}}$ and $W_{\dot{\boldsymbol{a}}}^{(\hat{n})}, W_{\ddot{\boldsymbol{a}}}^{(\hat{n})} \cong (\mathbb{C}^{\hat{n}})^{\otimes m_n}$ be defined analogously to (33). We define a *level-$(n-1)$ quantum space* by

$$L^{(n-1)} := W_{\dot{\boldsymbol{a}}}^{(\hat{n})} \otimes W_{\ddot{\boldsymbol{a}}}^{(\hat{n})} \otimes (L^{(n)})^0. \quad (34)$$

When $\hat{n} = n + 1$, we additionally introduce "reduced" vector spaces

$$\overline{W}_{\dot{\boldsymbol{a}}}^{(\hat{n})} := \overline{V}_{\dot{a}_1}^{(\hat{n})} \otimes \cdots \otimes \overline{V}_{\dot{a}_{m_n}}^{(\hat{n})}, \qquad \overline{W}_{\ddot{\boldsymbol{a}}}^{(\hat{n})} := \overline{V}_{\ddot{a}_1}^{(\hat{n})} \otimes \cdots \otimes \overline{V}_{\ddot{a}_{m_n}}^{(\hat{n})}, \quad (35)$$

where

$$\overline{V}_{\dot{a}_i}^{(\hat{n})} := \mathrm{span}_{\mathbb{C}}\{E_j^{(\hat{n})} : 2 \leq j \leq \hat{n}\} \subset V_{\dot{a}_i}^{(\hat{n})}, \qquad \overline{V}_{\ddot{a}_i}^{(\hat{n})} := \mathrm{span}_{\mathbb{C}}\{E_1^{(\hat{n})}\} \subset V_{\ddot{a}_i}^{(\hat{n})}. \quad (36)$$

Specifically, $\overline{W}_{\dot{\boldsymbol{a}}}^{(\hat{n})}$ is isomorphic to $(\mathbb{C}^n)^{\otimes m_n}$ and $\overline{W}_{\ddot{\boldsymbol{a}}}^{(\hat{n})}$ a 1-dimensional vector space. We then define a *reduced level-$(n-1)$ quantum space* by

$$\overline{L}^{(n-1)} := \overline{W}_{\dot{\boldsymbol{a}}}^{(\hat{n})} \otimes \overline{W}_{\ddot{\boldsymbol{a}}}^{(\hat{n})} \otimes (L^{(n)})^0 \subset L^{(n-1)}. \quad (37)$$

The spaces $L^{(n-1)}$ and $\overline{L}^{(n-1)}$ will serve as the full (nested) quantum spaces of the $Y(\mathfrak{gl}_n)$-based models obtained after the first step of nesting in the even and odd cases, respectively; see Remark 3.3.

Then, for each $k = n - 2, n - 3, \dots, 1$ we define a *level-$k$ quantum space* by

$$L^{(k)} := W_{\boldsymbol{a}^{k+1}}^{(k+1)} \otimes (L^{(k+1)})^0, \quad (38)$$

where $(L^{(k+1)})^0$ is a *level-$(k+1)$ vacuum subspace* given by

$$(L^{(k+1)})^0 := (W_{\boldsymbol{a}^{k+2}}^{(k+2)})^0 \otimes \cdots \otimes (W_{\boldsymbol{a}^{n-1}}^{(n-1)})^0 \otimes (W_{\dot{\boldsymbol{a}}}^{(\hat{n})})^0 \otimes (W_{\ddot{\boldsymbol{a}}}^{(\hat{n})})^0 \otimes (L^{(n)})^{n-k-1} \subset L^{(k+1)}, \quad (39)$$

where

$$(W_{\boldsymbol{a}^{k+2}}^{(k+2)})^0 \subset W_{\boldsymbol{a}^{k+2}}^{(k+2)}, \quad \dots, \quad (W_{\boldsymbol{a}^{n-1}}^{(n-1)})^0 \subset W_{\boldsymbol{a}^{n-1}}^{(n-1)}, \quad (W_{\dot{\boldsymbol{a}}}^{(\hat{n})})^0 \subset W_{\dot{\boldsymbol{a}}}^{(\hat{n})}, \quad (W_{\ddot{\boldsymbol{a}}}^{(\hat{n})})^0 \subset W_{\ddot{\boldsymbol{a}}}^{(\hat{n})},$$

are 1-dimensional subspaces spanned by vectors

$$E_1^{(k+2)} \otimes \cdots \otimes E_1^{(k+2)}, \quad \dots, \quad E_1^{(n-1)} \otimes \cdots \otimes E_1^{(n-1)}, \quad E_{\dot{1}}^{(\hat{n})} \otimes \cdots \otimes E_{\dot{1}}^{(\hat{n})}, \quad E_1^{(\hat{n})} \otimes \cdots \otimes E_1^{(\hat{n})},$$

respectively. When $\hat{n} = n + 1$, note that $(L^{(n-1)})^0 \subset \overline{L}^{(n-1)}$. Moreover, $(L^{(k+1)})^0 \cong (L^{(n)})^{n-k-1}$ for $1 \leq k \leq n - 2$. The spaces $L^{(k)}$ will serve as the full (nested) quantum spaces of the $Y(\mathfrak{gl}_{k+1})$-based models obtained after $n - k$ steps of nesting.

## 3.3 Monodromy matrices

We will say that the matrix $S^{(N)}(u)$, acting in the space $L^{(n)}$ via (27), is a *level-$n$ monodromy matrix*. In this setting, we will treat $u$ as a non-zero complex number not equal to any $c_i$, $\tilde{c}_i$ and $-(\rho \pm 1)/2$. We define a *level-$(n-1)$ nested monodromy matrix*, acting in the space $L^{(n-1)}$, by

$$T_a^{(\hat{n})}(v; \boldsymbol{u}^{(n)}) := \prod_{i \leq m_n}^{\leftarrow} \widehat{R}_{\dot{a}_i a}^{(\hat{n},\hat{n})}(u_i^{(n)} - v) \prod_{i \leq m_n}^{\leftarrow} \widehat{R}_{\ddot{a}_i a}^{(\hat{n},\hat{n})}(\tilde{u}_i^{(n)} - v) A_a^{(\hat{n})}(v). \tag{40}$$

When $\hat{n} = n + 1$, we introduce a *reduced level-$(n-1)$ nested monodromy matrix*, acting in the space $\overline{L}^{(n-1)}$, by

$$\overline{T_a^{(n)}}(v; \boldsymbol{u}^{(n)}) := \prod_{i \leq m_n}^{\leftarrow} \overline{\widehat{R}_{\dot{a}_i a}^{(n,n)}}(u_i^{(n)} - v) \left[A_a^{(\hat{n})}(v)\right]^{(n)}, \tag{41}$$

where $\overline{\widehat{R}_{\dot{a}_i a}^{(n,n)}}$ is the restriction of $\widehat{R}_{\dot{a}_i a}^{(n,n)}$ to $\overline{V}_{\dot{a}_i}^{(\hat{n})} \otimes V_a^{(n)} \subset V_{\dot{a}_i}^{(\hat{n})} \otimes V_a^{(\hat{n})}$ (recall (11) and (36)), and the notation $[\ ]^{(n)}$ means the restriction to the upper-left $(n \times n)$-dimensional submatrix; this notation will be used throughout the manuscript.

**Lemma 3.1.** *When $\hat{n} = n + 1$, in the space $\overline{L}^{(n-1)}$ we have the equality of operators*

$$\left[T_a^{(\hat{n})}(v; \boldsymbol{u}^{(n)})\right]^{(n)} = \overline{T_a^{(n)}}(v; \boldsymbol{u}^{(n)}). \tag{42}$$

*Moreover, the space $\overline{L}^{(n-1)}$ is stable under the action of $\overline{T_a^{(n)}}(v; \boldsymbol{u}^{(n)})$.*

*Proof.* From (11) observe that

$$\left[\widehat{R}_{ba}^{(\hat{n},\hat{n})}(v)\right]_{kl} E_j^{(\hat{n})} = \delta_{kl} E_j^{(\hat{n})} - v^{-1} \delta_{\hat{n}-l+1,j} E_{\hat{n}-k+1}^{(\hat{n})}, \tag{43}$$

where $[\ ]_{kl}$ selects the $(k, l)$-th matrix element of $\widehat{R}_{ba}^{(\hat{n},\hat{n})}$ in the $a$-space; this notation will be used throughout the manuscript. Therefore, for any $1 \leq k, l \leq n$ and any $\eta \in \overline{W}_{\dot{a}}^{(\hat{n})}$, $\zeta \in \overline{W}_{\ddot{a}}^{(\hat{n})}$, $\xi \in (L^{(n)})^0$, viz. (37), we have

$$\begin{aligned}
&\left[T_a^{(\hat{n})}(v; \boldsymbol{u}^{(n)})\right]_{kl} \cdot \eta \otimes \zeta \otimes \xi \\
&= \sum_{p,r} \left[\prod_{i \leq m_n}^{\leftarrow} \widehat{R}_{\dot{a}_i a}^{(\hat{n},\hat{n})}(u_i^{(n)} - v)\right]_{kp} \cdot \eta \otimes \left[\prod_{i \leq m_n}^{\leftarrow} \widehat{R}_{\ddot{a}_i a}^{(\hat{n},\hat{n})}(\tilde{u}_i^{(n)} - v)\right]_{pr} \cdot \zeta \otimes s_{rl}(v) \cdot \xi \\
&= \sum_{p \leq n} \left[\prod_{i \leq m_n}^{\leftarrow} \widehat{R}_{\dot{a}_i a}^{(\hat{n},\hat{n})}(u_i^{(n)} - v)\right]_{kp} \cdot \eta \otimes \zeta \otimes s_{pl}(v) \cdot \xi,
\end{aligned} \tag{44}$$

since $s_{\hat{n}l}(v) \cdot \xi = 0$ by definition of $(L^{(n)})^0$, and, by (43),

$$\left[\prod_{i \leq m_n}^{\leftarrow} \widehat{R}_{\ddot{a}_i a}^{(\hat{n},\hat{n})}(\tilde{u}_i^{(n)} - v)\right]_{pr} \cdot \zeta = \delta_{pr} \zeta,$$

when $r < \hat{n}$ because $\zeta$ is a scalar multiple of $E_1^{(\hat{n})} \otimes \cdots \otimes E_1^{(\hat{n})}$. But

$$\left[\prod_{i \leq m_n}^{\leftarrow} \widehat{R}_{\dot{a}_i a}^{(\hat{n},\hat{n})}(u_i^{(n)} - v)\right]_{kp} \cdot \eta \notin \overline{W}_{\dot{a}}^{(\hat{n})},$$

when $k, p \leq n$ only if the product includes $\left[\widehat{R}_{\dot{a}_i a}^{(\hat{n},\hat{n})}(u_i^{(n)} - v)\right]_{\hat{n}r}$ with $r \leq n$, but then it must also include $\left[\widehat{R}_{\dot{a}_i a}^{(\hat{n},\hat{n})}(u_i^{(n)} - v)\right]_{r\hat{n}}$ which acts by zero on $\eta$ since the spaces $\overline{V}_{\dot{a}_i}^{(\hat{n})}$ have no $E_1^{(\hat{n})}$'s. Thus

$$\left[\prod_{i \leq m_n}^{\leftarrow} \widehat{R}_{\dot{a}_i a}^{(\hat{n},\hat{n})}(u_i^{(n)} - v)\right]_{kp} \cdot \eta = \left[\prod_{i \leq m_n}^{\leftarrow} \overline{\widehat{R}_{\dot{a}_i a}^{(\hat{n},\hat{n})}}(u_i^{(n)} - v)\right]_{kp} \cdot \eta \in \overline{W}_{\dot{a}}^{(\hat{n})}, \tag{45}$$

implying (42). To prove the second part of the claim, notice that $(L^{(n)})^0$ is stable under the action of $s_{pl}(u)$ with $1 \leq p, l \leq n$. Indeed, by definition, it is the subspace of $L^{(n)}$ annihilated by $s_{\bar{i}j}(u)$ with $\bar{i} > n$, $j \leq \hat{n}$ and $\bar{i} > j$. Assuming $1 \leq i, j, k, l \leq n$, (13) gives $s_{\bar{i}j}(u) s_{kl}(v) = 0$ in the space $(L^{(n)})^0$ thus proving its stability. The stability of $\overline{L}^{(n-1)}$ under the action of $\overline{T_a^{(n)}}(v; \boldsymbol{u}^{(n)})$ then follows immediately from (44) and (45). $\qquad\square$

Next, for each $k = n-1, n-2, \ldots, 2$, we define a *level-$(k-1)$ nested monodromy matrix*, acting in the space $L^{(k-1)}$, by

$$T_a^{(k)}(v; \boldsymbol{u}^{(k\ldots n)}) := \prod_{i \leq m_k}^{\leftarrow} \widehat{R}_{a_i^k a}^{(k,k)}(u_i^{(k)} - v) \left[ T_a^{(k+1)}(v; \boldsymbol{u}^{(k+1\ldots n)}) \right]^{(k)}, \tag{46}$$

where $T_a^{(k+1)}$ should be $\overline{T_a^{(k+1)}}$ when $\hat{n} = n+1$ and $k = n$.

**Lemma 3.2.** *For each $2 \leq k \leq n$, the space $L^{(k-1)}$ is stable under the action of $T_a^{(k)}(v; \boldsymbol{u}^{(k\ldots n)})$ and*

$$R_{ab}^{(k,k)}(v - w) T_a^{(k)}(v; \boldsymbol{u}^{(k\ldots n)}) T_b^{(k)}(w; \boldsymbol{u}^{(k\ldots n)}) = T_b^{(k)}(w; \boldsymbol{u}^{(k\ldots n)}) T_a^{(k)}(v; \boldsymbol{u}^{(k\ldots n)}) R_{ab}^{(k,k)}(v - w), \tag{47}$$

*in this space, except, when $\hat{n} = n+1$ and $k = n$, $L^{(k-1)}$ should be $\overline{L}^{(k-1)}$ and $T^{(k)}$ should be $\overline{T^{(k)}}$.*

*Proof.* When $k = n$ and $\hat{n} = n$, this was shown in Proposition 3.13 in [12]. When $k = n$ and $\hat{n} = n+1$, the first part of the claim follows from Lemma 3.1; the second part follows from the observation that

$$R_{ab}^{(n,n)}(u - v) \left[A_a^{(\hat{n})}(u)\right]^{(n)} \left[A_b^{(\hat{n})}(v)\right]^{(n)} = \left[A_b^{(\hat{n})}(v)\right]^{(n)} \left[A_a^{(\hat{n})}(u)\right]^{(n)} R_{ab}^{(n,n)}(u - v), \tag{48}$$

in the space $\overline{L}^{(n-1)}$ and application of the transposed quantum Yang-Baxter equation (12). The (48) follows from (21) or directly from (13) upon restricting to $1 \leq i, j, k, l \leq n$. The $k < n$ cases then follow by the standard arguments. $\qquad\square$

*Remark* 3.3. Lemma 3.2 together with (40), (41) say that $Y^{\pm}(\mathfrak{gl}_{2n})$- and $Y^+(\mathfrak{gl}_{2n+1})$-based models, after the first step of nesting, are equivalent to $Y(\mathfrak{gl}_n)$-based models with off-shell Bethe roots given by $\boldsymbol{v}^{(1\ldots n-2)} := \boldsymbol{u}^{(1\ldots n-2)}$ and $\boldsymbol{v}^{(n)} := (\boldsymbol{u}^{(n)}, \tilde{\boldsymbol{u}}^{(n)})$ in the even case, and $\boldsymbol{v}^{(n)} := \boldsymbol{u}^{(n)}$ in the odd case. This property will be explored in Section 4.

## 3.4 Creation operators

We define a *level-$n$ creation operator* by

$$\mathscr{B}^{(n)}(\boldsymbol{u}^{(n)}) := \prod_{1 \leq i \leq m_n}^{\leftarrow} \left( \mathsf{L}_{\dot{a}_i \ddot{a}_i}^{(n)}(u_i^{(n)}) \prod_{i < j \leq m_n}^{\rightarrow} \frac{R_{\dot{a}_i \ddot{a}_j}^{(\hat{n},\hat{n})}(\tilde{u}_i^{(n)} - u_j^{(n)})}{(f^-(\tilde{u}_i^{(n)}, u_j^{(n)}))^{\delta_{\hat{n}n}}} \right), \tag{49}$$

where

$$\mathsf{L}_{\dot{a}_i \ddot{a}_i}^{(n)}(u_i^{(n)}) := \sum_{k,l \leq \hat{n}} (E_k^{(\hat{n})})^* \otimes (E_l^{(\hat{n})})^* \otimes \left[ B_a^{(\hat{n})}(u_i^{(n)}) \right]_{\bar{k},l} \in (V_{\dot{a}_i}^{(\hat{n})})^* \otimes (V_{\ddot{a}_i}^{(\hat{n})})^* \otimes \text{End}(L^{(n)}), \tag{50}$$

and $B_a^{(\hat{n})}(u_i^{(n)})$ is the $B$-block of the operator in the right hand side of (27). The $R$-matrices in (49) are necessary for the wanted order of the $\widehat{R}$-matrices in (40), which in turn is necessary for Lemma 3.2 to hold. The denominator is an overall normalisation factor.

From (49) it is clear that $\mathscr{B}^{(n)}(\boldsymbol{u}^{(n)})$ satisfies the recurrence relation

$$\mathscr{B}^{(n)}(\boldsymbol{u}^{(n)}) = \lfloor_{\dot{a}_{m_n}\ddot{a}_{m_n}}^{(n)}(u_{m_n}^{(n)})\,\mathscr{B}^{(n)}(\boldsymbol{u}^{(n)}\backslash u_{m_n}^{(n)})\,\mathscr{R}^{(\hat{n})}(u_{m_n}^{(n)};\boldsymbol{u}^{(n)}\backslash u_{m_n}^{(n)}), \tag{51}$$

where $\mathscr{B}^{(n)}(\boldsymbol{u}^{(n)}\backslash u_{m_n}^{(n)})$ is defined via (49) except the ranges of products are $1 \le i < m_n$ and $i < j < m_n$, and

$$\mathscr{R}^{(\hat{n})}(u_{m_n}^{(n)};\boldsymbol{u}^{(n)}\backslash u_{m_n}^{(n)}) := \prod_{1 \le i < m_n}^{\leftarrow} \frac{R_{\dot{a}_i\ddot{a}_{m_n}}^{(\hat{n},\hat{n})}(\tilde{u}_i^{(n)} - u_{m_n}^{(n)})}{(f^-(\tilde{u}_i^{(n)}, u_{m_n}^{(n)}))^{\delta_{\hat{n}n}}}. \tag{52}$$

We will later meet operators $\mathscr{B}^{(n)}(\boldsymbol{u}^{(n)}\backslash u_l^{(n)})$ and $\mathscr{R}^{(\hat{n})}(u_l^{(n)};\boldsymbol{u}^{(n)}\backslash u_l^{(n)})$ for any $l$ that are defined analogously except $u_i^{(n)}$ (resp. $\tilde{u}_i^{(n)}$) should be replaced with $u_{i+1}^{(n)}$ (resp. $\tilde{u}_{i+1}^{(n)}$) for all $l \le i < m_n$.

Next, for each $k = n-1, n-2, \ldots, 1$ we define a *level-k creation operator* by

$$\mathscr{B}^{(k)}(\boldsymbol{u}^{(k)};\boldsymbol{u}^{(k+1\ldots n)}) := \prod_{1 \le i \le m_k}^{\leftarrow} \lfloor_{a_i^k}^{(k)}(u_i^{(k)};\boldsymbol{u}^{(k+1\ldots n)}), \tag{53}$$

where

$$\lfloor_{a_i^k}^{(k)}(u_i^{(k)};\boldsymbol{u}^{(k+1\ldots n)}) := \sum_{1 \le j \le k} (E_j^{(k)})_{a_i^k}^* \otimes \big[ T_a^{(k+1)}(u_i^{(k)};\boldsymbol{u}^{(k+1\ldots n)}) \big]_{\bar{j},k+1} \in (V_{a_i^k}^{(k)})^* \otimes \mathrm{End}(L^{(k)}). \tag{54}$$

Note that $T_a^{(n)}(u_i^{(n-1)};\boldsymbol{u}^{(n)})$ should be replaced with $\overline{T_a^{(n)}}(u_i^{(n-1)};\boldsymbol{u}^{(n)})$ when $\hat{n} = n+1$.

Parameters of creation operators may be permuted using the following standard result, which follows from (20); see Lemma 3.6 in [12].

**Lemma 3.4.** *The level-n creation operator satisfies*

$$\mathscr{B}^{(n)}(\boldsymbol{u}^{(n)}) = \mathscr{B}^{(n)}(\boldsymbol{u}_{i\leftrightarrow i+1}^{(n)})\check{R}_{\dot{a}_{i+1}\dot{a}_i}^{(\hat{n},\hat{n})}(u_i^{(n)} - u_{i+1}^{(n)})\check{R}_{\ddot{a}_{i+1}\ddot{a}_i}^{(\hat{n},\hat{n})}(u_{i+1}^{(n)} - u_i^{(n)}). \tag{55}$$

*For each $1 \le k \le n-1$ the level-k creation operator satisfies*

$$\mathscr{B}^{(k)}(\boldsymbol{u}^{(k)};\boldsymbol{u}^{(k+1\ldots n)}) = \mathscr{B}^{(k)}(\boldsymbol{u}_{i\leftrightarrow i+1}^{(k)};\boldsymbol{u}^{(k+1\ldots n)})\check{R}_{a_{i+1}^k a_i^k}^{(k,k)}(u_i^{(k)} - u_{i+1}^{(k)}). \tag{56}$$

*Here the "check" $\check{R}$-matrices are defined by*

$$\check{R}_{ab}^{(k,k)}(u) := \frac{u}{u-1} P_{ab}^{(k,k)} R_{ab}^{(k,k)}(u), \tag{57}$$

*and $\boldsymbol{u}_{i\leftrightarrow i+1}^{(k)}$ denotes the tuple $\boldsymbol{u}^{(k)}$ with parameters $u_i^{(k)}$ and $u_{i+1}^{(k)}$ interchanged.*

Recall the notation $\tilde{v} = -v - \rho$ and introduce the following notation for a symmetrised combination of functions or operators

$$\{f(v)\}^v := f(v) + f(\tilde{v}), \tag{58}$$

and a rational function

$$p(v) := 1 \pm \frac{1}{v - \tilde{v}}, \tag{59}$$

representing the right hand side of the symmetry relation (17). The Lemma below rephrases the results obtained in [12] in a compact form.

**Lemma 3.5.** *The AB exchange relation for the level-$n$ creation operator* (49) *is*

$$\{p(v)A_a^{(\hat{n})}(v)\}^v \, \mathscr{B}^{(n)}(\boldsymbol{u}^{(n)})$$

$$= \mathscr{B}^{(n)}(\boldsymbol{u}^{(n)}) \, \{p(v) \, T_a^{(\hat{n})}(v; \boldsymbol{u}^{(n)})\}^v$$

$$+ \sum_i \frac{1}{p(u_i^{(n)})} \left\{ \frac{p(v)}{u_i^{(n)} - v} \, \lfloor_{\dot{a}_{m_n} \ddot{a}_{m_n}}^{(n)}(v) \right\}^v \, \mathscr{B}^{(n)}(\boldsymbol{u}^{(n)} \backslash u_i^{(n)}) \, \mathscr{R}^{(\hat{n})}(u_i^{(n)}; \boldsymbol{u}^{(n)} \backslash u_i^{(n)})$$

$$\times \underset{w \to u_i^{(n)}}{\mathrm{Res}} \{p(w) \, T_a^{(\hat{n})}(w; \boldsymbol{u}_{\sigma_i}^{(n)})\}^w \overset{\rightarrow}{\prod_{j>i}} \check{R}_{\dot{a}_j \dot{a}_{j-1}}^{(\hat{n}, \hat{n})}(u_i^{(n)} - u_j^{(n)}) \check{R}_{\ddot{a}_j \ddot{a}_{j-1}}^{(\hat{n}, \hat{n})}(u_j^{(n)} - u_i^{(n)}), \quad (60)$$

*where $\boldsymbol{u}^{(n)} \backslash u_i^{(n)} := (u_1, \ldots, u_{i-1}, u_{i+1}, \ldots, u_{m_n})$ and $\boldsymbol{u}_{\sigma_i}^{(n)} := (u_1, \ldots, u_{i-1}, u_{i+1}, \ldots, u_{m_n}, u_i)$.*

*Proof.* From [12], relations (19) and (23) and properties of the $Q^{(\hat{n}, \hat{n})}$ matrix operator (viz. (8)) lead to the following exchange relation with a single creation operator

$$\{p(v)A_a^{(\hat{n})}(v)\}^v \lfloor_{\ddot{a}_i \dot{a}_i}^{(n)}(u_i^{(n)}) = \lfloor_{\ddot{a}_i \dot{a}_i}^{(n)}(u_i^{(n)}) \{p(v) \, T_a^{(\hat{n})}(v; u_i^{(n)})\}^v \qquad (61)$$

$$+ \frac{1}{p(u_i^{(n)})} \left\{ \frac{p(v)}{u_i^{(n)} - v} \, \lfloor_{\ddot{a}_i \dot{a}_i}^{(n)}(v) \right\}^v \underset{w \to u_i^{(n)}}{\mathrm{Res}} \{p(w) \, T_a^{(\hat{n})}(w; u_i^{(n)})\}^w,$$

where $T_a^{(\hat{n})}(v; u_i^{(n)}) = \widehat{R}_{\dot{a}_i a}(u_i^{(n)} - v) \widehat{R}_{\ddot{a}_i a}(\tilde{u}_i^{(n)} - v) A_a^{(\hat{n})}(v)$. We extend this to the creation operator for $m_n$ excitations by the standard argument. Indeed, the right hand side of the equation consists of terms with $A_a^{(\hat{n})}(u)$ as the rightmost operator, for $u$ equal to each of $v, u_1^{(n)}, \ldots, u_{m_n}^{(n)}$ and the corresponding tilded elements. Due to the $w \mapsto \tilde{w}$ symmetry of $\{p(w)A_a^{(\hat{n})}(w)\}^w$ in (61), it is sufficient to find those terms corresponding to $v, u_1^{(n)}, \ldots, u_{m_n}^{(n)}$.

First, we find the term corresponding to $v$ to be $\mathscr{B}^{(n)}(\boldsymbol{u}^{(n)}) \{p(v) \, T_a^{(\hat{n})}(v; \boldsymbol{u}^{(n)})\}^v$. The required order of $\widehat{R}$-matrices inside $T_a^{(\hat{n})}(v; \boldsymbol{u}^{(n)})$ is a result of Yang-Baxter moves through the $R$-matrices inside $\mathscr{B}^{(n)}(\boldsymbol{u}^{(n)})$. Using factorisation (51) we find the term corresponding to $u_{m_n}^{(n)}$ to be

$$\frac{1}{p(u_{m_n}^{(n)})} \left\{ \frac{p(v)}{u_{m_n}^{(n)} - v} \, \lfloor_{\dot{a}_{m_n} \ddot{a}_{m_n}}^{(n)}(v) \right\}^v \, \mathscr{B}^{(n)}(\boldsymbol{u}^{(n)} \backslash u_{m_n}^{(n)}) \mathscr{R}^{(\hat{n})}(u_{m_n}^{(n)}; \boldsymbol{u}^{(n)} \backslash u_{m_n}^{(n)}) \underset{w \to u_{m_n}^{(n)}}{\mathrm{Res}} \{p(w) \, T_a^{(\hat{n})}(w; \boldsymbol{u}^{(n)})\}^w.$$

$$(62)$$

This is because, after applying (61) to $\lfloor_{\dot{a}_{m_n} \ddot{a}_{m_n}}^{(n)}(u_{m_n}^{(n)})$, there can be no further contributions from the parameter-swapped term in the subsequent applications of (61).

To find the remaining terms, we note that Lemma 3.4 allows us to apply any permutation to the spectral parameters of the level-$n$ creation operator before applying the above argument. By applying the permutation $\sigma_i : (1, \ldots, i-1, i, i+1, \ldots, m_n) \mapsto (1 \ldots, i-1, i+1, \ldots, m_n, i)$, we obtain the term corresponding to $u_i^{(n)}$. $\qquad \square$

The Lemma below states $Y(\mathfrak{gl}_{k+1})$-based column-nested *AB* and *DB* exchange relations. They follow from Lemma 3.2 using standard arguments, see e.g. [5].

**Lemma 3.6.** *The exchange relation for the level-$k$ creation operator* (53) *is*

$$\left[ T_a^{(k+1)}(v; \boldsymbol{u}^{(k+1\ldots n)}) \right]^{(k)} \mathscr{B}^{(k)}(\boldsymbol{u}^{(k)}; \boldsymbol{u}^{(k+1\ldots n)})$$

$$= \mathscr{B}^{(k)}(\boldsymbol{u}^{(k)}; \boldsymbol{u}^{(k+1\ldots n)}) \, T_a^{(k)}(v; \boldsymbol{u}^{(k\ldots n)})$$

$$+ \sum_i \frac{1}{u_i^{(k)} - v} \, \lfloor_{a_{m_k}^k}^{(k)}(v; \boldsymbol{u}^{k+1\ldots n}) \, \mathscr{B}^{(k)}(\boldsymbol{u}^{(k)} \backslash u_i^{(k)}; \boldsymbol{u}^{(k+1\ldots n)})$$

$$\times \underset{w \to u_i^{(k)}}{\mathrm{Res}} \, T_a^{(k)}(w; (\boldsymbol{u}_{\sigma_i}^{(k)}, \boldsymbol{u}^{(k+1\ldots n)})) \overset{\rightarrow}{\prod_{j>i}} \check{R}_{a_j^k a_{j-1}^k}^{(k,k)}(u_i^{(k)} - u_j^{(k)}). \qquad (63)$$

*Moreover,*

$$
\begin{aligned}
\big[T_a^{(k+1)}&(v;\boldsymbol{u}^{(k+1\ldots n)})\big]_{k+1,k+1}\,\mathscr{B}^{(k)}(\boldsymbol{u}^{(k)};\boldsymbol{u}^{(k+1\ldots n)})\\
&=\mathscr{B}^{(k)}(\boldsymbol{u}^{(k)};\boldsymbol{u}^{(k+1\ldots n)})f^-(v;\boldsymbol{u}^{(k)})\big[T_a^{(k+1)}(v;\boldsymbol{u}^{(k+1\ldots n)})\big]_{k+1,k+1}\\
&\quad+\sum_i\frac{1}{u_i^{(k)}-v}\mathsf{L}_{a_{m_k}^k}^{(k)}(v;\boldsymbol{u}^{k+1\ldots n})\,\mathscr{B}^{(k)}(\boldsymbol{u}^{(k)}\backslash u_i^{(k)};\boldsymbol{u}^{(k+1\ldots n)})\\
&\quad\quad\times\operatorname*{Res}_{w\to u_i^{(k)}}f^-(w;\boldsymbol{u}^{(k)})\big[T_a^{(k+1)}(w;\boldsymbol{u}^{(k+1\ldots n)})\big]_{k+1,k+1}\overrightarrow{\prod_{j>i}}\check{R}_{a_j^k a_{j-1}^k}^{(k,k)}(u_i^{(k)}-u_j^{(k)}).
\end{aligned}
\tag{64}
$$

*Here we used the notation*

$$
\mathscr{B}^{(k)}(\boldsymbol{u}^{(k)}\backslash u_i^{(k)};\boldsymbol{u}^{(k+1\ldots n)})=\overleftarrow{\prod_{1\le j<i}}\mathsf{L}_{a_j^k}^{(k)}(u_j^{(k)};\boldsymbol{u}^{(k+1\ldots n)})\overleftarrow{\prod_{i\le j<m_k}}\mathsf{L}_{a_j^k}^{(k)}(u_{j+1}^{(k)};\boldsymbol{u}^{(k+1\ldots n)}).
$$

## 3.5 Bethe vectors

Recall (28) and define a *nested vacuum vector* by

$$
\eta^{\boldsymbol{m}}:=(E_1^{(1)})^{\otimes m_1}\otimes\cdots\otimes(E_1^{(n-1)})^{\otimes m_{n-1}}\otimes(E_{\hat{1}}^{(\hat{n})})^{\otimes m_n}\otimes(E_1^{(\hat{n})})^{\otimes m_n}\otimes\eta.
\tag{65}
$$

Note that $E_{\hat{1}}^{(\hat{n})}=E_2^{(n+1)}$ when $\hat{n}=n+1$. For each $1\le k\le n$ we define a *level-k* (off-shell) Bethe vector with (off-shell) Bethe roots $\boldsymbol{u}^{(1\ldots k)}$ and free parameters $\boldsymbol{u}^{(k+1\ldots n)}$ by

$$
\Psi(\boldsymbol{u}^{(1\ldots k)}\,|\,\boldsymbol{u}^{(k+1\ldots n)}):=\overleftarrow{\prod_{i\le k}}\mathscr{B}^{(i)}(\boldsymbol{u}^{(i)};\boldsymbol{u}^{(i+1\ldots n)})\cdot\eta^{\boldsymbol{m}}.
\tag{66}
$$

We will say that vector $\eta^{\boldsymbol{m}}$ is the *reference vector* of this Bethe vector. Note that, by construction, $\Psi(\boldsymbol{u}^{(1\ldots k)}\,|\,\boldsymbol{u}^{(k+1\ldots n)})\in L^{(k)}$ except when $\hat{n}=n+1$ and $k=n-1$, $\Psi(\boldsymbol{u}^{(1\ldots n-1)}\,|\,\boldsymbol{u}^{(n)})\in\overline{L}^{(n-1)}$.

The Lemma below follows by a repeated application of Lemma 3.4.

**Lemma 3.7.** *Bethe vector $\Psi(\boldsymbol{u}^{(1\ldots k)}\,|\,\boldsymbol{u}^{(k+1\ldots n)})$ is invariant under interchange of any two of its Bethe roots, $u_i^{(l)}$ and $u_j^{(l)}$, for all admissible i, j, and l.*

The last technical result that we will need is the action of $s_{\hat{n}\hat{n}}(v)=[S_a^{(N)}(v)]_{\hat{n}\hat{n}}$, viz. (27), on a level-$n$ Bethe vector, when $\hat{n}=n+1$. It is motivated by the following relation in $Y^+(\mathfrak{gl}_{2n+1})((u^{-1},v^{-1}))$ for $1\le k\le n$:

$$
s_{\hat{n}\hat{n}}(v)s_{k\hat{n}}(u)=f^-(v,u)f^+(v,\tilde{u})s_{k\hat{n}}(u)s_{\hat{n}\hat{n}}(v)-\left\{\frac{p(v)}{u-v}s_{k\hat{n}}(v)\right\}^v s_{\hat{n}\hat{n}}(u).
$$

We postpone the proof of the Lemma below to Section 4.3.

**Lemma 3.8.** *When $\hat{n}=n+1$,*

$$
\begin{aligned}
s_{\hat{n}\hat{n}}(v)\Psi(\boldsymbol{u}^{(1\ldots n)})&=f^-(v,\boldsymbol{u}^{(n)})f^+(v,\tilde{\boldsymbol{u}}^{(n)})\mu_{\hat{n}}(v)\Psi(\boldsymbol{u}^{(1\ldots n)})\\
&\quad+\sum_i\frac{1}{p(u_i^{(n)})}\left\{\frac{p(v)}{u_i^{(n)}-v}\mathsf{L}_{\dot{a}_{m_n}\ddot{a}_{m_n}}^{(n)}(v)\right\}^v\mathscr{B}^{(n)}(\boldsymbol{u}^{(n)}\backslash u_i^{(n)})\mathscr{R}^{(\hat{n})}(u_i^{(n)},\boldsymbol{u}^{(n)}\backslash u_i^{(n)})\\
&\quad\quad\times\operatorname*{Res}_{w\to u_i^{(n)}}f^-(w,\boldsymbol{u}^{(n)})f^+(w,\tilde{\boldsymbol{u}}^{(n)})\mu_{\hat{n}}(w)\Psi(\boldsymbol{u}^{(1\ldots n-1)}\,|\,\boldsymbol{u}_{\sigma_i}^{(n)}).
\end{aligned}
\tag{67}
$$

## 3.6 Transfer matrix and Bethe equations

We define the *transfer matrix* by

$$\tau(v) := \mathrm{tr}_a\left(M_a^{(N)} S_a^{(N)}(v)\right) = \mathrm{tr}_a\left(\alpha_a^{(\hat{n})}\left[M_a^{(N)}\right]^{(\hat{n})}\left\{p(v)A_a^{(\hat{n})}(v)\right\}^v\right), \tag{68}$$

where $M^{(N)} = \sum_i \varepsilon_i E_{ii}^{(N)}$ with $\varepsilon_i \in \mathbb{C}^\times$ satisfying $\varepsilon_{N-i+1} = \varepsilon_i$ is a twist matrix, a solution to the dual twisted reflection equation

$$
\begin{aligned}
(M_b^{(N)}(v))^{t_b}\,\widehat{R}_{ab}^{(N,N)}(u-\tilde{v})\,(M_a^{(N)}(u))^{t_a}\,R_{ab}^{(N,N)}(v-u)\\
= R_{ab}^{(N,N)}(v-u)\,(M_a^{(N)}(u))^{t_a}\,\widehat{R}_{ab}^{(N,N)}(u-\tilde{v})\,(M_b^{(N)}(v))^{t_b}\,,
\end{aligned} \tag{69}
$$

ensuring commutativity of transfer matrices, see Appendix A.2. Here $t$ denotes the usual matrix transposition. The right hand side of (68) follows from the symmetry relation (23); the $\alpha^{(\hat{n})}$ is a diagonal matrix with entries $\alpha_k = 1$ for all $k$ except $\alpha_{\hat{n}} = 1/2$ when $\hat{n} = n+1$, which resolves the double-counting of $s_{\hat{n}\hat{n}}(v)$.

**Theorem 3.9.** *The Bethe vector $\Psi(\boldsymbol{u}^{(1...n)})$ is an eigenvector of $\tau(v)$ with the eigenvalue*

$$\Lambda(v;\boldsymbol{u}^{(1...n)}) := \sum_{k \leq \hat{n}} \alpha_k \varepsilon_k \left\{p(v)\,\Gamma_k(v;\boldsymbol{u}^{(1...n)})\right\}^v, \tag{70}$$

*where $p(v)$ is given by (59) and*

$$\Gamma_k(v;\boldsymbol{u}^{(1...n)}) := f^-(v,\boldsymbol{u}^{(k-1)})\,f^+(v,\boldsymbol{u}^{(k)})\,\mu_k(v), \quad \text{for } k < \hat{n}, \tag{71}$$

*and*

$$\Gamma_{\hat{n}}(v;\boldsymbol{u}^{(1...n)}) := \begin{cases} f^-(v,\boldsymbol{u}^{(n-1)})\,f^+(v,\boldsymbol{u}^{(n)})\,f^+(v,\tilde{\boldsymbol{u}}^{(n)})\,\mu_n(v), & \text{when } \hat{n} = n, \\ f^-(v,\boldsymbol{u}^{(n)})\,f^+(v,\tilde{\boldsymbol{u}}^{(n)})\,\mu_{n+1}(v), & \text{when } \hat{n} = n+1, \end{cases} \tag{72}$$

*provided $\displaystyle\mathop{\mathrm{Res}}_{v \to u_j^{(k)}} \Lambda(v;\boldsymbol{u}^{(1...n)}) = 0$ for all admissible $k$ and $j$; these equations are called Bethe equations.*

*Proof.* When $\hat{n} = n$, this is a restatement of Theorems 4.3 and 4.4 in [12]. We will briefly recall the main steps of the proofs therein. They will provide a backbone of the proof of the more complex $\hat{n} = n+1$ case.

*The $\hat{n} = n$ case.* We start by noticing that

$$\prod_{i<j\leq m_n}^{\longrightarrow} \check{R}_{\dot{a}_j\dot{a}_{j-1}}^{(\hat{n},\hat{n})}(u_i^{(n)}-u_j^{(n)})\,\check{R}_{\ddot{a}_j\ddot{a}_{j-1}}^{(\hat{n},\hat{n})}(u_j^{(n)}-u_i^{(n)})\,\Psi(\boldsymbol{u}^{(1...n-1)}\,|\,\boldsymbol{u}^{(n)}) = \Psi(\boldsymbol{u}^{(1...n-1)}\,|\,\boldsymbol{u}_{\sigma_i}^{(n)}), \tag{73}$$

where $\boldsymbol{u}_{\sigma_i}^{(n)} = (u_1,\ldots,u_{i-1},u_{i+1},\ldots,u_{m_n},u_i)$. This identity is a consequence of Yang-Baxter moves and the identities

$$\check{R}_{\dot{a}_j\dot{a}_{j-1}}^{(\hat{n},\hat{n})}(u_i^{(n)}-u_j^{(n)}) \cdot \eta^{\boldsymbol{m}} = \eta^{\boldsymbol{m}}, \qquad \check{R}_{\ddot{a}_j\ddot{a}_{j-1}}^{(\hat{n},\hat{n})}(u_j^{(n)}-u_i^{(n)}) \cdot \eta^{\boldsymbol{m}} = \eta^{\boldsymbol{m}}, \tag{74}$$

which are computed using (43) and (65).

Next, using (66) and (68), we write

$$\tau(v)\,\Psi(\boldsymbol{u}^{(1...n)}) = \mathrm{tr}_a\left(\left[M_a^{(N)}\right]^{(n)}\left\{p(v)A_a^{(n)}(v)\right\}^v \mathscr{B}^{(n)}(\boldsymbol{u}^{(n)})\right)\Psi(\boldsymbol{u}^{(1...n-1)}\,|\,\boldsymbol{u}^{(n)})\,.$$

Lemma 3.5 allows us to exchange $\{p(v)A_a^{(n)}(v)\}^v$ and $\mathscr{B}^{(n)}(\boldsymbol{u}^{(n)})$. Applying (73) to the result gives

$$
\begin{aligned}
\tau(v)\Psi(\boldsymbol{u}^{(1\ldots n)}) =\ & \mathscr{B}^{(n)}(\boldsymbol{u}^{(n)})\,\tau(v;\boldsymbol{u}^{(n)})\,\Psi(\boldsymbol{u}^{(1\ldots n-1)}\,|\,\boldsymbol{u}^{(n)}) \\
& + \sum_i \frac{1}{p(u_i^{(n)})}\left\{\frac{p(v)}{u_i^{(n)}-v}\mathsf{L}_{\dot{a}_{m_n}\ddot{a}_{m_n}}^{(n)}(v)\right\}^v \mathscr{B}^{(n)}(\boldsymbol{u}^{(n)}\backslash u_i^{(n)}) \\
& \times \mathscr{R}^{(\hat{n})}(u_i^{(n)};\boldsymbol{u}^{(n)}\backslash u_i^{(n)})\operatorname*{Res}_{w\to u_i^{(n)}}\tau(w;\boldsymbol{u}_{\sigma_i}^{(n)})\Psi(\boldsymbol{u}^{(1\ldots n-1)}\,|\,\boldsymbol{u}_{\sigma_i}^{(n)}),
\end{aligned}
$$
(75)

where

$$
\tau(v;\boldsymbol{u}^{(n)}) := \operatorname{tr}_a\left(\left[M_a^{(N)}\right]^{(n)}\{p(v)\,T_a^{(n)}(v;\boldsymbol{u}^{(n)})\}^v\right),
$$

is a nested transfer matrix. It remains to compute the action of $\tau(v;\boldsymbol{u}^{(n)})$ on the nested Bethe vector $\Psi(\boldsymbol{u}^{(1\ldots n-1)}\,|\,\boldsymbol{u}^{(n)})\in L^{(n-1)}$. By Lemma 42, this can be achieved using $Y(\mathfrak{gl}_n)$-type nested Bethe Ansatz techniques assisted by Lemmas 3.6 and 3.7 leading to the eigenvalue (70) and the corresponding Bethe equations.

*The $\hat{n}=n+1$ case.* In this case we can not apply Lemma 3.5 directly since this would lead to the following nested transfer matrix

$$
\begin{aligned}
\tau(v;\boldsymbol{u}^{(n)}) &= \operatorname{tr}_a\left(\alpha_a^{(\hat{n})}\left[M_a^{(N)}\right]^{(\hat{n})}\{p(v)\,T_a^{(\hat{n})}(v;\boldsymbol{u}^{(n)})\}^v\right) \\
&= \operatorname{tr}_a\left(\left[M_a^{(N)}\right]^{(n)}\{p(v)\left[T_a^{(\hat{n})}(v;\boldsymbol{u}^{(n)})\right]^{(n)}\}^v\right) + \frac{1}{2}\varepsilon_{\hat{n}}\{p(v)\left[T_a^{(\hat{n})}(v;\boldsymbol{u}^{(n)})\right]_{\hat{n}\hat{n}}\}^v.
\end{aligned}
$$

However, the space $\overline{L}^{(n-1)}$ is not stable under the action of $\left[T_a^{(\hat{n})}(v;\boldsymbol{u}^{(n)})\right]_{\hat{n}\hat{n}}$. This is because $\left[T_a^{(\hat{n})}(v;\boldsymbol{u}^{(n)})\right]_{\hat{n}\hat{n}}$ has operators $\left[\widehat{R}_{\dot{a}_i a}^{(\hat{n},\hat{n})}(u_i^{(n)}-v)\right]_{\hat{n}j}$ with $j\le n$ that map $E_{\hat{n}-j+1}^{(\hat{n})}\in\overline{V}_{\dot{a}_i}^{\hat{n}}$ to $E_1^{(\hat{n})}$. Therefore, the right hand side of (60) would no longer represent a splitting into "wanted" and "unwanted" terms. A resolution of this issue is to single-out the operator $s_{\hat{n}\hat{n}}(v)$ from the very beginning. From (14) we know that $s_{\hat{n}\hat{n}}(\tilde{u})=s_{\hat{n}\hat{n}}(u)$ giving $\{p(v)s_{\hat{n}\hat{n}}(v)\}^v=2s_{\hat{n}\hat{n}}(v)$. This allows us to rewrite the transfer matrix as

$$
\tau(v) = \operatorname{tr}_a\left(\left[M_a^{(N)}\right]^{(n)}\{p(v)\left[A_a^{(\hat{n})}(v)\right]^{(n)}\}^v\right) + \varepsilon_{\hat{n}}s_{\hat{n}\hat{n}}(v).
$$
(76)

We can now use Lemma 3.5 to exchange $\{p(v)\left[A_a^{(\hat{n})}(v)\right]^{(n)}\}^v$ and $\mathscr{B}^{(n)}(\boldsymbol{u}^{(n)})$, and Lemma 3.8 to compute the action of $s_{\hat{n}\hat{n}}(v)$ on $\Psi(\boldsymbol{u}^{(1\ldots n)})$. This gives an expressions equivalent to (75) except the nested transfer matrix is now given by

$$
\tau(v;\boldsymbol{u}^{(n)}) := \operatorname{tr}_a\left(\left[M_a^{(N)}\right]^{(n)}\{p(v)\overline{T_a^{(n)}}(v;\boldsymbol{u}^{(n)})\}^v\right) + \varepsilon_{\hat{n}}f^-(v,\boldsymbol{u}^{(n)})f^+(v,\tilde{\boldsymbol{u}}^{(n)})\mu_{\hat{n}}(v).
$$

Here we invoked Lemma 3.1 to replace $\left[T_a^{(n)}(v;\boldsymbol{u}^{(n)})\right]^{(n)}$ with $\overline{T_a^{(n)}}(v;\boldsymbol{u}^{(n)})$. The remaining steps are the same as in the $\hat{n}=n$ case. $\qquad\square$

*Remark* 3.10. Let $(a_{ij})_{i,j=1}^n$ denote Cartan matrix of type $A_n$. Let $(b_{ij})_{i,j=1}^n$ denote a zero matrix when $\hat{n}=n+1$ and let $b_{nn}=2$, $b_{n-1,n}=b_{n,n-1}=-1$, and $b_{ij}=0$ otherwise, when $\hat{n}=n$. Set $m_0:=0$ and $z_j^{(k)}:=u_j^{(k)}-\frac{1}{2}(k-\rho)$. Then Bethe equations can be written as, for each $k<n$,

$$
\prod_{l=k-1}^{k+1}\prod_{i=1}^{m_l}\frac{z_j^{(k)}-z_i^{(l)}+\frac{1}{2}a_{kl}}{z_j^{(k)}-z_i^{(l)}-\frac{1}{2}a_{kl}}\cdot\frac{z_j^{(k)}+z_i^{(l)}+n+\frac{1}{2}b_{kl}}{z_j^{(k)}+z_i^{(l)}+n-\frac{1}{2}b_{kl}} = -\frac{\varepsilon_{k+1}}{\varepsilon_k}\cdot\frac{\mu_{k+1}(u_j^{(k)})}{\mu_k(u_j^{(k)})},
$$
(77)

$$
\frac{z_j^{(n)}+\frac{1}{2}(n+1)}{z_j^{(n)}+\frac{1}{2}(\hat{n}-1)}\prod_{l=n-1}^{n}\prod_{i=1}^{m_l}\frac{z_j^{(n)}-z_i^{(l)}+\frac{1}{2}a_{nl}}{z_j^{(n)}-z_i^{(l)}-\frac{1}{2}a_{nl}}\prod_{l=\hat{n}-1}^{n}\prod_{i=1}^{m_l}\frac{z_j^{(n)}+z_i^{(l)}+n+\frac{1}{2}b_{nl}}{z_j^{(n)}+z_i^{(l)}+\hat{n}-\frac{1}{2}b_{nl}} = -\frac{\varepsilon_{\hat{n}}}{\varepsilon_n}\cdot\frac{\mu_{\hat{n}}(\tilde{u}_j^{(n)})}{\mu_n(u_j^{(n)})}.
$$
(78)

### 3.7 Trace formula

Define the "master" creation operator

$$
\mathscr{B}_N(\boldsymbol{u}^{(1\ldots n)}) := \prod_{k \le n} \prod_{j<i} \frac{1}{f^+(u_j^{(k)},u_i^{(k)})\,(f^+(u_j^{(k)},\tilde{u}_i^{(k)}))^{\delta_{\hat{n},n}}}
$$
$$
\times \operatorname{tr}\Bigg[\prod_{(k,i)\succ(l,j)} R_{a_i^k a_j^l}^{(N,N)}(u_i^{(k)}-u_j^{(l)}) \prod_{(k,i)}\Bigg(S_{a_i^k}^{(N)}(u_i^{(k)}) \prod_{(k,i)\succ(l,j)}\widehat{R}_{a_i^k a_j^l}^{(N,N)}(\tilde{u}_i^{(k)}-u_j^{(l)})\Bigg)
$$
$$
\times (E_{n+1,n}^{(N)})^{\otimes m_n}\otimes\cdots\otimes(E_{21}^{(N)})^{\otimes m_1}\Bigg], \qquad (79)
$$

where $(k,i)\succ(l,j)$ means that $k>l$ or $k=l$ and $i>j$, and the products over tuples are defined in terms of the following rule

$$
\prod_{(k,i)} = \overset{\leftarrow}{\prod_{k<n}}\ \overset{\leftarrow}{\prod_{i<m_k}}\ .
$$

In other words, these products are ordered in the reversed lexicographical order. The trace is taken over all $a_i^k$ spaces, including $a_i^n$, which are associated with level-$n$ excitations. Note that $(k,i)$ is fixed in the third product inside the trace. Diagrammatically, the operator inside the trace is of the form

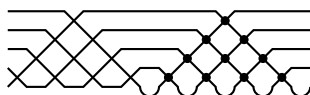

where $\times = R_{a_i^k a_j^l}(u_i^{(k)}-u_j^{(l)})$, $\pmb{\times} = \widehat{R}_{a_i^k a_j^l}(\tilde{u}_i^{(k)}-u_j^{(l)})$, and $\smile = S_{a_i^k}(u_i^{(k)})$.

*Example* 3.11. The "master" creation operators of low rank:

$$
\mathscr{B}_3(u_1^{(1)}) = s_{12}(u_1^{(1)}), \qquad \mathscr{B}_3(u_1^{(1)},u_2^{(1)}) = s_{12}(u_2^{(1)})s_{12}(u_1^{(1)}) + \frac{s_{13}(u_2^{(1)})s_{22}(u_1^{(1)})}{u_1^{(1)}-\tilde{u}_2^{(1)}},
$$

$$
\mathscr{B}_4(u_1^{(1)},u_1^{(2)}) = s_{23}(u_1^{(2)})s_{12}(u_1^{(1)}) + \frac{s_{24}(u_1^{(2)})s_{22}(u_1^{(1)})}{u_1^{(1)}-\tilde{u}_1^{(2)}} + \frac{(u_1^{(1)}-\tilde{u}_1^{(2)}+1)s_{13}(u_1^{(2)})s_{22}(u_1^{(1)})}{(u_1^{(1)}-u_1^{(2)})(u_1^{(1)}-\tilde{u}_1^{(2)})},
$$

$$
\mathscr{B}_5(u_1^{(1)},u_1^{(2)}) = s_{23}(u_1^{(2)})s_{12}(u_1^{(1)}) + \frac{s_{13}(u_1^{(2)})s_{22}(u_1^{(1)})}{u_1^{(1)}-u_1^{(2)}} + \frac{s_{25}(u_1^{(2)})s_{32}(u_1^{(1)})}{u_1^{(1)}-\tilde{u}_1^{(2)}}
$$
$$
+ \frac{s_{14}(u_1^{(2)})s_{32}(u_1^{(1)})}{(u_1^{(1)}-u_1^{(2)})(u_1^{(1)}-\tilde{u}_1^{(2)})}\ .
$$

**Proposition 3.12.** *The level-n Bethe vector* (66) *can be written as*

$$
\Psi(\boldsymbol{u}^{(1..n)}) = \mathscr{B}_N(\boldsymbol{u}^{(1\ldots n)})\cdot\eta\,. \qquad (80)
$$

*Proof.* First, notice that $R$-matrices $R_{a_i^k a_j^k}^{(N,N)}(u_i^{(k)}-u_j^{(k)})$ in (79) evaluate to $f^+(u_j^{(k)}-u_i^{(k)})$ under the trace. This cancels the first overall factor in (79). The second overall factor is the choice of normalisation in (49). Next, let $V_a^{(N)}$ and $V_b^{(N)}$ denote copies of $\mathbb{C}^N$. Then, for any $\zeta\in(L^{(n)})^0$ and $E_i^{(N)}\otimes E_j^{(N)}\in V_a^{(N)}\otimes V_b^{(N)}$ with $1\le i,j\le n$, we have

$$
Q_{ab}^{(N,N)}E_i^{(N)}\otimes E_j^{(N)} = 0\,,
$$

and

$$Q_{ab}^{(N,N)} S_a^{(N)}(v) \cdot E_i^{(N)} \otimes E_j^{(N)} \otimes \zeta = \sum_k Q_{ab}^{(N,N)} \cdot E_k^{(N)} \otimes E_j^{(N)} \otimes s_{ki}(v) \zeta = 0.$$

Thus $\widehat{R}_{a_i^k a_j^l}^{(N,N)}(\tilde{u}_i^{(k)} - u_j^{(l)})$ with $1 \leq k, l < n$ act as identity operators in (80). This gives an expression analogous (up to Yang-Baxter moves) to that in Proposition 4.7 of [12]. The $N = 2n$ case then follows from that proposition. The $N = 2n + 1$ case is proven analogously. □

## 4 Recurrence relations

### 4.1 Notation

Given any tuple $\boldsymbol{u}$ of complex parameters, let $(\boldsymbol{u}_{\mathrm{I}}, \boldsymbol{u}_{\mathrm{II}}) \vdash \boldsymbol{u}$ be a partition of this tuple and let $\boldsymbol{u}_{\mathrm{I,II}} := \boldsymbol{u}_{\mathrm{I}} \cup \boldsymbol{u}_{\mathrm{II}} = \boldsymbol{u}$. Assume that $1 \leq k < |\boldsymbol{u}|$ and set

$$\sum_{|\boldsymbol{u}_{\mathrm{II}}|=k} f(\boldsymbol{u}_{\mathrm{I}}) := \sum_{i_1 < i_2 < \cdots < i_k} f(\boldsymbol{u} \backslash (u_{i_1}, u_{i_2}, \ldots, u_{i_k})),$$

for any function or operator $f$. We will use a natural generalisation of this notation for any partition of $\boldsymbol{u}$. For instance, for $(\boldsymbol{u}_{\mathrm{I}}, \boldsymbol{u}_{\mathrm{II}}, \boldsymbol{u}_{\mathrm{III}}) \vdash \boldsymbol{u}$ we have $\boldsymbol{u}_{\mathrm{I,II}} = \boldsymbol{u}_{\mathrm{I}} \cup \boldsymbol{u}_{\mathrm{II}}$, $\boldsymbol{u}_{\mathrm{II,III}} = \boldsymbol{u}_{\mathrm{II}} \cup \boldsymbol{u}_{\mathrm{III}}$, etc., and e.g.

$$\sum_{|\boldsymbol{u}_{\mathrm{III}}|=1} \sum_{|\boldsymbol{u}_{\mathrm{II}}|=2} f(\boldsymbol{u}_{\mathrm{II}}) g(\boldsymbol{u}_{\mathrm{I}}) = \sum_j \sum_{\substack{i_1 < i_2 \\ i_1 \neq j, \, i_2 \neq j}} f((u_{i_1}, u_{i_2})) g(\boldsymbol{u} \backslash (u_{i_1}, u_{i_2}, u_j)).$$

We extend the notation above to partitions of tuples $\boldsymbol{u}^{(1\ldots n)}$ by allowing empty partitions. The empty partitions will be the ones that are missing from the expressions. For instance, an expression of the form

$$\sum_{\substack{|\boldsymbol{u}_{\mathrm{II}}^{(r)}|=k \\ i < r \leq n}} f(\boldsymbol{u}_{\mathrm{II}}^{(r)}) g(\boldsymbol{u}_{\mathrm{I}}^{(1\ldots n)}),$$

will mean that $\boldsymbol{u}_{\mathrm{II}}^{(1)} = \ldots = \boldsymbol{u}_{\mathrm{II}}^{(i)} = \emptyset$ so that $\boldsymbol{u}_{\mathrm{I}}^{(1\ldots n)} = (\boldsymbol{u}^{(1)}, \ldots, \boldsymbol{u}^{(i)}, \boldsymbol{u}_{\mathrm{I}}^{(i+1)}, \ldots, \boldsymbol{u}_{\mathrm{I}}^{(n)})$. We will also use the notation $|\boldsymbol{u}_{\mathrm{III}}^{(r)}| = 0$ meaning $\boldsymbol{u}_{\mathrm{III}}^{(r)} = \emptyset$.

The notation $|\boldsymbol{u}_{\mathrm{II,III}}^{(r)}| = (k, l)$ will mean that $|\boldsymbol{u}_{\mathrm{II}}^{(r)}| = k$ and $|\boldsymbol{u}_{\mathrm{III}}^{(r)}| = l$ and the notation $|\boldsymbol{u}_{\mathrm{II}}^{(r,s)}| = (k, l)$ will mean that $|\boldsymbol{u}_{\mathrm{II}}^{(r)}| = k$ and $|\boldsymbol{u}_{\mathrm{II}}^{(s)}| = l$ so that

$$\sum_{|\boldsymbol{u}_{\mathrm{II,III}}^{(r)}|=(k,l)} = \sum_{|\boldsymbol{u}_{\mathrm{III}}^{(r)}|=l} \sum_{|\boldsymbol{u}_{\mathrm{II}}^{(r)}|=k}, \qquad \text{and} \qquad \sum_{|\boldsymbol{u}_{\mathrm{II}}^{(r,s)}|=(k,l)} = \sum_{|\boldsymbol{u}_{\mathrm{II}}^{(s)}|=l} \sum_{|\boldsymbol{u}_{\mathrm{II}}^{(r)}|=k}.$$

A notation of the form $\boldsymbol{u}_{\mathrm{II,III}}^{(r,s)}$ will not be used.

### 4.2 Recurrence relations

We will combine the composite model method and the known $Y(\mathfrak{gl}_n)$-type recurrence relations to obtain recurrence relations for $Y^{\pm}(\mathfrak{g}_N)$-based Bethe vectors. The composite model method was introduced in [24]. For a pedagogical review, see [32]. Recurrence relations for $Y(\mathfrak{gl}_n)$-based Bethe vectors were obtained in [20]. We will need the following statement which follows directly from those in [20] recalled in Appendix A.3. Recall the notation (32) of rational functions.

**Proposition 4.1.** *Consider a $Y(\mathfrak{gl}_n)$-based Bethe vector $\Phi(\mathbf{v}^{(1...n-1)}|\mathbf{v}^{(n)})$ in the quantum space*

$$V_{a_{m_n}}^{(n)} \otimes \cdots \otimes V_{a_1}^{(n)} \otimes L(\boldsymbol{\lambda}), \tag{81}$$

*with $V_{a_i}^{(n)} \cong \mathbb{C}^n$, a finite-dimensional irreducible $Y(\mathfrak{gl}_n)$-module $L(\boldsymbol{\lambda})$, Bethe roots $\mathbf{v}^{(1...n-1)}$ and inhomogeneities $\mathbf{v}^{(n)}$ associated with spaces $V_{a_i}^{(n)}$. Set*

$$\Lambda_k(z;\mathbf{v}^{(1...n-1)}) := f^-(z,\mathbf{v}^{(k-1)})f^+(z,\mathbf{v}^{(k)})\lambda_k(z). \tag{82}$$

*An expansion of $\Phi(\mathbf{v}^{(1...n-1)}|\mathbf{v}^{(n)})$ in the space $V_{a_{m_n}}^{(n)}$ is given by*

$$\Phi(\mathbf{v}^{(1...n-1)}|\mathbf{v}^{(n)}) = \sum_{1\le i\le n} \sum_{\substack{|\mathbf{v}_{\mathrm{II}}^{(r)}|=1 \\ i\le r<n}} \prod_{i<k\le n} \frac{\Lambda_k(\mathbf{v}_{\mathrm{II}}^{(k-1)};\mathbf{v}_{\mathrm{I}}^{(1...n)})}{\mathbf{v}_{\mathrm{II}}^{(k-1)}-\mathbf{v}_{\mathrm{II}}^{(k)}} E_{\bar{\imath}}^{(n)} \otimes \Phi(\mathbf{v}_{\mathrm{I}}^{(1...n-1)}|\mathbf{v}_{\mathrm{I}}^{(n)}), \tag{83}$$

*where $\mathbf{v}_{\mathrm{II}}^{(n)} = v_{m_n}^{(n)}$ and $\mathbf{v}_{\mathrm{II}}^{(r)} = \emptyset$ for all $1\le r<i$ so that $\mathbf{v}_{\mathrm{I}}^{(1...n)} = (\mathbf{v}^{(1)},\ldots,\mathbf{v}^{(i-1)},\mathbf{v}_{\mathrm{I}}^{(i)},\ldots,\mathbf{v}_{\mathrm{I}}^{(n)})$.*

**Corollary 4.2.** *An expansion of Bethe vector $\Phi(\mathbf{v}^{(1...n-1)}|\mathbf{v}^{(n)})$ in the space $V_{a_{m_n}}^{(n)}\otimes V_{a_{m_n-1}}^{(n)}$ is given by*

$$\sum_{1\le i\le n} \sum_{\substack{|\mathbf{v}_{\mathrm{II,III}}^{(r)}|=(2,0) \\ i\le r<n}} \prod_{i<k\le n} K(\mathbf{v}_{\mathrm{II}}^{(k-1)}|\mathbf{v}_{\mathrm{II,III}}^{(k)}) \Lambda_k(\mathbf{v}_{\mathrm{II}}^{(k-1)};\mathbf{v}_{\mathrm{I}}^{(1...n)}) E_{\bar{\imath}}^{(n)} \otimes E_{\bar{\imath}}^{(n)} \otimes \Phi(\mathbf{v}_{\mathrm{I}}^{(1...n-1)}|\mathbf{v}_{\mathrm{I}}^{(n)})$$

$$+ \sum_{1\le i<j\le n} \sum_{\substack{|\mathbf{v}_{\mathrm{III}}^{(r)}|=1 \\ i\le r<n}} \sum_{\substack{|\mathbf{v}_{\mathrm{II}}^{(s)}|=1 \\ j\le s<n}} \prod_{i<k<j} \frac{\Lambda_k(\mathbf{v}_{\mathrm{III}}^{(k-1)};\mathbf{v}_{\mathrm{I}}^{(1...n-1)})}{\mathbf{v}_{\mathrm{III}}^{(k-1)}-\mathbf{v}_{\mathrm{III}}^{(k)}} \cdot \Lambda_j(\mathbf{v}_{\mathrm{III}}^{(j-1)};\mathbf{v}_{\mathrm{I}}^{(1...n-1)})$$

$$\times \prod_{j<k\le n} \frac{\Lambda_k(\mathbf{v}_{\mathrm{II}}^{(k-1)};\mathbf{v}_{\mathrm{I}}^{(1...n-1)}) \Lambda_k(\mathbf{v}_{\mathrm{III}}^{(k-1)};\mathbf{v}_{\mathrm{I,II}}^{(1...n-1)})}{(\mathbf{v}_{\mathrm{II}}^{(k-1)}-\mathbf{v}_{\mathrm{II}}^{(k)})(\mathbf{v}_{\mathrm{III}}^{(k-1)}-\mathbf{v}_{\mathrm{III}}^{(k)})}$$

$$\times \left( \frac{f^+(\mathbf{u}_{\mathrm{III}}^{(j-1)},\mathbf{v}_{\mathrm{II}}^{(j)})}{\mathbf{v}_{\mathrm{III}}^{(j-1)}-\mathbf{v}_{\mathrm{III}}^{(j)}} E_{\bar{\imath}}^{(n)} \otimes E_{\bar{\jmath}}^{(n)} + \frac{1}{\mathbf{v}_{\mathrm{III}}^{(j-1)}-\mathbf{v}_{\mathrm{II}}^{(j)}} E_{\bar{\jmath}}^{(n)} \otimes E_{\bar{\imath}}^{(n)} \right) \otimes \Phi(\mathbf{v}_{\mathrm{I}}^{(1...n-1)}|\mathbf{v}_{\mathrm{I}}^{(n)}), \quad (84)$$

*where $\mathbf{v}_{\mathrm{III}}^{(n)} = v_{m_n}^{(n)}$, $\mathbf{v}_{\mathrm{II}}^{(n)} = v_{m_n-1}^{(n)}$ and $\mathbf{v}_{\mathrm{III}}^{(r)} = \mathbf{v}_{\mathrm{II}}^{(r)} = \emptyset$ for all $1\le r<i$ in the first sum and $\mathbf{v}_{\mathrm{III}}^{(r)} = \mathbf{v}_{\mathrm{II}}^{(s)} = \emptyset$ for all $1\le r<i$ and $1\le s<j$ in the second sum, and*

$$K(\mathbf{u}|\mathbf{v}) := \frac{\prod_{i,j}(u_i-v_j+1)}{\prod_{i<j}(u_i-u_j)(v_j-v_i)} \det_{i,j}\left(\frac{1}{(u_i-v_j)(u_i-v_j+1)}\right), \tag{85}$$

*is the domain wall boundary partition function.*

*Proof.* Applying (83) to $\Phi(\mathbf{v}^{(1...n-1)}|\mathbf{v}^{(n)})$ twice gives

$$\sum_{1\le i,j\le n} \sum_{\substack{|\mathbf{v}_{\mathrm{III}}^{(r)}|=1 \\ i\le r<n}} \sum_{\substack{|\mathbf{v}_{\mathrm{II}}^{(s)}|=1 \\ j\le s<n}} \prod_{i<k\le n} \frac{\Lambda_k(\mathbf{v}_{\mathrm{III}}^{(k-1)};\mathbf{v}_{\mathrm{I,II}}^{(1...n)})}{\mathbf{v}_{\mathrm{III}}^{(k-1)}-\mathbf{v}_{\mathrm{III}}^{(k)}} \prod_{j<l\le n} \frac{\Lambda_l(\mathbf{v}_{\mathrm{II}}^{(l-1)};\mathbf{v}_{\mathrm{I}}^{(1...n)})}{\mathbf{v}_{\mathrm{II}}^{(l-1)}-\mathbf{v}_{\mathrm{II}}^{(l)}} \Phi_{\bar{\imath}\bar{\jmath}}, \tag{86}$$

*where $\mathbf{v}_{\mathrm{III}}^{(r)} = \mathbf{v}_{\mathrm{II}}^{(s)} = \emptyset$ for all $1\le r<i$ and $1\le s<j$, and $\Phi_{ij} := E_i^{(n)}\otimes E_i^{(n)}\otimes \Phi(\mathbf{v}_{\mathrm{I}}^{(1...n-1)}|\mathbf{v}_{\mathrm{I}}^{(n)})$.*

*Cases $i=j$.* Notice that

$$\Lambda k(\mathbf{v}_{\mathrm{III}}^{(k-1)};\mathbf{v}_{\mathrm{I,II}}^{(1...n)}) = f^-(\mathbf{v}_{\mathrm{III}}^{(k-1)},\mathbf{v}_{\mathrm{II}}^{(k-1)})f^+(\mathbf{v}_{\mathrm{III}}^{(k-1)},\mathbf{v}_{\mathrm{II}}^{(k)})\Lambda_k(\mathbf{v}_{\mathrm{III}}^{(k-1)};\mathbf{v}_{\mathrm{I}}^{(1...n)}),$$

and

$$\frac{f^-(\boldsymbol{v}_{\mathrm{III}}^{(k-1)}, \boldsymbol{v}_{\mathrm{II}}^{(k-1)}) f^+(\boldsymbol{v}_{\mathrm{III}}^{(k-1)}, \boldsymbol{v}_{\mathrm{II}}^{(k)})}{(\boldsymbol{v}_{\mathrm{III}}^{(k-1)} - \boldsymbol{v}_{\mathrm{III}}^{(k)})(\boldsymbol{v}_{\mathrm{II}}^{(k-1)} - \boldsymbol{v}_{\mathrm{II}}^{(k)})} + \frac{f^-(\boldsymbol{v}_{\mathrm{II}}^{(k-1)}, \boldsymbol{v}_{\mathrm{III}}^{(k-1)}) f^+(\boldsymbol{v}_{\mathrm{II}}^{(k-1)}, \boldsymbol{v}_{\mathrm{II}}^{(k)})}{(\boldsymbol{v}_{\mathrm{II}}^{(k-1)} - \boldsymbol{v}_{\mathrm{II}}^{(k)})(\boldsymbol{v}_{\mathrm{III}}^{(k-1)} - \boldsymbol{v}_{\mathrm{II}}^{(k)})} = K(\boldsymbol{v}_{\mathrm{II},\mathrm{II}}^{(k-1)} | \boldsymbol{v}_{\mathrm{II},\mathrm{III}}^{(k)}).$$

These identities allow us to rewrite the $i = j$ cases of (86) as

$$\sum_{\substack{1 \le i \le n}} \sum_{\substack{|\boldsymbol{v}_{\mathrm{II},\mathrm{III}}^{(r)}|=(1,1) \\ i \le r < n}} \prod_{i < k \le n} K(\boldsymbol{v}_{\mathrm{II},\mathrm{III}}^{(k-1)} | \boldsymbol{v}_{\mathrm{II},\mathrm{III}}^{(k)}) \Lambda_k(\boldsymbol{v}_{\mathrm{II},\mathrm{III}}^{(k-1)}; \boldsymbol{v}_{\mathrm{I}}^{(1...n)}) \Phi_{\bar{i}\bar{i}},$$

giving the first sum in (84).

*Cases $i < j$.* Since $\boldsymbol{v}_{\mathrm{II}}^{(s)} = \emptyset$ for $s < j$ in (86) we have

$$\Lambda_k(\boldsymbol{v}_{\mathrm{III}}^{(k-1)}; \boldsymbol{v}_{\mathrm{I},\mathrm{II}}^{(1...n)}) = \Lambda_k(\boldsymbol{v}_{\mathrm{III}}^{(k-1)}; \boldsymbol{v}_{\mathrm{I}}^{(1...n)}), \quad \text{for} \quad k < j,$$

and

$$\Lambda_j(\boldsymbol{v}_{\mathrm{III}}^{(j-1)}; \boldsymbol{v}_{\mathrm{I},\mathrm{II}}^{(1...n)}) = f^+(\boldsymbol{v}_{\mathrm{III}}^{(j-1)}, \boldsymbol{v}_{\mathrm{II}}^{(j)}) \Lambda_j(\boldsymbol{v}_{\mathrm{III}}^{(j-1)}; \boldsymbol{v}_{\mathrm{I}}^{(1...n)}),$$

allowing us to rewrite the $i < j$ cases as

$$\sum_{\substack{1 \le i < j \le n}} \sum_{\substack{|\boldsymbol{v}_{\mathrm{III}}^{(r)}|=1 \\ i \le r < n}} \sum_{\substack{|\boldsymbol{v}_{\mathrm{II}}^{(s)}|=1 \\ j \le s < n}} \prod_{i < k < j} \frac{\Lambda_k(\boldsymbol{v}_{\mathrm{III}}^{(k-1)}; \boldsymbol{v}_{\mathrm{I}}^{(1...n)})}{\boldsymbol{v}_{\mathrm{III}}^{(k-1)} - \boldsymbol{v}_{\mathrm{III}}^{(k)}} \cdot \Lambda_j(\boldsymbol{v}_{\mathrm{III}}^{(j-1)}; \boldsymbol{v}_{\mathrm{I}}^{(1...n)})$$

$$\times \prod_{j < k \le n} \frac{\Lambda_k(\boldsymbol{v}_{\mathrm{II}}^{(k-1)}; \boldsymbol{v}_{\mathrm{I}}^{(1...n)}) \Lambda_k(\boldsymbol{v}_{\mathrm{III}}^{(k-1)}; \boldsymbol{v}_{\mathrm{I},\mathrm{II}}^{(1...n)})}{(\boldsymbol{v}_{\mathrm{II}}^{(k-1)} - \boldsymbol{v}_{\mathrm{II}}^{(k)})(\boldsymbol{v}_{\mathrm{III}}^{(k-1)} - \boldsymbol{v}_{\mathrm{III}}^{(k)})} \cdot \frac{f^+(\boldsymbol{v}_{\mathrm{III}}^{(j-1)}, \boldsymbol{v}_{\mathrm{II}}^{(j)})}{\boldsymbol{v}_{\mathrm{III}}^{(j-1)} - \boldsymbol{v}_{\mathrm{III}}^{(j)}} \Phi_{\bar{i}j}. \tag{87}$$

*Cases $i > j$.* Interchanging indices $i$ and $j$ in (86) gives

$$\sum_{\substack{1 \le i < j \le n}} \sum_{\substack{|\boldsymbol{v}_{\mathrm{III}}^{(s)}|=1 \\ j \le s < n}} \sum_{\substack{|\boldsymbol{v}_{\mathrm{II}}^{(r)}|=1 \\ i \le r < n}} \prod_{i < k < j} \frac{\Lambda_k(\boldsymbol{v}_{\mathrm{II}}^{(k-1)}; \boldsymbol{v}_{\mathrm{I}}^{(1...n)})}{\boldsymbol{v}_{\mathrm{II}}^{(k-1)} - \boldsymbol{v}_{\mathrm{II}}^{(k)}} \cdot \Lambda_j(\boldsymbol{v}_{\mathrm{II}}^{(j-1)}; \boldsymbol{v}_{\mathrm{I}}^{(1...n)})$$

$$\times \prod_{j < k \le n} \frac{\Lambda_k(\boldsymbol{v}_{\mathrm{II}}^{(k-1)}; \boldsymbol{v}_{\mathrm{I}}^{(1...n)}) \Lambda_k(\boldsymbol{v}_{\mathrm{III}}^{(k-1)}; \boldsymbol{v}_{\mathrm{I},\mathrm{II}}^{(1...n)})}{(\boldsymbol{v}_{\mathrm{II}}^{(k-1)} - \boldsymbol{v}_{\mathrm{II}}^{(k)})(\boldsymbol{v}_{\mathrm{III}}^{(k-1)} - \boldsymbol{v}_{\mathrm{III}}^{(k)})} \cdot \frac{1}{\boldsymbol{v}_{\mathrm{II}}^{(j-1)} - \boldsymbol{v}_{\mathrm{II}}^{(j)}} \Phi_{\bar{j}\bar{i}}. \tag{88}$$

Since $i < j$ we can rename $\boldsymbol{v}_{\mathrm{II}}^{(r)}$ by $\boldsymbol{v}_{\mathrm{III}}^{(r)}$ for $i \le r < j$ and combine the result with (87). This gives the second sum in (84). $\qquad\square$

*Example* 4.3. When $N = 3$, expansion (84) of $\Phi(\boldsymbol{v}^{(1,2)} | \boldsymbol{v}^{(3)})$ is

$$\Phi_{11} + \sum_{|\boldsymbol{v}_{\mathrm{II}}^{(2)}|=2} K(\boldsymbol{v}_{\mathrm{II}}^{(2)} | \boldsymbol{v}_{\mathrm{II},\mathrm{III}}^{(3)}) \Lambda_3(\boldsymbol{v}_{\mathrm{II}}^{(2)}; \boldsymbol{v}_{\mathrm{I}}^{(1,2,3)}) \Phi_{22}$$

$$+ \sum_{|\boldsymbol{v}_{\mathrm{II}}^{(1,2)}|=(2,2)} K(\boldsymbol{v}_{\mathrm{II}}^{(1)} | \boldsymbol{v}_{\mathrm{II}}^{(2)}) K(\boldsymbol{v}_{\mathrm{II}}^{(2)} | \boldsymbol{v}_{\mathrm{II},\mathrm{III}}^{(3)}) \Lambda_2(\boldsymbol{v}_{\mathrm{II}}^{(1)}; \boldsymbol{v}_{\mathrm{I}}^{(1,2,3)}) \Lambda_3(\boldsymbol{v}_{\mathrm{II}}^{(2)}; \boldsymbol{v}_{\mathrm{I}}^{(1,2,3)}) \Phi_{33}$$

$$+ \sum_{|\boldsymbol{v}_{\mathrm{III}}^{(2)}|=1} \Lambda_3(\boldsymbol{v}_{\mathrm{III}}^{(2)}; \boldsymbol{v}_{\mathrm{I}}^{(1,2,3)}) \left( \frac{f^+(\boldsymbol{v}_{\mathrm{III}}^{(2)}, \boldsymbol{v}_{\mathrm{II}}^{(3)})}{\boldsymbol{v}_{\mathrm{III}}^{(2)} - \boldsymbol{v}_{\mathrm{III}}^{(3)}} \Phi_{21} + \frac{1}{\boldsymbol{v}_{\mathrm{III}}^{(2)} - \boldsymbol{v}_{\mathrm{II}}^{(3)}} \Phi_{12} \right)$$

$$+ \sum_{|\boldsymbol{v}_{\mathrm{III}}^{(1,2)}|=(1,1)} \frac{\Lambda_2(\boldsymbol{v}_{\mathrm{III}}^{(1)}; \boldsymbol{v}_{\mathrm{I}}^{(1,2,3)})}{\boldsymbol{v}_{\mathrm{III}}^{(1)} - \boldsymbol{v}_{\mathrm{III}}^{(2)}} \Lambda_3(\boldsymbol{v}_{\mathrm{III}}^{(2)}; \boldsymbol{v}_{\mathrm{I}}^{(1,2,3)}) \left( \frac{f^+(\boldsymbol{v}_{\mathrm{III}}^{(2)}, \boldsymbol{v}_{\mathrm{II}}^{(3)})}{\boldsymbol{v}_{\mathrm{III}}^{(2)} - \boldsymbol{v}_{\mathrm{III}}^{(3)}} \Phi_{31} + \frac{1}{\boldsymbol{v}_{\mathrm{III}}^{(2)} - \boldsymbol{v}_{\mathrm{II}}^{(3)}} \Phi_{13} \right)$$

$$+ \sum_{|\boldsymbol{v}_{\mathrm{III}}^{(1,2)}|=(1,1)} \sum_{|\boldsymbol{v}_{\mathrm{II}}^{(2)}|=1} \Lambda_2(\boldsymbol{v}_{\mathrm{III}}^{(1)}; \boldsymbol{v}_{\mathrm{I}}^{(1,2,3)}) \frac{\Lambda_3(\boldsymbol{v}_{\mathrm{II}}^{(2)}; \boldsymbol{v}_{\mathrm{I}}^{(1,2,3)}) \Lambda_3(\boldsymbol{v}_{\mathrm{III}}^{(2)}; \boldsymbol{v}_{\mathrm{I,II}}^{(1,2,3)})}{(\boldsymbol{v}_{\mathrm{II}}^{(2)} - \boldsymbol{v}_{\mathrm{II}}^{(3)})(\boldsymbol{v}_{\mathrm{III}}^{(2)} - \boldsymbol{v}_{\mathrm{III}}^{(3)})}$$

$$\times \left( \frac{f^+(\boldsymbol{v}_{\mathrm{III}}^{(1)}, \boldsymbol{v}_{\mathrm{II}}^{(2)})}{\boldsymbol{v}_{\mathrm{III}}^{(1)} - \boldsymbol{v}_{\mathrm{III}}^{(2)}} \Phi_{32} + \frac{1}{\boldsymbol{v}_{\mathrm{III}}^{(1)} - \boldsymbol{v}_{\mathrm{II}}^{(2)}} \Phi_{23} \right),$$

where $\boldsymbol{v}_{\mathrm{III}}^{(3)} = v_{m_3}^{(3)}$, $\boldsymbol{v}_{\mathrm{II}}^{(3)} = v_{m_3-1}^{(3)}$, and $\boldsymbol{v}_{\mathrm{III}}^{(2)} = \boldsymbol{v}_{\mathrm{III}}^{(1)} = \boldsymbol{v}_{\mathrm{II}}^{(1)} = \emptyset$ in the first sum, $\boldsymbol{v}_{\mathrm{III}}^{(2)} = \boldsymbol{v}_{\mathrm{III}}^{(1)} = \emptyset$ in the second sum, $\boldsymbol{v}_{\mathrm{III}}^{(1)} = \boldsymbol{v}_{\mathrm{II}}^{(2)} = \boldsymbol{v}_{\mathrm{II}}^{(1)} = \emptyset$ in the third sum, $\boldsymbol{v}_{\mathrm{II}}^{(2)} = \boldsymbol{v}_{\mathrm{II}}^{(1)} = \emptyset$ in the fourth sum and $\boldsymbol{v}_{\mathrm{II}}^{(1)} = \emptyset$ in the last sum, and $\Phi_{ij} = E_i^{(3)} \otimes E_j^{(3)} \otimes \Phi(\boldsymbol{v}_{\mathrm{I}}^{(1,2)} | \boldsymbol{v}_{\mathrm{I}}^{(3)})$.

We are ready to state the main results of this section, recurrence relations for twisted Yangian based Bethe vectors. The even case follows almost immediately from Corollary 4.2. The odd case will require additional steps which are due to the $E_{\hat{1}}^{(\hat{n})} = E_2^{(n+1)}$ factors in the reference vector $\eta^m$.

**Proposition 4.4.** $Y^\pm(\mathfrak{gl}_{2n})$-based Bethe vectors satisfy the recurrence relation

$$\Psi(\boldsymbol{u}^{(1...n)}) = \sum_{\substack{1 \le i \le n \\ |\boldsymbol{u}_{\mathrm{II,III}}^{(r)}|=(2,0) \\ i \le r < n}} \sum_{i<k\le n} \prod K(\boldsymbol{u}_{\mathrm{II}}^{(k-1)} | \boldsymbol{u}_{\mathrm{II,III}}^{(k)}) \Gamma_k(\boldsymbol{u}_{\mathrm{II}}^{(k-1)}; \boldsymbol{u}_{\mathrm{I}}^{(1...n)}) s_{i,2n-i+1}(\boldsymbol{u}_{\mathrm{III}}^{(n)}) \Psi(\boldsymbol{u}_{\mathrm{I}}^{(1...n)})$$

$$+ \sum_{\substack{1 \le i < j \le n \\ |\boldsymbol{u}_{\mathrm{III}}^{(r)}|=1 \\ i \le r < n}} \sum_{\substack{|\boldsymbol{u}_{\mathrm{II}}^{(s)}|=1 \\ j \le s < n}} \sum_{i<k<j} \prod \frac{\Gamma_k(\boldsymbol{u}_{\mathrm{III}}^{(k-1)}; \boldsymbol{u}_{\mathrm{I}}^{(1...n)})}{\boldsymbol{u}_{\mathrm{III}}^{(k-1)} - \boldsymbol{u}_{\mathrm{III}}^{(k)}} \cdot \Gamma_j(\boldsymbol{u}_{\mathrm{III}}^{(j-1)}; \boldsymbol{u}_{\mathrm{I}}^{(1...n)})$$

$$\times \prod_{j<k\le n} \frac{\Gamma_k(\boldsymbol{u}_{\mathrm{II}}^{(k-1)}; \boldsymbol{u}_{\mathrm{I}}^{(1...n)}) \Gamma_k(\boldsymbol{u}_{\mathrm{III}}^{(k-1)}; \boldsymbol{u}_{\mathrm{I,II}}^{(1...n)})}{(\boldsymbol{u}_{\mathrm{II}}^{(k-1)} - \boldsymbol{u}_{\mathrm{II}}^{(k)})(\boldsymbol{u}_{\mathrm{III}}^{(k-1)} - \boldsymbol{u}_{\mathrm{III}}^{(k)})}$$

$$\times \left( \frac{f^+(\boldsymbol{u}_{\mathrm{III}}^{(j-1)}, \boldsymbol{u}_{\mathrm{II}}^{(j)})}{\boldsymbol{u}_{\mathrm{III}}^{(j-1)} - \boldsymbol{u}_{\mathrm{III}}^{(j)}} s_{i,2n-j+1}(\boldsymbol{u}_{\mathrm{III}}^{(n)}) + \frac{1}{\boldsymbol{u}_{\mathrm{III}}^{(j-1)} - \boldsymbol{u}_{\mathrm{II}}^{(j)}} s_{j,2n-i+1}(\boldsymbol{u}_{\mathrm{III}}^{(n)}) \right) \Psi(\boldsymbol{u}_{\mathrm{I}}^{(1...n)}),$$

$$(89)$$

where $\boldsymbol{u}_{\mathrm{III}}^{(n)} = u_j^{(n)}$, $\boldsymbol{u}_{\mathrm{II}}^{(n)} = \tilde{u}_j^{(n)}$ and $\boldsymbol{u}_{\mathrm{I}}^{(n)} = \boldsymbol{u}^{(n)} \backslash u_j^{(n)}$ for any $1 \le j \le m_n$, and $\boldsymbol{u}_{\mathrm{III}}^{(r)} = \boldsymbol{u}_{\mathrm{II}}^{(r)} = \emptyset$ for all $1 \le r < i$ in the first sum, $\boldsymbol{u}_{\mathrm{III}}^{(r)} = \boldsymbol{u}_{\mathrm{II}}^{(s)} = \emptyset$ for all $1 \le r < i$ and $1 \le s < j$ in the second sum, and $\Gamma_k(\boldsymbol{u}_{\mathrm{III}}^{(k-1)}; \boldsymbol{u}_{\mathrm{I,II}}^{(1...n)})$ when $k = n$ denotes $f^+(\boldsymbol{u}_{\mathrm{III}}^{(n-1)}, \boldsymbol{u}_{\mathrm{II}}^{(n)}) \Gamma_n(\boldsymbol{u}_{\mathrm{III}}^{(n-1)}; \boldsymbol{u}_{\mathrm{I}}^{(1...n)})$.

*Example* 4.5. When $n = 2$, the recurrence relation (89) gives

$$\Psi(\boldsymbol{u}^{(1,2)}) = s_{23}(\boldsymbol{u}_{\mathrm{III}}^{(2)}) \Psi(\boldsymbol{u}_{\mathrm{I}}^{(1,2)}) + \sum_{|\boldsymbol{u}_{\mathrm{II}}^{(1)}|=2} K(\boldsymbol{u}_{\mathrm{II}}^{(1)} | \boldsymbol{u}_{\mathrm{II,III}}^{(2)}) \Gamma_2(\boldsymbol{u}_{\mathrm{II}}^{(1)}; \boldsymbol{u}_{\mathrm{I}}^{(1,2)}) s_{14}(\boldsymbol{u}_{\mathrm{III}}^{(2)}) \Psi(\boldsymbol{u}_{\mathrm{I}}^{(1,2)})$$

$$+ \sum_{|\boldsymbol{u}_{\mathrm{III}}^{(1)}|=1} \Gamma_2(\boldsymbol{u}_{\mathrm{III}}^{(1)}; \boldsymbol{u}_{\mathrm{I}}^{(1,2)}) \left( \frac{f^+(\boldsymbol{u}_{\mathrm{III}}^{(1)}, \boldsymbol{u}_{\mathrm{II}}^{(2)})}{\boldsymbol{u}_{\mathrm{III}}^{(1)} - \boldsymbol{u}_{\mathrm{III}}^{(2)}} s_{13}(\boldsymbol{u}_{\mathrm{III}}^{(2)}) + \frac{1}{\boldsymbol{u}_{\mathrm{III}}^{(1)} - \boldsymbol{u}_{\mathrm{II}}^{(2)}} s_{24}(\boldsymbol{u}_{\mathrm{III}}^{(2)}) \right) \Psi(\boldsymbol{u}_{\mathrm{I}}^{(1,2)}), \quad (90)$$

where $\boldsymbol{u}_{\mathrm{III}}^{(2)} = u_j^{(2)}$, $\boldsymbol{u}_{\mathrm{II}}^{(2)} = \tilde{u}_j^{(2)}$ and $\boldsymbol{u}_{\mathrm{I}}^{(2)} = \boldsymbol{u}^{(2)} \backslash u_j^{(2)}$ for any $1 \le j \le m_2$, and $\boldsymbol{u}_{\mathrm{III}}^{(1)} = \emptyset$ in the first sum and $\boldsymbol{u}_{\mathrm{II}}^{(1)} = \emptyset$ in the second sum.

*Proof of Proposition 4.4.* By Lemma 3.7, it is sufficient to consider the $j = m_n$ case. Recall (51), (66) and consider a level-$(n-1)$ vector

$$\mathscr{R}^{(\hat{n})}(u_{m_n}^{(n)}; \boldsymbol{u}^{(n)} \backslash u_{m_n}^{(n)}) \Psi(\boldsymbol{u}^{(1...n-1)} | \boldsymbol{u}^{(n)}). \quad (91)$$

With the help of Yang-Baxter equation we can move operator $\mathcal{R}^{(\hat{n})}(u_{m_n}^{(n)}; \boldsymbol{u}^{(n)}\backslash u_{m_n}^{(n)})$ all way to the reference vector $\eta^{\boldsymbol{m}}$. As a result of this, the level-$(n-1)$ nested monodromy matrix (40) factorises as

$$\widehat{R}_{\dot{a}_{m_n}a}^{(\hat{n},\hat{n})}(u_{m_n}^{(n)}-v)\,\widehat{R}_{\ddot{a}_{m_n}a}^{(\hat{n},\hat{n})}(\tilde{u}_{m_n}^{(n)}-v)\,T_a^{(\hat{n})}(v;\boldsymbol{u}^{(n)}\backslash u_{m_n}^{(n)})\,. \tag{92}$$

Since $\mathcal{R}^{(\hat{n})}(u_{m_n}^{(n)}; \boldsymbol{u}^{(n)}\backslash u_{m_n}^{(n)})\cdot\eta^{\boldsymbol{m}}=\eta^{\boldsymbol{m}}$ when $\hat{n}=n$, we may view vector (91) as a $Y(\mathfrak{gl}_n)$-based Bethe vector with monodromy matrix (92) and apply expansion (84) in the space $V_{\dot{a}_{m_n}}^{(n)}\otimes V_{\ddot{a}_{m_n}}^{(n)}$. Recall (50), (71), (72) and act with $\lfloor_{\dot{a}_{m_n}\ddot{a}_{m_n}}^{(n)}(u_{m_n}^{(n)})\,\mathcal{B}^{(n)}(\boldsymbol{u}^{(n)}\backslash u_{m_n}^{(n)})$ on the resulting expression. This immediately gives the wanted result. $\qquad\square$

**Proposition 4.6.** $Y^+(\mathfrak{gl}_{2n+1})$-*based Bethe vectors satisfy the recurrence relation*

$$\Psi(\boldsymbol{u}^{(1\dots n)})=\sum_{\substack{1\le i\le n}}\sum_{\substack{|\boldsymbol{u}_{\mathrm{III}}^{(r)}|=1\\i\le r<n}}\prod_{i<k\le n}\frac{\Gamma_k(\boldsymbol{u}_{\mathrm{III}}^{(k-1)};\boldsymbol{u}_{\mathrm{I}}^{(1\dots n)})}{\boldsymbol{u}_{\mathrm{III}}^{(k-1)}-\boldsymbol{u}_{\mathrm{III}}^{(k)}}\,s_{i,\hat{n}}(\boldsymbol{u}_{\mathrm{III}}^{(n)})\,\Psi(\boldsymbol{u}_{\mathrm{I}}^{(1\dots n)})$$

$$+\sum_{\substack{1\le i<n}}\sum_{\substack{|\boldsymbol{u}_{\mathrm{II}}^{(r)}|=1\\i\le r\le n}}\prod_{i<k\le n}\frac{\Gamma_k(\boldsymbol{u}_{\mathrm{II}}^{(k-1)};\boldsymbol{u}_{\mathrm{I}}^{(1\dots n)})}{\boldsymbol{u}_{\mathrm{II}}^{(k-1)}-\boldsymbol{u}_{\mathrm{II}}^{(k)}}\cdot\frac{\Gamma_{\hat{n}}(\boldsymbol{u}_{\mathrm{II}}^{(n)};\boldsymbol{u}_{\mathrm{I}}^{(1\dots n)})}{\boldsymbol{u}_{\mathrm{II}}^{(n)}-\tilde{\boldsymbol{u}}_{\mathrm{III}}^{(n)}}$$

$$\times\left(\frac{\boldsymbol{u}_{\mathrm{II}}^{(n-1)}-\boldsymbol{u}_{\mathrm{II}}^{(n)}+1}{\boldsymbol{u}_{\mathrm{II}}^{(n-1)}-\boldsymbol{u}_{\mathrm{III}}^{(n)}}\,s_{i,\hat{n}+1}(\boldsymbol{u}_{\mathrm{III}}^{(n)})+s_{n,\hat{n}+i+1}(\boldsymbol{u}_{\mathrm{III}}^{(n)})\right)\Psi(\boldsymbol{u}_{\mathrm{I}}^{(1\dots n)})$$

$$+\sum_{\substack{1\le i\le n}}\sum_{\substack{|\boldsymbol{u}_{\mathrm{II,III}}^{(r)}|=(2,0)\\i\le r<n}}\sum_{|\boldsymbol{u}_{\mathrm{II}}^{(n)}|=1}\prod_{i<k\le n}\Gamma_k(\boldsymbol{u}_{\mathrm{II}}^{(k-1)};\boldsymbol{u}_{\mathrm{I}}^{(1\dots n)})\,K(\boldsymbol{u}_{\mathrm{II}}^{(k-1)}|\boldsymbol{u}_{\mathrm{II,III}}^{(k)})$$

$$\times\frac{\Gamma_{\hat{n}}(\boldsymbol{u}_{\mathrm{II}}^{(n)};\boldsymbol{u}_{\mathrm{I}}^{(1\dots n)})}{\boldsymbol{u}_{\mathrm{II}}^{(n)}-\tilde{\boldsymbol{u}}_{\mathrm{III}}^{(n)}}\,s_{i,2\hat{n}-i}(\boldsymbol{u}_{\mathrm{III}}^{(n)})\,\Psi(\boldsymbol{u}_{\mathrm{I}}^{(1\dots n)})$$

$$+\sum_{\substack{1\le i<j<n}}\sum_{\substack{|\boldsymbol{u}_{\mathrm{III}}^{(r)}|=1\\i\le r<n}}\sum_{\substack{|\boldsymbol{u}_{\mathrm{II}}^{(s)}|=1\\j\le s\le n}}\prod_{i<k<j}\frac{\Gamma_k(\boldsymbol{u}_{\mathrm{III}}^{(k-1)};\boldsymbol{u}_{\mathrm{I}}^{(1\dots n)})}{\boldsymbol{u}_{\mathrm{III}}^{(k-1)}-\boldsymbol{u}_{\mathrm{III}}^{(k)}}\cdot\Gamma_j(\boldsymbol{u}_{\mathrm{III}}^{(j-1)};\boldsymbol{u}_{\mathrm{I}}^{(1\dots n)})$$

$$\times\prod_{j<k<n}\frac{\Gamma_k(\boldsymbol{u}_{\mathrm{II}}^{(k-1)};\boldsymbol{u}_{\mathrm{I}}^{(1\dots n)})\,\Gamma_k(\boldsymbol{u}_{\mathrm{III}}^{(k-1)};\boldsymbol{u}_{\mathrm{I,II}}^{(1\dots n)})}{(\boldsymbol{u}_{\mathrm{II}}^{(k-1)}-\boldsymbol{u}_{\mathrm{II}}^{(k)})(\boldsymbol{u}_{\mathrm{III}}^{(k-1)}-\boldsymbol{u}_{\mathrm{III}}^{(k)})}\cdot\Gamma_n(\boldsymbol{u}_{\mathrm{II,III}}^{(k-1)};\boldsymbol{u}_{\mathrm{I}}^{(1\dots n)})\,\Gamma_{\hat{n}}(\boldsymbol{u}_{\mathrm{II}}^{(n)};\boldsymbol{u}_{\mathrm{I}}^{(1\dots n)})$$

$$\times\left[\left(\left(\beta_0+\frac{\beta_2}{2\gamma}\right)\frac{f^+(\boldsymbol{u}_{\mathrm{III}}^{(j-1)},\boldsymbol{u}_{\mathrm{II}}^{(j)})}{\boldsymbol{u}_{\mathrm{III}}^{(j-1)}-\boldsymbol{u}_{\mathrm{III}}^{(j)}}+\frac{\beta_1}{2\gamma}\cdot\frac{1}{\boldsymbol{u}_{\mathrm{III}}^{(j-1)}-\boldsymbol{u}_{\mathrm{II}}^{(j)}}\right)s_{i,2\hat{n}-j}(\boldsymbol{u}_{\mathrm{III}}^{(n)})\right.$$

$$\left.+\left(\frac{\beta_1}{2\gamma}\cdot\frac{f^+(\boldsymbol{u}_{\mathrm{III}}^{(j-1)},\boldsymbol{u}_{\mathrm{II}}^{(j)})}{\boldsymbol{u}_{\mathrm{III}}^{(j-1)}-\boldsymbol{u}_{\mathrm{III}}^{(j)}}+\left(\beta_0+\frac{\beta_2}{2\gamma}\right)\frac{1}{\boldsymbol{u}_{\mathrm{III}}^{(j-1)}-\boldsymbol{u}_{\mathrm{II}}^{(j)}}\right)s_{j,2\hat{n}-i}(\boldsymbol{u}_{\mathrm{III}}^{(n)})\right]$$

$$\times\Psi(\boldsymbol{u}_{\mathrm{I}}^{(1\dots n)})\,,\quad(93)$$

*where*

$$\beta_0=\frac{f^-(\boldsymbol{u}_{\mathrm{III}}^{(n-1)},\boldsymbol{u}_{\mathrm{II}}^{(n-1)})\,f^+(\boldsymbol{u}_{\mathrm{III}}^{(n-1)},\tilde{\boldsymbol{u}}_{\mathrm{III}}^{(n)})}{(\boldsymbol{u}_{\mathrm{II}}^{(n-1)}-\tilde{\boldsymbol{u}}_{\mathrm{III}}^{(n)})(\boldsymbol{u}_{\mathrm{III}}^{(n-1)}-\boldsymbol{u}_{\mathrm{III}}^{(n)})(\boldsymbol{u}_{\mathrm{III}}^{(n)}-\tilde{\boldsymbol{u}}_{\mathrm{III}}^{(n)})}\,,$$

$$\beta_1=\frac{\boldsymbol{u}_{\mathrm{III}}^{(n-1)}-\boldsymbol{u}_{\mathrm{III}}^{(n)}}{\boldsymbol{u}_{\mathrm{III}}^{(n-1)}-\boldsymbol{u}_{\mathrm{II}}^{(n)}}\left(\boldsymbol{u}_{\mathrm{II}}^{(n-1)}-\tilde{\boldsymbol{u}}_{\mathrm{III}}^{(n)}+1+\frac{\boldsymbol{u}_{\mathrm{III}}^{(n-1)}-\tilde{\boldsymbol{u}}_{\mathrm{III}}^{(n)}}{\boldsymbol{u}_{\mathrm{II}}^{(n-1)}-\boldsymbol{u}_{\mathrm{II}}^{(n)}}\right),\tag{94}$$

$$\beta_2=f^+(\boldsymbol{u}_{\mathrm{II}}^{(n-1)},\boldsymbol{u}_{\mathrm{II}}^{(n)})\frac{\boldsymbol{u}_{\mathrm{II}}^{(n-1)}-\tilde{\boldsymbol{u}}_{\mathrm{III}}^{(n)}}{\boldsymbol{u}_{\mathrm{III}}^{(n-1)}-\boldsymbol{u}_{\mathrm{II}}^{(n)}}+\frac{(\boldsymbol{u}_{\mathrm{II}}^{(n-1)}-\boldsymbol{u}_{\mathrm{III}}^{(n)})(\boldsymbol{u}_{\mathrm{III}}^{(n-1)}-\tilde{\boldsymbol{u}}_{\mathrm{III}}^{(n)})+1}{\boldsymbol{u}_{\mathrm{II}}^{(n-1)}-\boldsymbol{u}_{\mathrm{II}}^{(n)}}\,,$$

*and*

$$\gamma = (\boldsymbol{u}_{\mathrm{II}}^{(n-1)} - \tilde{\boldsymbol{u}}_{\mathrm{III}}^{(n)})(\boldsymbol{u}_{\mathrm{III}}^{(n-1)} - \tilde{\boldsymbol{u}}_{\mathrm{III}}^{(n)})(\boldsymbol{u}_{\mathrm{II}}^{(n-1)} - \boldsymbol{u}_{\mathrm{III}}^{(n)})(\boldsymbol{u}_{\mathrm{III}}^{(n-1)} - \boldsymbol{u}_{\mathrm{III}}^{(n)}), \tag{95}$$

and $\boldsymbol{u}_{\mathrm{III}}^{(n)} = u_j^{(n)}$ for any $1 \le j \le m_n$, and $\boldsymbol{u}_{\mathrm{III}}^{(r)} = \boldsymbol{u}_{\mathrm{II}}^{(s)} = \emptyset$ for all $1 \le r < i$ and $1 \le s \le n$ in the first sum, $\boldsymbol{u}_{\mathrm{II}}^{(r)} = \boldsymbol{u}_{\mathrm{III}}^{(s)} = \emptyset$ for all $1 \le r < i$ and $1 \le s < n$ in the second sum, $\boldsymbol{u}_{\mathrm{II}}^{(r)} = \boldsymbol{u}_{\mathrm{III}}^{(s)} = \emptyset$ for all $1 \le r < i$ and $1 \le s < n$ in the third sum and $\boldsymbol{u}_{\mathrm{III}}^{(r)} = \boldsymbol{u}_{\mathrm{II}}^{(s)} = \emptyset$ for all $1 \le r < i$ and $1 \le s < j$ in the last sum.

*Example* 4.7. When $n = 1$, the recurrence relation (93) gives

$$\Psi(\boldsymbol{u}^{(1)}) = s_{12}(\boldsymbol{u}_{\mathrm{III}}^{(1)})\Psi(\boldsymbol{u}_{\mathrm{I}}^{(1)}) + \sum_{|\boldsymbol{u}_{\mathrm{II}}^{(1)}|=1} \frac{\Gamma_2(\boldsymbol{u}_{\mathrm{II}}^{(1)}; \boldsymbol{u}_{\mathrm{I}}^{(1)})}{\boldsymbol{u}_{\mathrm{II}}^{(1)} - \tilde{\boldsymbol{u}}_{\mathrm{III}}^{(1)}} s_{13}(\boldsymbol{u}_{\mathrm{III}}^{(1)})\Psi(\boldsymbol{u}_{\mathrm{I}}^{(1)}), \tag{96}$$

where $\boldsymbol{u}_{\mathrm{III}}^{(1)} = u_j^{(1)}$ for any $1 \le j \le m_1$. When $n = 2$, we have

$$\Psi(\boldsymbol{u}^{(1,2)}) = s_{23}(\boldsymbol{u}_{\mathrm{III}}^{(2)})\Psi(\boldsymbol{u}_{\mathrm{I}}^{(1,2)}) + \sum_{|\boldsymbol{u}_{\mathrm{III}}^{(1)}|=1} \frac{\Gamma_2(\boldsymbol{u}_{\mathrm{III}}^{(1)}; \boldsymbol{u}_{\mathrm{I}}^{(1,2)})}{\boldsymbol{u}_{\mathrm{III}}^{(1)} - \boldsymbol{u}_{\mathrm{III}}^{(2)}} s_{13}(\boldsymbol{u}_{\mathrm{III}}^{(2)})\Psi(\boldsymbol{u}_{\mathrm{I}}^{(1,2)})$$

$$+ \sum_{|\boldsymbol{u}_{\mathrm{II}}^{(1,2)}|=(1,1)} \frac{\Gamma_2(\boldsymbol{u}_{\mathrm{II}}^{(1)}; \boldsymbol{u}_{\mathrm{I}}^{(1,2)})\Gamma_3(\boldsymbol{u}_{\mathrm{II}}^{(2)}; \boldsymbol{u}_{\mathrm{I}}^{(1,2)})}{\boldsymbol{u}_{\mathrm{II}}^{(2)} - \tilde{\boldsymbol{u}}_{\mathrm{III}}^{(2)}}$$

$$\times \left( \frac{f^+(\boldsymbol{u}_{\mathrm{II}}^{(1)}, \boldsymbol{u}_{\mathrm{II}}^{(2)})}{\boldsymbol{u}_{\mathrm{II}}^{(1)} - \boldsymbol{u}_{\mathrm{III}}^{(2)}} s_{14}(\boldsymbol{u}_{\mathrm{III}}^{(2)}) + \frac{1}{\boldsymbol{u}_{\mathrm{II}}^{(1)} - \boldsymbol{u}_{\mathrm{II}}^{(2)}} s_{25}(\boldsymbol{u}_{\mathrm{III}}^{(2)}) \right) \Psi(\boldsymbol{u}_{\mathrm{I}}^{(1,2)})$$

$$+ \sum_{|\boldsymbol{u}_{\mathrm{II}}^{(1,2)}|=(2,1)} \frac{\Gamma_2(\boldsymbol{u}_{\mathrm{II}}^{(1)}; \boldsymbol{u}_{\mathrm{I}}^{(1,2)})\Gamma_3(\boldsymbol{u}_{\mathrm{II}}^{(2)}; \boldsymbol{u}_{\mathrm{I}}^{(1,2)})}{\boldsymbol{u}_{\mathrm{II}}^{(2)} - \tilde{\boldsymbol{u}}_{\mathrm{III}}^{(2)}} K(\boldsymbol{u}_{\mathrm{II}}^{(1)} | \boldsymbol{u}_{\mathrm{II,III}}^{(2)}) s_{15}(\boldsymbol{u}_{\mathrm{III}}^{(2)})\Psi(\boldsymbol{u}_{\mathrm{I}}^{(1,2)})$$

$$+ \sum_{|\boldsymbol{u}_{\mathrm{II}}^{(2)}|=1} \frac{\Gamma_3(\boldsymbol{u}_{\mathrm{II}}^{(2)}; \boldsymbol{u}_{\mathrm{I}}^{(2)})}{\boldsymbol{u}_{\mathrm{II}}^{(2)} - \tilde{\boldsymbol{u}}_{\mathrm{III}}^{(2)}} s_{24}(\boldsymbol{u}_{\mathrm{III}}^{(2)})\Psi(\boldsymbol{u}_{\mathrm{I}}^{(1,2)}), \tag{97}$$

where $\boldsymbol{u}_{\mathrm{III}}^{(2)} = u_j^{(2)}$ for any $1 \le j \le m_2$, and $\boldsymbol{u}_{\mathrm{II}}^{(1)} = \boldsymbol{u}_{\mathrm{II}}^{(2)} = \emptyset$ in the first sum, $\boldsymbol{u}_{\mathrm{III}}^{(1)} = \emptyset$ in the second sum, $\boldsymbol{u}_{\mathrm{III}}^{(1)} = \emptyset$ in the third sum and $\boldsymbol{u}_{\mathrm{III}}^{(1)} = \boldsymbol{u}_{\mathrm{II}}^{(1)} = \emptyset$ in the last sum.

The technical Lemma below will assist us in proving Proposition 4.6.

**Lemma 4.8.** *Let* $\Psi_j(\boldsymbol{u}^{(1\ldots n)})$ *denote a* $Y^+(\mathfrak{gl}_{2n+1})$*-based Bethe vector with the reference vector* $\eta_j^{\boldsymbol{m}} := (E_{12}^{(\hat{n})})_{\dot{a}_j} \eta^{\boldsymbol{m}}$. *Then*

$$\Psi_j(\boldsymbol{u}^{(1\ldots n)}) = \sum_{1 \le i \le j} \frac{1}{u_j^{(n)} - u_i^{(n)} + 1} \frac{\Gamma_{\hat{n}}(u_i^{(n)}, \boldsymbol{u}^{(1\ldots n)} \backslash u_i^{(n)})}{\prod_{k>j} f^+(u_k^{(n)}, u_i^{(n)})} \Psi(\boldsymbol{u}^{(1\ldots n)} \backslash u_i^{(n)}). \tag{98}$$

*Proof.* Recall (49) and consider level-$(n-1)$ vector

$$\overrightarrow{\prod_{j>1}} R_{\dot{a}_1 \ddot{a}_j}^{(\hat{n},\hat{n})}(\tilde{u}_1^{(n)} - u_j^{(n)}) \Psi_1(\boldsymbol{u}^{(1\ldots n-1)} | \boldsymbol{u}^{(n)}). \tag{99}$$

With the help of Yang-Baxter equation we can move the product of $R$-matrices all way to the reference vector $\eta_1^{\boldsymbol{m}}$. As a result of this, the level-$(n-1)$ nested monodromy matrix (40) takes the form

$$\overleftarrow{\prod_{i>1}} \widehat{R}_{\dot{a}_i a}^{(\hat{n},\hat{n})}(u_i^{(n)} - \nu) \overleftarrow{\prod_{i>1}} \widehat{R}_{\ddot{a}_i a}^{(\hat{n},\hat{n})}(\tilde{u}_i^{(n)} - \nu) \widehat{R}_{\ddot{a}_1 a}^{(\hat{n},\hat{n})}(u_1^{(n)} - \nu) \widehat{R}_{\ddot{a}_1 a}^{(\hat{n},\hat{n})}(\tilde{u}_1^{(n)} - \nu) A_a^{(\hat{n})}(\nu). \tag{100}$$

In the space $L^{(n-1)\prime}$, it is equivalent to $T_a^{(n)\prime}(\nu; \mathbf{u}^{(n)}\backslash u_1^{(n)})$. Next, recall (65) and note that

$$\overrightarrow{\prod_{j>1}} R_{\dot{a}_1\ddot{a}_j}^{(\hat{n},\hat{n})}(\tilde{u}_1^{(n)} - u_j^{(n)}) \cdot \eta_1^{\mathbf{m}} = f^+(u_1^{(n)}, \tilde{\mathbf{u}}^{(n)}\backslash\tilde{u}_1^{(n)})\, \eta_1^{\mathbf{m}}\,. \tag{101}$$

Hence, vector (99) can be expanded in the space $V_{\dot{a}_1}^{(\hat{n})} \otimes V_{\ddot{a}_1}^{(\hat{n})}$ as

$$f^+(u_1^{(n)}, \tilde{\mathbf{u}}^{(n)}\backslash\tilde{u}_1^{(n)}) \cdot E_1^{(\hat{n})} \otimes E_1^{(\hat{n})} \otimes \Psi_1(\mathbf{u}^{(1\ldots n-1)}\,|\,\mathbf{u}^{(n)}\backslash u_1^{(n)})\,. \tag{102}$$

From (50) note that $\mathsf{L}_{\dot{a}_1\ddot{a}_1}^{(n)}(\nu) \cdot E_1^{(\hat{n})} \otimes E_1^{(\hat{n})} = s_{\hat{n}\hat{n}}(\nu)$. Defining relations of $Y^+(\mathfrak{gl}_{2n+1})$ imply that

$$s_{\hat{n}\hat{n}}(u_1^{(n)})\overleftarrow{\prod_{i<n}}\mathscr{B}^{(i)}(\mathbf{u}^{(i)}; \mathbf{u}^{(i+1\ldots n)}\backslash u_1^{(n)}) = \overleftarrow{\prod_{i<n}}\mathscr{B}^{(i)}(\mathbf{u}^{(i)}; \mathbf{u}^{(i+1\ldots n)}\backslash u_1^{(n)})s_{\hat{n}\hat{n}}(u_1^{(n)}) + UWT\,,$$

where $UWT$ denotes "unwanted" terms, all of which act by 0 on $\eta_1^{\mathbf{m}}$. We have thus shown that

$$\Psi_1(\mathbf{u}^{(1\ldots n)}) = \mathscr{B}^{(n)}(\mathbf{u}^{(n)}\backslash u_1^{(n)})\mathsf{L}_{\dot{a}_1\ddot{a}_1}^{(n)}(u_1^{(n)})\overrightarrow{\prod_{j>1}}R_{\dot{a}_1\ddot{a}_j}^{(\hat{n},\hat{n})}(\tilde{u}_1^{(n)} - u_j^{(n)})\Psi_1(\mathbf{u}^{(1\ldots n-1)}\,|\,\mathbf{u}^{(n)})$$

$$= \mu_{\hat{n}}(\nu)f^+(u_1^{(n)}, \tilde{\mathbf{u}}^{(n)}\backslash\tilde{u}_1^{(n)})\Psi(\mathbf{u}^{(1\ldots n)}\backslash u_1^{(n)})\,. \tag{103}$$

This gives the $j = 1$ case of the claim. Then, using Yang-Baxter equation, Lemma 3.4, and the identity

$$\eta_{j+1}^{\mathbf{m}} = f^+(u_j^{(n)}, u_{j+1}^{(n)})\check{R}_{\dot{a}_{j+1}\dot{a}_j}^{(\hat{n},\hat{n})}(u_{j+1}^{(n)} - u_j^{(n)})\check{R}_{\ddot{a}_{j+1}\ddot{a}_j}^{(\hat{n},\hat{n})}(u_j^{(n)} - u_{j+1}^{(n)}) \cdot \eta_j^{\mathbf{m}} + \frac{1}{u_{j+1}^{(n)} - u_j^{(n)}}\,\eta_j^{\mathbf{m}}\,,$$

we find

$$\Psi_{j+1}(\mathbf{u}^{(1\ldots n)}) = f^+(u_j^{(n)}, u_{j+1}^{(n)})\Psi_j(\mathbf{u}_{u_j^{(n)}\leftrightarrow u_{j+1}^{(n)}}^{(1\ldots n)}) + \frac{1}{u_{j+1}^{(n)} - u_j^{(n)}}\,\Psi_j(\mathbf{u}^{(1\ldots n)})\,. \tag{104}$$

A simple induction on $j$ together with Lemma 3.7 gives the wanted result. $\qquad\square$

*Proof of Proposition 4.6.* The main idea of the proof is similar to that of Proposition 4.4. We start from the level-$(n-1)$ vector (91) and move operator $\mathscr{R}^{(\hat{n})}(u_{m_n}^{(n)}; \mathbf{u}^{(n)}\backslash u_{m_n}^{(n)})$ all way to the reference vector $\eta^{\mathbf{m}}$. In the odd case $E_{\hat{1}}^{(\hat{n})} = E_2^{(n+1)}$ giving (recall (52))

$$\mathscr{R}^{(\hat{n})}(u_{m_n}^{(n)}, \mathbf{u}^{(n)}\backslash u_{m_n}^{(n)}) \cdot \eta^{\mathbf{m}} = \eta^{\mathbf{m}} + \sum_{j<m_n}\frac{\prod_{j<k<m_n}f^+(u_k^{(n)}, \tilde{u}_{m_n}^{(n)})}{u_j^{(n)} - \tilde{u}_{m_n}^{(n)}}P_{\dot{a}_j\ddot{a}_{m_n}}^{(\hat{n},\hat{n})}\eta^{\mathbf{m}}\,. \tag{105}$$

Hence, in the odd case we can rewrite (91) as

$$\dot{\Psi}_{2,1}(\mathbf{u}^{(1\ldots n-1)}\,|\,\mathbf{u}^{(n)}) + \sum_{j<m_n}\frac{\prod_{j<k<m_n}f^+(u_k^{(n)}, \tilde{u}_{m_n}^{(n)})}{u_j^{(n)} - \tilde{u}_{m_n}^{(n)}}\,\dot{\Psi}_{2,2;j}(\mathbf{u}^{(1\ldots n-1)}\,|\,\mathbf{u}^{(n)})\,, \tag{106}$$

where $\dot{\Psi}_{k,l}$ and $\dot{\Psi}_{k,l;j}$ denote level-$(n-1)$ Bethe vectors based on the transfer matrix (92) and reference vectors $(E_{k,2}^{(\hat{n})})_{\dot{a}_{m_n}}(E_{l,1}^{(\hat{n})})_{\ddot{a}_{m_n}}\eta^{\mathbf{m}}$ and $(E_{k,2}^{(\hat{n})})_{\dot{a}_{m_n}}(E_{l,1}^{(\hat{n})})_{\ddot{a}_{m_n}}(E_{1,2}^{(\hat{n})})_{\dot{a}_j}\eta^{\mathbf{m}}$, respectively.

Consider the second term in (106). Acting with $\mathscr{B}^{(n)}(\boldsymbol{u}^{(n)}\backslash u_{m_n}^{(n)})$ and applying Lemma 4.8 gives

$$
\sum_{i\leq j<m_n}\prod_{j<k<m_n}\frac{f^+(u_k^{(n)},\tilde{u}_{m_n}^{(n)})}{f^+(u_k^{(n)},u_i^{(n)})}\cdot\frac{\Gamma_{\hat{n}}(u_i^{(n)},\boldsymbol{u}^{(1...n)}\backslash(u_i^{(n)},u_{m_n}^{(n)}))}{(u_j^{(n)}-u_i^{(n)}+1)(u_j^{(n)}-\tilde{u}_{m_n}^{(n)})}
$$
$$
\times\mathscr{B}^{(n)}(\boldsymbol{u}^{(n)}\backslash(u_i^{(n)},u_{m_n}^{(n)}))\,\dot{\Psi}_{2,2}(\boldsymbol{u}^{(1...n-1)}\,|\,\boldsymbol{u}^{(n)}\backslash u_i^{(n)}). \tag{107}
$$

Using the identity

$$
\frac{1}{u_i^{(n)}-\tilde{u}_{m_n}^{(n)}}=\sum_{i\leq j<m_n}\frac{1}{(u_j^{(n)}-u_i^{(n)}+1)(u_j^{(n)}-\tilde{u}_{m_n}^{(n)})}\prod_{j<k<m_n}\frac{f^+(u_k^{(n)},\tilde{u}_{m_n}^{(n)})}{f^+(u_k^{(n)},u_i^{(n)})}, \tag{108}
$$

which follows by a descending induction on $i$, expression (107) becomes

$$
\sum_{i<m_n}\frac{\Gamma_{\hat{n}}(u_i^{(n)};\boldsymbol{u}^{(n)}\backslash(u_i^{(n)},u_{m_n}^{(n)}))}{u_i^{(n)}-\tilde{u}_{m_n}^{(n)}}\,\mathscr{B}^{(n)}(\boldsymbol{u}^{(n)}\backslash(u_i^{(n)},u_{m_n}^{(n)}))\,\dot{\Psi}_{2,2}(\boldsymbol{u}^{(1...n-1)}\,|\,\boldsymbol{u}^{(n)}\backslash u_i^{(n)}). \tag{109}
$$

Thus, acting with $\lfloor_{\dot{a}_{m_n}\ddot{a}_{m_n}}^{(n)}(u_{m_n}^{(n)})\,\mathscr{B}^{(n)}(\boldsymbol{u}^{(n)}\backslash u_{m_n}^{(n)})$ on (106) we obtain

$$
\Psi(\boldsymbol{u}^{(1...n)})=\lfloor_{\dot{a}_{m_n}\ddot{a}_{m_n}}^{(n)}(u_{m_n}^{(n)})\Bigg(\mathscr{B}^{(n)}(\boldsymbol{u}^{(n)}\backslash u_{m_n}^{(n)})\,\dot{\Psi}_{2,1}(\boldsymbol{u}^{(1...n-1)}\,|\,\boldsymbol{u}^{(n)})
$$
$$
+\sum_{i<m_n}\frac{\Gamma_{\hat{n}}(u_i^{(n)};\boldsymbol{u}^{(1...n)}\backslash(u_i^{(n)},u_{m_n}^{(n)}))}{u_i^{(n)}-\tilde{u}_{m_n}^{(n)}}
$$
$$
\times\mathscr{B}^{(n)}(\boldsymbol{u}^{(n)}\backslash(u_i^{(n)},u_{m_n}^{(n)}))\,\dot{\Psi}_{2,2}(\boldsymbol{u}^{(1...n-1)}\,|\,\boldsymbol{u}^{(n)}\backslash u_i^{(n)})\Bigg). \tag{110}
$$

We will view vectors $\dot{\Psi}_{2,1}$ and $\dot{\Psi}_{2,2}$ as $Y(\mathfrak{gl}_n)$-based Bethe vectors and apply $Y(\mathfrak{gl}_n)$-based recurrence relations.

First, consider vector $\dot{\Psi}_{2,2}$. Its reference vector is annihilated by the $(j,i)$-th entries of the monodromy matrix (92) satisfying the condition $i<j$. Hence, we may use (84) to obtain an expansion in the space $V_{\dot{a}_{m_n}}^{(\hat{n})}\otimes V_{\ddot{a}_{m_n}}^{(\hat{n})}$. Taking $\boldsymbol{u}_{\mathrm{III}}^{(n)}=u_{m_n}^{(n)}$, the second term inside the brackets of (110) becomes (we have singled out the $i<j=n$ terms for further convenience)

$$
\sum_{1\leq i\leq n}\sum_{\substack{|\boldsymbol{u}_{\mathrm{II,III}}^{(r)}|=(2,0)\\i\leq r<n}}\sum_{|\boldsymbol{u}_{\mathrm{II}}^{(n)}|=1}\prod_{i<k\leq n}K(\boldsymbol{u}_{\mathrm{II}}^{(k-1)}\,|\,\boldsymbol{u}_{\mathrm{II,III}}^{(k)})\,\Gamma_k(\boldsymbol{u}_{\mathrm{II}}^{(k-1)};\boldsymbol{u}_{\mathrm{I}}^{(1...n)})
$$
$$
\times\frac{\Gamma_{\hat{n}}(\boldsymbol{u}_{\mathrm{II}}^{(n)};\boldsymbol{u}_{\mathrm{I}}^{(1...n)})}{\boldsymbol{u}_{\mathrm{II}}^{(n)}-\tilde{\boldsymbol{u}}_{\mathrm{III}}^{(n)}}\,E_{\bar{\imath}}^{(\hat{n})}\otimes E_{\bar{\imath}}^{(\hat{n})}\otimes\Psi(\boldsymbol{u}_{\mathrm{I}}^{(1...n)}) \tag{111}
$$
$$
+\sum_{1\leq i<n}\sum_{|\boldsymbol{u}_{\mathrm{II}}^{(r)}|=1}\prod_{i<k<n}\frac{\Gamma_k(\boldsymbol{u}_{\mathrm{II}}^{(k-1)};\boldsymbol{u}_{\mathrm{I}}^{(1...n)})}{\boldsymbol{u}_{\mathrm{II}}^{(k-1)}-\boldsymbol{u}_{\mathrm{II}}^{(k)}}\cdot\frac{\Gamma_n(\boldsymbol{u}_{\mathrm{II}}^{(n-1)};\boldsymbol{u}_{\mathrm{I}}^{(1...n)})\Gamma_{\hat{n}}(\boldsymbol{u}_{\mathrm{II}}^{(n)};\boldsymbol{u}_{\mathrm{I}}^{(1...n)})}{\boldsymbol{u}_{\mathrm{II}}^{(n)}-\tilde{\boldsymbol{u}}_{\mathrm{III}}^{(n)}}
$$
$$
\times\Bigg(\frac{f^+(\boldsymbol{u}_{\mathrm{II}}^{(n-1)},\tilde{\boldsymbol{u}}_{\mathrm{III}}^{(n)})}{\boldsymbol{u}_{\mathrm{II}}^{(n-1)}-\boldsymbol{u}_{\mathrm{III}}^{(n)}}E_{\bar{\imath}}^{(\hat{n})}\otimes E_2^{(\hat{n})}+\frac{1}{\boldsymbol{u}_{\mathrm{II}}^{(n-1)}-\tilde{\boldsymbol{u}}_{\mathrm{III}}^{(n)}}E_2^{(\hat{n})}\otimes E_{\bar{\imath}}^{(\hat{n})}\Bigg)\otimes\Psi(\boldsymbol{u}_{\mathrm{I}}^{(1...n)}) \tag{112}
$$

$$+ \sum_{\substack{1 \leq i < j < n}} \sum_{\substack{|u_{\mathrm{III}}^{(r)}|=1 \\ i \leq r < n}} \sum_{\substack{|u_{\mathrm{II}}^{(s)}|=1 \\ j \leq s \leq n}} \prod_{i < k < j} \frac{\Gamma_k(u_{\mathrm{III}}^{(k-1)}; u_{\mathrm{I}}^{(1...n)})}{u_{\mathrm{III}}^{(k-1)} - u_{\mathrm{III}}^{(k)}} \cdot \Gamma_j(u_{\mathrm{III}}^{(j-1)}; u_{\mathrm{I}}^{(1...n)})$$

$$\times \prod_{j < k < n} \frac{\Gamma_k(u_{\mathrm{II}}^{(k-1)}; u_{\mathrm{I}}^{(1...n)}) \Gamma_k(u_{\mathrm{III}}^{(k-1)}; u_{\mathrm{I,II}}^{(1...n)})}{(u_{\mathrm{II}}^{(k-1)} - u_{\mathrm{II}}^{(k)})(u_{\mathrm{III}}^{(k-1)} - u_{\mathrm{III}}^{(k)})} \cdot \frac{\Gamma_n(u_{\mathrm{II,III}}^{(n-1)}; u_{\mathrm{I}}^{(1...n)}) \Gamma_{\hat{n}}(u_{\mathrm{II}}^{(n)}; u_{\mathrm{I}}^{(1...n)})}{u_{\mathrm{II}}^{(n)} - \tilde{u}_{\mathrm{III}}^{(n)}}$$

$$\times \frac{f^-(u_{\mathrm{III}}^{(n-1)}, u_{\mathrm{II}}^{(n-1)}) f^+(u_{\mathrm{III}}^{(n-1)}, \tilde{u}_{\mathrm{III}}^{(n)})}{(u_{\mathrm{II}}^{(n-1)} - \tilde{u}_{\mathrm{III}}^{(n)})(u_{\mathrm{III}}^{(n-1)} - u_{\mathrm{III}}^{(n)})}$$

$$\times \left( \frac{f^+(u_{\mathrm{III}}^{(j-1)}, u_{\mathrm{II}}^{(j)})}{u_{\mathrm{III}}^{(j-1)} - u_{\mathrm{III}}^{(j)}} E_{\bar{i}}^{(\hat{n})} \otimes E_{\bar{j}}^{(\hat{n})} + \frac{1}{u_{\mathrm{III}}^{(j-1)} - u_{\mathrm{II}}^{(j)}} E_{\bar{j}}^{(\hat{n})} \otimes E_{\bar{i}}^{(\hat{n})} \right) \otimes \Psi(u_{\mathrm{I}}^{(1...n)}). \tag{113}$$

Next, consider vector $\dot{\Psi}_{2,1}$. This time we can not apply expansion (84). Instead, we will use the composite model approach to obtain the wanted expansion. Set $L^{\mathrm{II}} := V_{\dot{a}_{m_n}}^{(\hat{n})} \otimes V_{\ddot{a}_{m_n}}^{(\hat{n})}$ and $L^{\mathrm{I}} := W_{\dot{a} \backslash \dot{a}_{m_n}}^{(\hat{n})} \otimes W_{\ddot{a} \backslash \ddot{a}_{m_n}}^{(\hat{n})} \otimes (L^{(n)})^0$ so that $L^{(n-1)} \cong L^{\mathrm{II}} \otimes L^{\mathrm{I}}$. Recall (54) and set

$$\dashv_{a_i^{n-1}, k}^{\mathrm{II}}(v) := \sum_{j < n} (E_j^{(n-1)})_{a_i^{n-1}}^* \otimes \left[ R_{\dot{a}_{m_n} a}^{(\hat{n}, \hat{n})}(v - u_{m_n}^{(n)}) R_{\ddot{a}_{m_n} a}^{(\hat{n}, \hat{n})}(v - \tilde{u}_{m_n}^{(n)}) \right]_{n-j, k},$$

$$\lfloor_k^{\mathrm{I}}(v) := \left[ T_a^{(k+1)}(u_i^{(k)}; u^{(n)} \backslash u_{m_n}^{(n)}) \right]_{k, n}.$$

The cases when $k = n, \hat{n}$ will be denoted by

$$\lfloor_{a_i^{n-1}}^{\mathrm{II}}(v) := \dashv_{a_i^{n-1}, n}^{\mathrm{II}}(v), \qquad \sqrt{}_{a_i^{n-1}}^{\mathrm{II}}(v) := \dashv_{a_i^{n-1}, \hat{n}}^{\mathrm{II}}(v), \qquad \lceil^{\mathrm{I}}(v) := \lfloor_n^{\mathrm{I}}(v), \qquad \rfloor^{\mathrm{I}}(v) := \lfloor_{\hat{n}}^{\mathrm{I}}(v),$$

so that

$$\lfloor_{a_i^{n-1}}^{(n-1)}(v; u^{(n)}) = \sum_{k < n} \dashv_{a_i^{n-1}, k}^{\mathrm{II}}(v) \lfloor_k^{\mathrm{I}}(v) + \lfloor_{a_i^{n-1}}^{\mathrm{II}}(v) \lceil^{\mathrm{I}}(v) + \sqrt{}_{a_i^{n-1}}^{\mathrm{II}}(v) \rfloor^{\mathrm{I}}(v).$$

This notation is reminiscent of the Bethe Ansatz notation commonly used in the composite model approach only $\sqrt{}_{a_i^{n-1}}^{\mathrm{II}}$ is an additional creation operator specific to the case at hand. Consider the II-labelled operators. Their action on the reference state $E_2^{(\hat{n})} \otimes E_1^{(\hat{n})} \in L^{\mathrm{II}}$ is given by

$$\dashv_{a_i^{n-1}, j}^{\mathrm{II}}(v) \cdot E_2^{(\hat{n})} \otimes E_1^{(\hat{n})} = (E_{n-j}^{(n-1)})_{a_i^{n-1}}^* \cdot E_2^{(\hat{n})} \otimes E_1^{(\hat{n})},$$

$$\lfloor_{a_i^{n-1}}^{\mathrm{II}}(v) \cdot E_2^{(\hat{n})} \otimes E_1^{(\hat{n})} = \frac{1}{v - u_{m_n}^{(n)}} \sum_{j < n} (E_j^{(n-1)})_{a_i^{n-1}}^* \cdot E_{j+2}^{(\hat{n})} \otimes E_1^{(\hat{n})},$$

$$\sqrt{}_{a_i^{n-1}}^{\mathrm{II}}(v) \cdot E_2^{(\hat{n})} \otimes E_1^{(\hat{n})} = \frac{1}{v - \tilde{u}_{m_n}^{(n)}} \sum_{j < n} (E_j^{(n-1)})_{a_i^{n-1}}^* \left( \frac{1}{v - u_{m_n}^{(n)}} E_{j+2}^{(\hat{n})} \otimes E_2^{(\hat{n})} + E_2^{(\hat{n})} \otimes E_{j+2}^{(\hat{n})} \right),$$

$$\sqrt{}_{a_l^{n-1}}^{\mathrm{II}}(w) \lfloor_{a_i^{n-1}}^{\mathrm{II}}(v) \cdot E_2^{(\hat{n})} \otimes E_1^{(\hat{n})} = \frac{1}{(w - \tilde{u}_{m_n}^{(n)})(v - u_{m_n}^{(n)})} \sum_{j, k < n} (E_j^{(n-1)})_{a_l^{n-1}}^* (E_k^{(n-1)})_{a_i^{n-1}}^*$$

$$\times \left( \frac{1}{w - u_{m_n}^{(n)}} E_{j+2}^{(\hat{n})} \otimes E_{k+2}^{(\hat{n})} + E_{k+2}^{(\hat{n})} \otimes E_{j+2}^{(\hat{n})} \right).$$

The products $\lfloor^{\text{II}}_{a^{n-1}_j}(v)\lfloor^{\text{II}}_{a^{n-1}_i}(u)$, $\sqrt{}^{\text{II}}_{a^{n-1}_j}(v)\sqrt{}^{\text{II}}_{a^{n-1}_i}(u)$, and $\sqrt{}^{\text{II}}_{a^{n-1}_k}(w)\sqrt{}^{\text{II}}_{a^{n-1}_j}(v)\lfloor^{\text{II}}_{a^{n-1}_i}(u)$ act by zero on $E^{(\hat{n})}_2 \otimes E^{(\hat{n})}_1$. The homogeneous ($aa$ and $bb$, $pp$) exchange relations of the II-labelled operators are analogous to (55) and (56), respectively. The mixed ($ab$, $ap$, $bp$) exchange relations have the form

$$\dashv^{\text{II}}_{a^{n-1}_j}(v)\lfloor^{\text{II}}_{a^{n-1}_i}(u) = \lfloor^{\text{II}}_{a^{n-1}_i}(u)\dashv^{\text{II}}_{a^{n-1}_j}(v)R^{(n-1,n-1)}_{a^{n-1}_i,a^{n-1}_j}(u-v) + \frac{1}{u-v}\lfloor^{\text{II}}_{a^{n-1}_i}(v)\dashv^{\text{II}}_{a^{n-1}_j}(u)P^{(n-1,n-1)}_{a^{n-1}_j,a^{n-1}_i}.$$

Consider the I-labelled operators. The $dc$, $cb$, $db$ exchange relations have the form

$$\lceil^{\text{I}}(v)\rfloor^{\text{I}}(u) = f^-(v,u)\rfloor^{\text{I}}(u)\lceil^{\text{I}}(v) + \frac{1}{v-u}\rfloor^{\text{I}}(v)\lceil^{\text{I}}(u).$$

The standard Bethe Ansatz arguments then imply

$$\overleftarrow{\prod_i}\lfloor^{(n-1)}_{a^{n-1}_i}(u^{(n-1)}_i;\boldsymbol{u}^{(n)}) \cdot E^{(\hat{n})}_2 \otimes E^{(\hat{n})}_1 \otimes \Psi^{(n-2)}(\boldsymbol{u}^{(1\ldots n-2)}\,|\,\boldsymbol{u}^{(n-1,n)}\backslash u^{(n)}_{m_n})$$

$$= \Bigg[ E^{(\hat{n})}_2 \otimes E^{(\hat{n})}_1 \otimes \overleftarrow{\prod_i}\lfloor^{\text{I}}_{a^{n-1}_i}(u^{(n-1)}_i) \tag{114}$$

$$+ \sum_j \frac{f^-(u^{(n-1)}_j,\boldsymbol{u}^{(n-1)}\backslash u^{(n-1)}_j)}{u^{(n-1)}_j-u^{(n)}_{m_n}}\sum_{k<n}E^{(\hat{n})}_{k+2}\otimes E^{(\hat{n})}_1 \otimes (E^{(n-1)}_k)^*_{a^{n-1}_j}$$

$$\times \overleftarrow{\prod_{i\neq j}}\lfloor^{\text{I}}_{a^{n-1}_i}(u^{(n-1)}_i)\lceil^{\text{I}}(u^{(n-1)}_j) \tag{115}$$

$$+ \sum_j \frac{f^-(u^{(n-1)}_j,\boldsymbol{u}^{(n-1)}\backslash u^{(n-1)}_j)}{u^{(n-1)}_j-\tilde{u}^{(n)}_{m_n}}\sum_{k<n}\left(\frac{1}{u^{(n-1)}_j-u^{(n)}_{m_n}}E^{(\hat{n})}_{k+2}\otimes E^{(\hat{n})}_2 + E^{(\hat{n})}_2\otimes E^{(\hat{n})}_{k+2}\right)$$

$$\otimes (E^{(n-1)}_k)^*_{a^{n-1}_j}\overleftarrow{\prod_{i\neq j}}\lfloor^{\text{I}}_{a^{n-1}_i}(u^{(n-1)}_i)\rfloor^{\text{I}}(u^{(n-1)}_j) \tag{116}$$

$$+ \sum_{j<j'}f^-((u^{(n-1)}_j,u^{(n-1)}_{j'}),\boldsymbol{u}^{(n-1)}\backslash(u^{(n-1)}_j,u^{(n-1)}_{j'}))$$

$$\times \sum_{k,l<n}\Bigg(\frac{1}{\gamma}\Big(\alpha_{11}E^{(\hat{n})}_{k+2}\otimes E^{(\hat{n})}_{l+2} + \alpha_{12}E^{(\hat{n})}_{l+2}\otimes E^{(\hat{n})}_{k+2}\Big)\otimes(E^{(n-1)}_k)^*_{a^{n-1}_j}(E^{(n-1)}_l)^*_{a^{n-1}_{j'}}$$

$$\times \overleftarrow{\prod_{i\neq j,j'}}\lfloor^{\text{I}}_{a^{n-1}_i}(u^{(n-1)}_i)\rfloor^{\text{I}}(u^{(n-1)}_{j'})\lceil^{\text{I}}(u^{(n-1)}_j)$$

$$+ \frac{1}{\gamma}\Big(\alpha_{21}E^{(\hat{n})}_{k+2}\otimes E^{(\hat{n})}_{l+2} + \alpha_{22}E^{(\hat{n})}_{l+2}\otimes E^{(\hat{n})}_{k+2}\Big)\otimes(E^{(n-1)}_k)^*_{a^{n-1}_j}(E^{(n-1)}_l)^*_{a^{n-1}_{j'}}$$

$$\times \overleftarrow{\prod_{i\neq j,j'}}\lfloor^{\text{I}}_{a^{n-1}_i}(u^{(n-1)}_i)\rfloor^{\text{I}}(u^{(n-1)}_j)\lceil^{\text{I}}(u^{(n-1)}_{j'})\Bigg)\Bigg] \tag{117}$$

$$\times \Psi(\boldsymbol{u}^{(1\ldots n-2)}\,|\,\boldsymbol{u}^{(n-1,n)}\backslash u^{(n)}_{m_n}),$$

where

$$\alpha_{11} := (u_{j'}^{(n-1)} - u_{m_n}^{(n)})(u_j^{(n-1)} - \tilde{u}_{m_n}^{(n)}) - (u_{j'}^{(n-1)} - u_{m_n}^{(n)})/(u_j^{(n-1)} - u_{j'}^{(n-1)}),$$

$$\alpha_{12} := u_j^{(n-1)} - \tilde{u}_{m_n}^{(n)} - ((u_j^{(n-1)} - u_{m_n}^{(n)})(u_{j'}^{(n-1)} - \tilde{u}_{m_n}^{(n)}) + 1)/(u_j^{(n-1)} - u_{j'}^{(n-1)}),$$

$$\alpha_{21} := f^+(u_j^{(n-1)}, u_{j'}^{(n-1)})(u_{j'}^{(n-1)} - u_{m_n}^{(n)}), \tag{118}$$

$$\alpha_{22} := f^+(u_j^{(n-1)}, u_{j'}^{(n-1)})((u_j^{(n-1)} - u_{m_n}^{(n)})(u_{j'}^{(n-1)} - \tilde{u}_{m_n}^{(n)}) + 1),$$

$$\gamma := (u_j^{(n-1)} - u_{m_n}^{(n)})(u_j^{(n-1)} - \tilde{u}_{m_n}^{(n)})(u_{j'}^{(n-1)} - u_{m_n}^{(n)})(u_{j'}^{(n-1)} - \tilde{u}_{m_n}^{(n)}).$$

We will consider the terms (114–117) individually.

First, consider the term (114). Acting with $\lfloor_{\dot{a}_{m_n} \ddot{a}_{m_n}}^{(n)}(u_{m_n}^{(n)}) \mathscr{B}^{(n)}(\boldsymbol{u}^{(n)} \backslash u_{m_n}^{(n)})$ gives the $i = n$ case of the first term on the right hand side of (93).

Next, consider the term (115). The operator $\lceil^{\mathsf{I}}(u_j^{(n-1)})$ acts on $\Psi(\boldsymbol{u}^{(1...n-2)} | \boldsymbol{u}^{(n-1,n)} \backslash u_{m_n}^{(n)})$ via multiplication by $f^+(u_j^{(n-1)}, \boldsymbol{u}^{(n)} \backslash u_{m_n}^{(n)}) \mu_n(u_j^{(n-1)})$ giving

$$\sum_j \frac{\Gamma_n(u_j^{(n-1)}; \boldsymbol{u}^{(1...n)} \backslash (u_j^{(n-1)}, u_{m_n}^{(n)}))}{u_j^{(n-1)} - u_{m_n}^{(n)}} \sum_{k<n} E_{k+2}^{(\hat{n})} \otimes E_1^{(\hat{n})} \otimes (E_k^{(n-1)})_{a_j^{n-1}}^*$$
$$\times \Psi(\boldsymbol{u}^{(1...n-1)} \backslash u_j^{(n-1)} | u_j^{(n-1)}, \boldsymbol{u}^{(n)} \backslash u_{m_n}^{(n)}). \tag{119}$$

Using (83), we expand $\Psi(\boldsymbol{u}^{(1...n-1)} \backslash u_j^{(n-1)} | u_j^{(n-1)}, \boldsymbol{u}^{(n)} \backslash u_{m_n}^{(n)})$ in the space $V_{a_j^{n-1}}^{(n-1)}$:

$$\sum_{\substack{i<n \\ i \le r < n-1}} \sum_{|\boldsymbol{u}_{\mathrm{III}}^{(r)}|=1} \prod_{i<k<n} \frac{\Gamma_k(\boldsymbol{u}_{\mathrm{III}}^{(k-1)}; \boldsymbol{u}_{\mathrm{I}}^{(1...n)})}{\boldsymbol{u}_{\mathrm{III}}^{(k-1)} - \boldsymbol{u}_{\mathrm{III}}^{(k)}} E_{n-i}^{(n-1)} \otimes \Psi(\boldsymbol{u}_{\mathrm{I}}^{(1...n-1)} | \boldsymbol{u}_{\mathrm{I}}^{(n)}), \tag{120}$$

where $\boldsymbol{u}_{\mathrm{III}}^{(n-1)} := u_j^{(n-1)}$ and $\boldsymbol{u}_{\mathrm{I}}^{(n)} := \boldsymbol{u}^{(n)} \backslash u_{m_n}^{(n)}$. Substituting (120) into (119) yields

$$\sum_{\substack{i<n \\ i \le r < n}} \sum_{|\boldsymbol{u}_{\mathrm{II}}^{(r)}|=1} \prod_{i<k\le n} \frac{\Gamma_k(\boldsymbol{u}_{\mathrm{III}}^{(k-1)}; \boldsymbol{u}_{\mathrm{I}}^{(1...n)})}{\boldsymbol{u}_{\mathrm{III}}^{(k-1)} - \boldsymbol{u}_{\mathrm{III}}^{(k)}} E_{\hat{n}-i+1}^{(\hat{n})} \otimes E_1^{(\hat{n})} \otimes \Psi(\boldsymbol{u}_{\mathrm{I}}^{(1...n-1)} | \boldsymbol{u}_{\mathrm{I}}^{(n)}). \tag{121}$$

Acting with $\lfloor_{\dot{a}_{m_n} \ddot{a}_{m_n}}^{(n)}(u_{m_n}^{(n)}) \mathscr{B}^{(n)}(\boldsymbol{u}^{(n)} \backslash u_{m_n}^{(n)})$ gives the $i < n$ cases of the first term on the right hand side of (93).

We are now ready to consider the term (116). Let $\eta^{\mathsf{I}}$ denote the restriction of $\eta^{\boldsymbol{m}}$ to the space $L^{\mathsf{I}}$. Set $\eta_l^{\mathsf{I}} := (E_{12}^{(\hat{n})})_{\ddot{a}_l} \cdot \eta^{\mathsf{I}}$. Using the explicit form of $\mathsf{J}^{\mathsf{I}}(u_j^{(n-1)})$ we find

$$\mathsf{J}^{\mathsf{I}}(u_j^{(n-1)}) \cdot \eta^{\mathsf{I}} = \sum_{l<m_n} \frac{\prod_{k<l} f^+(u_j^{(n-1)}, u_k^{(n)})}{u_j^{(n-1)} - u_l^{(n)}} \mu_n(u_j^{(n-1)}) \eta_l^{\mathsf{I}}, \tag{122}$$

giving

$$\sum_j \frac{f^-(u_j^{(n-1)}, \boldsymbol{u}^{(n-1)} \backslash u_j^{(n-1)})}{u_j^{(n-1)} - \tilde{u}_{m_n}^{(n)}} \sum_{k<n} \left( \frac{1}{u_j^{(n-1)} - u_{m_n}^{(n)}} E_{k+2}^{(\hat{n})} \otimes E_2^{(\hat{n})} + E_2^{(\hat{n})} \otimes E_{k+2}^{(\hat{n})} \right) \otimes (E_k^{(n-1)})_{a_j^{n-1}}^*$$

$$\times \sum_{l<m_n} \frac{\prod_{k<l} f^+(u_j^{(n-1)}, u_k^{(n)})}{u_j^{(n-1)} - u_l^{(n)}} \mu_n(u_j^{(n-1)}) \Psi_l(\boldsymbol{u}^{(1...n-1)} \backslash u_j^{(n-1)} | u_j^{(n-1)}, \boldsymbol{u}^{(n)} \backslash u_{m_n}^{(n)}). \tag{123}$$

Acting with $\mathscr{B}^{(n)}(\boldsymbol{u}^{(n)}\backslash u_{m_n}^{(n)})$ and applying Lemma 4.8 to the second line of (123) gives

$$
\sum_{i \le l < m_n} \frac{\prod_{k<l} f^+(u_j^{(n-1)}, u_k^{(n)})}{\prod_{l<k<m_n} f^+(u_k^{(n)}, u_i^{(n)})} \cdot \frac{\Gamma_{\hat{n}}(u_i^{(n)}, \boldsymbol{u}^{(1...n)}\backslash(u_i^{(n)}, u_{m_n}^{(n)}))}{(u_l^{(n)} - u_i^{(n)} + 1)(u_j^{(n-1)} - u_l^{(n)})}
$$
$$
\times \mu_n(u_j^{(n-1)}) \, \Psi(\boldsymbol{u}^{(1...n)}\backslash(u_j^{(n-1)}, u_i^{(n)}, u_{m_n}^{(n)}) | u_j^{(n-1)}). \tag{124}
$$

Using the identity

$$
\frac{f^+(u_j^{(n-1)}, \boldsymbol{u}^{(n)}\backslash(u_i^{(n)}, u_{m_n}^{(n)}))}{u_j^{(n-1)} - u_i^{(n)}} = \sum_{i \le l < m_n} \frac{\prod_{k<l} f^+(u_j^{(n-1)}, u_k^{(n)})}{\prod_{l<k<m_n} f^+(u_k^{(n)}, u_i^{(n)})} \cdot \frac{1}{(u_l^{(n)} - u_i^{(n)} + 1)(u_j^{(n-1)} - u_l^{(n)})},
$$

which follows by a descending induction on $i$, expression (124) becomes

$$
\sum_{i < m_n} \frac{f^+(u_j^{(n-1)}, \boldsymbol{u}^{(n)}\backslash(u_i^{(n)}, u_{m_n}^{(n)}))}{u_j^{(n-1)} - u_i^{(n)}} \, \Gamma_{\hat{n}}(u_i^{(n)}, \boldsymbol{u}^{(1...n)}\backslash(u_i^{(n)}, u_{m_n}^{(n)}))
$$
$$
\times \mu_n(u_j^{(n-1)}) \, \Psi(\boldsymbol{u}^{(1...n)}\backslash(u_j^{(n-1)}, u_i^{(n)}, u_{m_n}^{(n)}) | u_j^{(n-1)}). \tag{125}
$$

Therefore, action of $\mathscr{B}^{(n)}(\boldsymbol{u}^{(n)}\backslash u_{m_n}^{(n)})$ on (123) gives

$$
\sum_j \sum_{i < m_n} \frac{\Gamma_n(u_j^{(n-1)}; \boldsymbol{u}^{(1...n)}\backslash(u_j^{(n-1)}, u_i^{(n)}, u_{m_n}^{(n)})) \, \Gamma_{\hat{n}}(u_i^{(n)}; \boldsymbol{u}^{(1...n)}\backslash(u_i^{(n)}, u_{m_n}^{(n)}))}{(u_j^{(n-1)} - u_i^{(n)})(u_j^{(n-1)} - \tilde{u}_{m_n}^{(n)})}
$$
$$
\times \sum_{k<n} \left( \frac{1}{u_j^{(n-1)} - u_{m_n}^{(n)}} E_{k+2}^{(\hat{n})} \otimes E_2^{(\hat{n})} + E_2^{(\hat{n})} \otimes E_{k+2}^{(\hat{n})} \right) \otimes (E_k^{(n-1)})_{a_j^{n-1}}^*
$$
$$
\times \Psi(\boldsymbol{u}^{(1...n)}\backslash(u_j^{(n-1)}, u_i^{(n)}, u_{m_n}^{(n)}) | u_j^{(n-1)}). \tag{126}
$$

Finally, we expand $\Psi(\boldsymbol{u}^{(1...n)}\backslash(u_j^{(n-1)}, u_i^{(n)}, u_{m_n}^{(n)}) | u_j^{(n-1)})$ in the space $V_{a_j^{n-1}}^{(n-1)}$ analogously to (120). This gives

$$
\sum_{i<n} \sum_{\substack{|\boldsymbol{u}_{\mathrm{II}}^{(r)}|=1 \\ i \le r \le n}} \prod_{i<k<n} \frac{\Gamma_k(\boldsymbol{u}_{\mathrm{II}}^{(k-1)}; \boldsymbol{u}_{\mathrm{I}}^{(1...n)})}{\boldsymbol{u}_{\mathrm{II}}^{(k-1)} - \boldsymbol{u}_{\mathrm{II}}^{(k)}} \cdot \frac{\Gamma_n(\boldsymbol{u}_{\mathrm{II}}^{(n-1)}; \boldsymbol{u}_{\mathrm{I}}^{(1...n)}) \, \Gamma_{\hat{n}}(\boldsymbol{u}_{\mathrm{II}}^{(n)}; \boldsymbol{u}_{\mathrm{I}}^{(1...n)})}{(\boldsymbol{u}_{\mathrm{II}}^{(n-1)} - \boldsymbol{u}_{\mathrm{II}}^{(n)})(\boldsymbol{u}_{\mathrm{II}}^{(n-1)} - \tilde{\boldsymbol{u}}_{\mathrm{III}}^{(n)})}
$$
$$
\times \left( \frac{1}{\boldsymbol{u}_{\mathrm{II}}^{(n-1)} - \boldsymbol{u}_{\mathrm{III}}^{(n)}} E_{\bar{i}}^{(\hat{n})} \otimes E_2^{(\hat{n})} + E_2^{(\hat{n})} \otimes E_{\bar{i}}^{(\hat{n})} \right) \otimes \Psi(\boldsymbol{u}_{\mathrm{I}}^{(1...n)}). \tag{127}
$$

Combining (127) with (112) and acting with $\mathsf{L}_{\dot{a}_{m_n} \ddot{a}_{m_n}}^{(n)}(u_{m_n}^{(n)})$ gives the second term on the right hand side of (93).

It remains to consider the term (117). Using the same arguments as above, and renaming $j \to p$, $j' \to p'$, we obtain

$$
\sum_{i < m_n} \sum_{p < p'} \Gamma_n((u_p^{(n-1)}, u_{p'}^{(n-1)}); \boldsymbol{u}^{(1...n)}\backslash(u_p^{(n-1)}, u_{p'}^{(n-1)}, u_i^{(n)}, u_{m_n}^{(n)})) \, \Gamma_{\hat{n}}(u_i^{(n)}; \boldsymbol{u}^{(1...n)}\backslash(u_i^{(n)}, u_{m_n}^{(n)}))
$$
$$
\times \sum_{k,l<n} \frac{1}{\gamma} \left( \beta_1 \, E_{k+2}^{(\hat{n})} \otimes E_{l+2}^{(\hat{n})} + \beta_2 \, E_{l+2}^{(\hat{n})} \otimes E_{k+2}^{(\hat{n})} \right) \otimes (E_k^{(n-1)})_{a_p^{n-1}}^* (E_l^{(n-1)})_{a_{p'}^{n-1}}^*
$$
$$
\times \Psi(\boldsymbol{u}^{(1...n)}\backslash(u_p^{(n-1)}, u_{p'}^{(n-1)}, u_i^{(n)}, u_{m_n}^{(n)}) | u_p^{(n-1)}, u_{p'}^{(n-1)}), \tag{128}
$$

where

$$
\begin{aligned}
\beta_1 &:= \frac{f^+(u_p^{(n-1)}, u_i^{(n)})}{u_{p'}^{(n-1)} - u_i^{(n)}} \alpha_{11} + \frac{f^+(u_{p'}^{(n-1)}, u_i^{(n)})}{u_p^{(n-1)} - u_i^{(n)}} \alpha_{21} \\
&= \frac{u_{p'}^{(n-1)} - u_{m_n}^{(n)}}{u_{p'}^{(n-1)} - u_i^{(n)}} \left( u_p^{(n-1)} - \tilde{u}_{m_n}^{(n)} + 1 + \frac{u_{p'}^{(n-1)} - \tilde{u}_{m_n}^{(n)}}{u_p^{(n-1)} - u_i^{(n)}} \right), \\
\beta_2 &:= \frac{f^+(u_p^{(n-1)}, u_i^{(n)})}{u_{p'}^{(n-1)} - u_i^{(n)}} \alpha_{12} + \frac{f^+(u_{p'}^{(n-1)}, u_i^{(n)})}{u_p^{(n-1)} - u_i^{(n)}} \alpha_{22} \\
&= f^+(u_p^{(n-1)}, u_i^{(n)}) \frac{u_p^{(n-1)} - \tilde{u}_{m_n}^{(n)}}{u_{p'}^{(n-1)} - u_i^{(n)}} + \frac{(u_p^{(n-1)} - u_{m_n}^{(n)})(u_{p'}^{(n-1)} - \tilde{u}_{m_n}^{(n)}) + 1}{u_p^{(n-1)} - u_i^{(n)}}.
\end{aligned}
\tag{129}
$$

Note that

$$
\beta_1 + \beta_2 = \frac{\gamma}{u_i^{(n)} - \tilde{u}_{m_n}^{(n)}} \left( K(u_p^{(n-1)}, u_{p'}^{(n-1)} \,|\, u_i^{(n)}, u_{m_n}^{(n)}) - K(u_p^{(n-1)}, u_{p'}^{(n-1)} \,|\, \tilde{u}_{m_n}^{(n)}, u_{m_n}^{(n)}) \right). \tag{130}
$$

We can now use (84) to expand vector

$$
\Psi(\boldsymbol{u}^{(1\ldots n)} \backslash (u_p^{(n-1)}, u_{p'}^{(n-1)}, u_i^{(n)}, u_{m_n}^{(n)}) \,|\, u_p^{(n-1)}, u_{p'}^{(n-1)}),
$$

in the space $V_{a_{p'}^{n-1}}^{(n-1)} \otimes V_{a_p^{n-1}}^{(n-1)}$:

$$
\sum_{\substack{1 \leq i < n \\ |\boldsymbol{u}_{\text{II,III}}^{(r)}| = (2,0) \\ i \leq r < n-1}} \sum \prod_{i < k < n} \Gamma_k(\boldsymbol{u}_{\text{II}}^{(k-1)}; \boldsymbol{u}_{\text{I}}^{(1\ldots n)}) \, K(\boldsymbol{u}_{\text{II}}^{(k-1)} \,|\, \boldsymbol{u}_{\text{II,III}}^{(k)}) \, E_{n-i}^{(n-1)} \otimes E_{n-i}^{(n-1)} \otimes \Psi(\boldsymbol{u}_{\text{I}}^{(1\ldots n)}) \quad (131)
$$

$$
+ \sum_{\substack{1 \leq i < j < n \\ |\boldsymbol{u}_{\text{III}}^{(r)}| = 1 \\ i \leq r < n-1}} \sum_{\substack{|\boldsymbol{u}_{\text{III}}^{(s)}| = 1 \\ j \leq s < n-1}} \sum \prod_{i < k < j} \frac{\Gamma_k(\boldsymbol{u}_{\text{III}}^{(k-1)}; \boldsymbol{u}_{\text{I}}^{(1\ldots n)})}{\boldsymbol{u}_{\text{III}}^{(k-1)} - \boldsymbol{u}_{\text{III}}^{(k)}} \cdot \Gamma_j(\boldsymbol{u}_{\text{III}}^{(j-1)}; \boldsymbol{u}_{\text{I}}^{(1\ldots n)})
$$

$$
\times \prod_{j < k < n} \frac{\Gamma_k(\boldsymbol{u}_{\text{II}}^{(k-1)}; \boldsymbol{u}_{\text{I}}^{(1\ldots n)}) \, \Gamma_k(\boldsymbol{u}_{\text{III}}^{(k-1)}; \boldsymbol{u}_{\text{I,II}}^{(1\ldots n)})}{(\boldsymbol{u}_{\text{II}}^{(k-1)} - \boldsymbol{u}_{\text{II}}^{(k)})(\boldsymbol{u}_{\text{III}}^{(k-1)} - \boldsymbol{u}_{\text{III}}^{(k)})}
$$

$$
\times \left( \frac{f^+(\boldsymbol{u}_{\text{III}}^{(j-1)}, \boldsymbol{u}_{\text{II}}^{(j)})}{\boldsymbol{u}_{\text{III}}^{(j-1)} - \boldsymbol{u}_{\text{III}}^{(j)}} E_{n-i}^{(n-1)} \otimes E_{n-j}^{(n-1)} + \frac{1}{\boldsymbol{u}_{\text{III}}^{(j-1)} - \boldsymbol{u}_{\text{II}}^{(j)}} E_{n-j}^{(n-1)} \otimes E_{n-i}^{(n-1)} \right) \otimes \Psi(\boldsymbol{u}_{\text{I}}^{(1\ldots n)}), \quad (132)
$$

where $\boldsymbol{u}_{\text{II}}^{(n-1)} := u_p^{(n-1)}$, $\boldsymbol{u}_{\text{III}}^{(n-1)} := u_{p'}^{(n-1)}$ and $\boldsymbol{u}_{\text{I}}^{(n)} := \boldsymbol{u}^{(n)} \backslash (u_i^{(n)}, u_{m_n}^{(n)})$.
Substituting the term (131) into (128) and applying (130) gives

$$
\sum_{\substack{1 \leq i < n \\ |\boldsymbol{u}_{\text{II}}^{(r)}| = 2 \\ i \leq r < n}} \sum_{|\boldsymbol{u}_{\text{II}}^{(n)}| = 1} \sum \prod_{i < k \leq n} \Gamma_k(\boldsymbol{u}_{\text{II}}^{(k-1)}; \boldsymbol{u}_{\text{I}}^{(1\ldots n)}) \prod_{i < k < n} K(\boldsymbol{u}_{\text{II}}^{(k-1)} \,|\, \boldsymbol{u}_{\text{II}}^{(k)})
$$

$$
\times \frac{\Gamma_{\hat{n}}(\boldsymbol{u}_{\text{II}}^{(n)}; \boldsymbol{u}_{\text{I}}^{(1\ldots n)})}{\boldsymbol{u}_{\text{II}}^{(n)} - \tilde{\boldsymbol{u}}_{\text{III}}^{(n)}} \left( K(\boldsymbol{u}_{\text{II}}^{(n-1)} \,|\, \boldsymbol{u}_{\text{II,III}}^{(n)}) - K(\boldsymbol{u}_{\text{II}}^{(n-1)} \,|\, \tilde{\boldsymbol{u}}_{\text{III}}^{(n)}, \boldsymbol{u}_{\text{III}}^{(n)}) \right) E_{\bar{\imath}}^{(\hat{n})} \otimes E_{\bar{\imath}}^{(\hat{n})} \otimes \Phi(\boldsymbol{u}_{\text{I}}^{(1\ldots n)}). \quad (133)
$$

Upon combining (133) with (111) and acting with $\mathsf{L}_{\hat{a}_{m_n} \check{a}_{m_n}}^{(n)}(u_{m_n}^{(n)})$ gives the third term on the right hand side of (93).

Finally, substituting (132) into (128) and exploiting symmetry of Bethe vectors gives

$$
\sum_{\substack{1\leq i<j<n \\ i\leq r<n}} \sum_{\substack{|\boldsymbol{u}_{\mathrm{III}}^{(r)}|=1 \\ }} \sum_{\substack{|\boldsymbol{u}_{\mathrm{II}}^{(s)}|=1 \\ j\leq s\leq n}} \prod_{i<k<j} \frac{\Gamma_k(\boldsymbol{u}_{\mathrm{III}}^{(k-1)};\boldsymbol{u}_{\mathrm{I}}^{(1\ldots n)})}{\boldsymbol{u}_{\mathrm{III}}^{(k-1)}-\boldsymbol{u}_{\mathrm{III}}^{(k)}} \cdot \Gamma_j(\boldsymbol{u}_{\mathrm{III}}^{(j-1)};\boldsymbol{u}_{\mathrm{I}}^{(1\ldots n)})
$$

$$
\times \prod_{j<k<n} \frac{\Gamma_k(\boldsymbol{u}_{\mathrm{II}}^{(k-1)};\boldsymbol{u}_{\mathrm{I}}^{(1\ldots n)})\,\Gamma_k(\boldsymbol{u}_{\mathrm{III}}^{(k-1)};\boldsymbol{u}_{\mathrm{I,II}}^{(1\ldots n)})}{(\boldsymbol{u}_{\mathrm{II}}^{(k-1)}-\boldsymbol{u}_{\mathrm{II}}^{(k)})(\boldsymbol{u}_{\mathrm{III}}^{(k-1)}-\boldsymbol{u}_{\mathrm{III}}^{(k)})} \cdot \Gamma_n(\boldsymbol{u}_{\mathrm{II,III}}^{(n-1)};\boldsymbol{u}_{\mathrm{I}}^{(1\ldots n)})\,\Gamma_{\hat{n}}(\boldsymbol{u}_{\mathrm{II}}^{(n)};\boldsymbol{u}_{\mathrm{I}}^{(1\ldots n)})
$$

$$
\times \frac{1}{2\gamma}\Bigg[\Bigg(\beta_2 \frac{f^+(\boldsymbol{u}_{\mathrm{III}}^{(j-1)},\boldsymbol{u}_{\mathrm{II}}^{(j)})}{\boldsymbol{u}_{\mathrm{III}}^{(j-1)}-\boldsymbol{u}_{\mathrm{III}}^{(j)}} + \beta_1 \frac{1}{\boldsymbol{u}_{\mathrm{III}}^{(j-1)}-\boldsymbol{u}_{\mathrm{II}}^{(j)}}\Bigg) E_{\bar{\imath}}^{(\hat{n})} \otimes E_{\bar{\jmath}}^{(\hat{n})}
$$

$$
+ \Bigg(\beta_1 \frac{f^+(\boldsymbol{u}_{\mathrm{III}}^{(j-1)},\boldsymbol{u}_{\mathrm{II}}^{(j)})}{\boldsymbol{u}_{\mathrm{III}}^{(j-1)}-\boldsymbol{u}_{\mathrm{III}}^{(j)}} + \beta_2 \frac{1}{\boldsymbol{u}_{\mathrm{III}}^{(j-1)}-\boldsymbol{u}_{\mathrm{II}}^{(j)}}\Bigg) E_{\bar{\jmath}}^{(\hat{n})} \otimes E_{\bar{\imath}}^{(\hat{n})}\Bigg] \otimes \Psi(\boldsymbol{u}_{\mathrm{I}}^{(1\ldots n)}). \tag{134}
$$

Combining (134) with (113) and acting with $\lfloor_{\dot{a}_{m_n}\ddot{a}_{m_n}}^{(n)}(u_{m_n}^{(n)})$ gives the last term on the right hand side of (93). □

## 4.3 Proof of Lemma 3.8

The idea of the proof is to construct a certain Bethe vector and evaluate this vector in two different ways. Equating the resulting expressions will yield the claim of the Lemma.

We begin by rewriting the wanted relation in a more convenient way. From (23) and (50) we find that

$$
\left\{\frac{p(v)}{u_i^{(n)}-v}\lfloor_{\dot{a}_{m_n}\ddot{a}_{m_n}}^{(n)}(v)\right\}^{v} = \lfloor_{\dot{a}_{m_n}\ddot{a}_{m_n}}^{(n)}(v)\left(\frac{f^+(u_i^{(n)},\tilde{v})}{u_i^{(n)}-v} + \frac{1}{u_i^{(n)}-\tilde{v}}\,P_{\dot{a}_{m_n}\ddot{a}_{m_n}}^{(\hat{n},\hat{n})}\right). \tag{135}
$$

Repeating the steps used in deriving (110) and applying (135) we rewrite (67) as

$$
s_{\hat{n}\hat{n}}(v)\Psi(\boldsymbol{u}^{(1\ldots n)}) = \Gamma_{\hat{n}}(v,\boldsymbol{u}^{(1\ldots n)})\Psi(\boldsymbol{u}^{(1\ldots n)})
$$

$$
- \sum_i \lfloor_{\dot{a}_{m_n}\ddot{a}_{m_n}}^{(n)}(v)\left(\frac{f^+(u_i^{(n)},\tilde{v})}{u_i^{(n)}-v} + \frac{1}{u_i^{(n)}-\tilde{v}}\,P_{\dot{a}_{m_n}\ddot{a}_{m_n}}^{(\hat{n},\hat{n})}\right)
$$

$$
\times \Gamma_{\hat{n}}(u_i^{(n)},\boldsymbol{u}^{(1\ldots n)}\backslash u_i^{(n)})\,\mathscr{B}^{(n)}(\boldsymbol{u}_{\sigma_i}^{(n)}\backslash u_i^{(n)})\,\dot{\Psi}_{2,1}(\boldsymbol{u}^{(1\ldots n-1)}\,|\,\boldsymbol{u}_{\sigma_i}^{(n)})
$$

$$
- \sum_{i\neq i'} \lfloor_{\dot{a}_{m_n-1}\ddot{a}_{m_n-1}}^{(n)}(v)\left(\frac{f^+(u_i^{(n)},\tilde{v})}{u_i^{(n)}-v} + \frac{1}{u_i^{(n)}-\tilde{v}}\,P_{\dot{a}_{m_n-1}\ddot{a}_{m_n-1}}^{(\hat{n},\hat{n})}\right)
$$

$$
\times \Gamma_{\hat{n}}((u_i^{(n)},u_{i'}^{(n)});\boldsymbol{u}^{(1\ldots n)}\backslash(u_i^{(n)},u_{i'}^{(n)}))\,\frac{f^-(u_i^{(n)},u_{i'}^{(n)})f^+(u_i^{(n)},\tilde{u}_{i'}^{(n)})}{u_{i'}^{(n)}-\tilde{u}_i^{(n)}}
$$

$$
\times \mathscr{B}^{(n)}(\boldsymbol{u}^{(n)}\backslash(u_i^{(n)},u_{i'}^{(n)}))\,\dot{\Psi}_{2,2}(\boldsymbol{u}^{(1\ldots n-1)}\,|\,\boldsymbol{u}_{\sigma_i}^{(n)}\backslash u_{i'}^{(n)}). \tag{136}
$$

Let $\Psi_{m_n+1}(\boldsymbol{u}^{(1\ldots n)}\cup v)$ denote a Bethe vector with $m_n+1$ level-$n$ excitations and the reference vector $\eta_{m_n+1}^{m} := (E_{12}^{(\hat{n})})_{\dot{a}_{m_n+1}}\eta^{m}$; here $v$ denotes the $(m_n+1)$-st level-$n$ Bethe root.

Applying (98) and (110) to this Bethe vector we obtain

$$
\Psi_{m_n+1}(\boldsymbol{u}^{(1\ldots n)}\cup v) = \Gamma_{\hat{n}}(v,\boldsymbol{u}^{(1\ldots n)})\Psi(\boldsymbol{u}^{(1\ldots n)})
$$

$$
-\sum_i \frac{f^+(u_i^{(n)},\tilde{v})}{u_i^{(n)}-v}\,\Gamma_{\hat{n}}(u_i^{(n)};\boldsymbol{u}^{(1\ldots n)}\backslash u_i^{(n)})
$$

$$
\times \lfloor_{\dot{a}_{m_n}\ddot{a}_{m_n}}^{(n)}(v)\,\mathscr{B}^{(n)}(\boldsymbol{u}^{(n)}\backslash u_i^{(n)})\dot{\Psi}_{2,1}(\boldsymbol{u}^{(1\ldots n-1)}\,|\,\boldsymbol{u}^{(n)}\backslash u_i^{(n)}\cup v)
$$

$$
-\sum_{i'\neq i}\Gamma_{\hat{n}}((u_i^{(n)},u_{i'}^{(n)});\boldsymbol{u}^{(1\ldots n)}\backslash(u_i^{(n)},u_{i'}^{(n)}))K(u_i^{(n)},u_{i'}^{(n)}\,|\,v,\tilde{v})f^+(u_i^{(n)},u_{i'}^{(n)})
$$

$$
\times \lfloor_{\dot{a}_{m_n-1}\ddot{a}_{m_n-1}}^{(n)}(v)\,\mathscr{B}^{(n)}(\boldsymbol{u}^{(n)}\backslash(u_i^{(n)},u_{i'}^{(n)}))
$$

$$
\times \dot{\Psi}_{2,2}(\boldsymbol{u}^{(1\ldots n-1)}\,|\,\boldsymbol{u}^{(n)}\backslash(u_i^{(n)},u_{i'}^{(n)})\cup v). \quad (137)
$$

Next, recall (105) and note that $P_{\dot{a}_i\ddot{a}_{m_n}}^{(\hat{n},\hat{n})}\eta_{m_n}^{\boldsymbol{m}} = P_{\dot{a}_{m_n}\ddot{a}_{m_n}}^{(\hat{n},\hat{n})}\eta_i^{\boldsymbol{m}}$ giving

$$
\mathscr{R}^{(\hat{n})}(u_{m_n}^{(n)};\boldsymbol{u}^{(n)}\backslash u_{m_n}^{(n)})\cdot\eta_{m_n}^{\boldsymbol{m}} = \eta_{m_n}^{\boldsymbol{m}} + \sum_{i<m_n}\frac{\prod_{i<k<m_n}f^+(u_k^{(n)},\tilde{u}_{m_n}^{(n)})}{u_i^{(n)}-\tilde{u}_{m_n}^{(n)}}\,P_{\dot{a}_{m_n}\ddot{a}_{m_n}}^{(\hat{n},\hat{n})}\eta_i^{\boldsymbol{m}}. \quad (138)
$$

This yields an analogue of (110) for $\Psi_{m_n+1}(\boldsymbol{u}^{(1\ldots n)}\cup v)$:

$$
\Psi_{m_n+1}(\boldsymbol{u}^{(1\ldots n)}\cup v) = \lfloor_{\dot{a}_{m_n+1}\ddot{a}_{m_n+1}}^{(n)}(v)\,\mathscr{B}^{(n)}(\boldsymbol{u}^{(n)})\dot{\Psi}_{1,1}(\boldsymbol{u}^{(1\ldots n-1)}\,|\,\boldsymbol{u}^{(n)}\cup v)
$$

$$
+\sum_i \frac{\Gamma_{\hat{n}}(u_i^{(n)};\boldsymbol{u}^{(1\ldots n)}\backslash u_i^{(n)})}{u_i^{(n)}-\tilde{v}}
$$

$$
\times \lfloor_{\dot{a}_{m_n}\ddot{a}_{m_n}}^{(n)}(v)\,\mathscr{B}^{(n)}(\boldsymbol{u}^{(n)}\backslash u_i^{(n)})\dot{\Psi}_{1,2}(\boldsymbol{u}^{(1\ldots n-1)}\,|\,\boldsymbol{u}^{(n)}\backslash u_i^{(n)}\cup v). \quad (139)
$$

The next step is to evaluate products of creation operators $\mathscr{B}^{(n)}$ and the dotted Bethe vectors $\dot{\Psi}$. This is done by applying the same techniques used in the proof of Proposition 4.6. Hence, we will skip the technical details and state the final expressions only.

Evaluating the named products in (137) and (139) gives

$$
\mathscr{B}^{(n)}(\boldsymbol{u}^{(n)}\backslash u_i^{(n)})\dot{\Psi}_{2,1}(\boldsymbol{u}^{(1\ldots n-1)}\,|\,\boldsymbol{u}^{(n)}\backslash u_i^{(n)}\cup v)
$$

$$
= E_2^{(\hat{n})}\otimes E_1^{(\hat{n})}\otimes\Psi(\boldsymbol{u}^{(1\ldots n)}\backslash u_i^{(n)})
$$

$$
+\sum_j \frac{\Gamma_n(u_j^{(n-1)};\boldsymbol{u}^{(1\ldots n)}\backslash(u_j^{(n-1)},u_i^{(n)}))}{u_j^{(n-1)}-v}
$$

$$
\times \sum_{1\le k<n} E_{k+2}^{(\hat{n})}\otimes E_1^{(\hat{n})}\otimes(E_k^{(n-1)})_{a_j^{n-1}}^*\Psi(\boldsymbol{u}^{(1\ldots n)}\backslash(u_j^{(n-1)},u_i^{(n)}\,|\,u_j^{(n-1)}))
$$

$$
+\sum_j\sum_{i'\neq i}\frac{\Gamma_n(u_j^{(n-1)};\boldsymbol{u}^{(1\ldots n)}\backslash(u_j^{(n-1)},u_i^{(n)},u_{i'}^{(n)}))\Gamma_{\hat{n}}(u_{i'}^{(n)};\boldsymbol{u}^{(1\ldots n)}\backslash(u_i^{(n)},u_{i'}^{(n)}))}{(u_j^{(n-1)}-\tilde{v})(u_j^{(n-1)}-u_{i'}^{(n)})}
$$

$$
\times \sum_{1\le k<n}\left(\frac{1}{u_j^{(n-1)}-v}E_{k+2}^{(\hat{n})}\otimes E_2^{(\hat{n})}+E_2^{(\hat{n})}\otimes E_{k+2}^{(\hat{n})}\right)\otimes(E_k^{(n-1)})_{a_j^{n-1}}^*
$$

$$
\times \Psi(\boldsymbol{u}^{(1\ldots n)}\backslash(u_j^{(n-1)},u_i^{(n)},u_{i'}^{(n)})\,|\,u_j^{(n-1)})
$$

$$+ \sum_{j<j'} \sum_{i'\neq i} \Gamma_n((u_j^{(n-1)}, u_{j'}^{(n-1)}); \boldsymbol{u}^{(1...n)}\backslash(u_j^{(n-1)}, u_{j'}^{(n-1)}, u_i^{(n)}, u_{i'}^{(n)})) \, \Gamma_{\hat{n}}(u_{i'}^{(n)}; \boldsymbol{u}^{(1...n)}\backslash(u_i^{(n)}, u_{i'}^{(n)}))$$

$$\times \sum_{1\leq k,l<n} \frac{1}{\gamma}\Big(\beta_1^{(21)} E_{k+2}^{(\hat{n})} \otimes E_{l+2}^{(\hat{n})} + \beta_2^{(21)} E_{l+2}^{(\hat{n})} \otimes E_{k+2}^{(\hat{n})}\Big) \otimes (E_k^{(n-1)})^*_{a_j^{n-1}} (E_l^{(n-1)})^*_{a_{j'}^{n-1}}$$

$$\times \Psi(\boldsymbol{u}^{(1...n)}\backslash(u_j^{(n-1)}, u_{j'}^{(n-1)}, u_i^{(n)}, u_{i'}^{(n)}) \,|\, u_j^{(n-1)}, u_{j'}^{(n-1)}), \quad (140)$$

and

$$\mathscr{B}^{(n)}(\boldsymbol{u}^{(n)}\backslash u_i^{(n)}) \, \dot{\Psi}_{1,2}(\boldsymbol{u}^{(1...n-1)} \,|\, \boldsymbol{u}^{(n)}\backslash u_i^{(n)} \cup v)$$

$$= E_1^{(\hat{n})} \otimes E_2^{(\hat{n})} \otimes \Psi(\boldsymbol{u}^{(1...n)}\backslash u_i^{(n)})$$

$$+ \sum_{j} \frac{\Gamma_n(u_j^{(n-1)}; \boldsymbol{u}^{(1...n)}\backslash(u_j^{(n-1)}, u_i^{(n)}))}{u_j^{(n-1)} - \tilde{v}}$$

$$\times \sum_{1\leq k<n} \left(\frac{1}{u_j^{(n-1)} - v} E_{k+2}^{(\hat{n})} \otimes E_1^{(\hat{n})} + E_1^{(\hat{n})} \otimes E_{k+2}^{(\hat{n})}\right) \otimes (E_k^{(n-1)})^*_{a_j^{n-1}}$$

$$\times \Psi(\boldsymbol{u}^{(1...n)}\backslash(u_j^{(n-1)}, u_i^{(n)}) \,|\, u_j^{(n-1)})$$

$$+ \sum_{j} \sum_{i'\neq i} \frac{\Gamma_n(u_j^{(n-1)}; \boldsymbol{u}^{(1...n)}\backslash(u_j^{(n-1)}, u_i^{(n)}, u_{i'}^{(n)})) \, \Gamma_{\hat{n}}(u_{i'}^{(n)}; \boldsymbol{u}^{(1...n)}\backslash(u_i^{(n)}, u_{i'}^{(n)}))}{(u_j^{(n-1)} - v)(u_j^{(n-1)} - u_{i'}^{(n)})}$$

$$\times \sum_{1\leq k<n} E_{k+2}^{(\hat{n})} \otimes E_2^{(\hat{n})} \otimes (E_k^{(n-1)})^*_{a_j^{n-1}} \Psi(\boldsymbol{u}^{(1...n)}\backslash(u_j^{(n-1)}, u_i^{(n)}, u_{i'}^{(n)}) \,|\, u_j^{(n-1)})$$

$$+ \sum_{j<j'} \sum_{i'\neq i} \Gamma_n((u_j^{(n-1)}, u_{j'}^{(n-1)}); \boldsymbol{u}^{(1...n)}\backslash(u_j^{(n-1)}, u_{j'}^{(n-1)}, u_i^{(n)}, u_{i'}^{(n)})) \, \Gamma_{\hat{n}}(u_{i'}^{(n)}; \boldsymbol{u}^{(1...n)}\backslash(u_i^{(n)}, u_{i'}^{(n)}))$$

$$\times \sum_{1\leq k,l<n} \frac{1}{\gamma}\Big(\beta_1^{(12)} E_{k+2}^{(\hat{n})} \otimes E_{l+2}^{(\hat{n})} + \beta_{12}^{(12)} E_{l+2}^{(\hat{n})} \otimes E_{k+2}^{(\hat{n})}\Big) \otimes (E_k^{(n-1)})^*_{a_j^{n-1}} (E_l^{(n-1)})^*_{a_{j'}^{n-1}}$$

$$\times \Psi(\boldsymbol{u}^{(1...n)}\backslash(u_j^{(n-1)}, u_{j'}^{(n-1)}, u_i^{(n)}, u_{i'}^{(n)}) \,|\, u_j^{(n-1)}, u_{j'}^{(n-1)}), \quad (141)$$

and

$$\mathscr{B}^{(n)}(\boldsymbol{u}^{(n)}) \, \dot{\Psi}_{1,1}(\boldsymbol{u}^{(1...n-1)} \,|\, \boldsymbol{u}^{(n)} \cup v)$$

$$= E_1^{(\hat{n})} \otimes E_1^{(\hat{n})} \otimes \Psi(\boldsymbol{u}^{(1...n)})$$

$$+ \sum_{j} \sum_{i} \frac{\Gamma_n(u_j^{(n-1)}; \boldsymbol{u}^{(1...n)}\backslash(u_j^{(n-1)}, u_i^{(n)})) \, \Gamma_{\hat{n}}(u_i^{(n)}; \boldsymbol{u}^{(1...n)}\backslash u_i^{(n)})}{u_j^{(n-1)} - u_i^{(n)}}$$

$$\times \sum_{1\leq k<n} \left(\frac{f^+(u_j^{(n-1)}, \tilde{v})}{u_j^{(n-1)} - v} E_{k+2}^{(\hat{n})} \otimes E_1^{(\hat{n})} + \frac{1}{u_j^{(n-1)} - \tilde{v}} E_1^{(\hat{n})} \otimes E_{k+2}^{(\hat{n})}\right) \otimes (E_k^{(n-1)})^*_{a_j^{n-1}}$$

$$\times \Psi(\boldsymbol{u}^{(1...n)}\backslash(u_j^{(n-1)}, u_i^{(n)}) \,|\, u_j^{(n-1)})$$

$$+ \sum_{j<j'} \sum_{i<i'} \Gamma_n((u_j^{(n-1)}, u_{j'}^{(n-1)}); \boldsymbol{u}^{(1...n)}\backslash(u_j^{(n-1)}, u_{j'}^{(n-1)}, u_i^{(n)}, u_{i'}^{(n)}))$$

$$\times \Gamma_{\hat{n}}((u_i^{(n)}, u_{i'}^{(n)}); \boldsymbol{u}^{(1...n)}\backslash(u_i^{(n)}, u_{i'}^{(n)})) K(u_j^{(n-1)}, u_{j'}^{(n-1)} \,|\, u_i^{(n)}, u_{i'}^{(n)}) f^+(u_i^{(n)}, \tilde{u}_{i'}^{(n)})$$

$$\times \sum_{1\leq k,l<n} \Big(\beta_1^{(11)} E_{k+2}^{(\hat{n})} \otimes E_{l+2}^{(\hat{n})} + \beta_2^{(11)} E_{l+2}^{(\hat{n})} \otimes E_{k+2}^{(\hat{n})}\Big) \otimes (E_k^{(n-1)})^*_{a_j^{n-1}} (E_l^{(n-1)})^*_{a_{j'}^{n-1}}$$

$$\times \Psi(\boldsymbol{u}^{(1...n)}\backslash(u_j^{(n-1)}, u_{j'}^{(n-1)}, u_i^{(n)}, u_{i'}^{(n)}) \,|\, u_j^{(n-1)}, u_{j'}^{(n-1)}), \quad (142)$$

and

$$
\begin{aligned}
&\mathscr{B}^{(n)}(\boldsymbol{u}^{(n)}\backslash(u_i^{(n)}, u_{i'}^{(n)}))\,\dot{\Psi}_{2,2}(\boldsymbol{u}^{(1\ldots n-1)}\,|\,\boldsymbol{u}^{(n)}\backslash(u_i^{(n)}, u_{i'}^{(n)})\cup v) \\
&\quad = E_2^{(\hat{n})}\otimes E_2^{(\hat{n})}\otimes\Psi(\boldsymbol{u}^{(1\ldots n)}\backslash(u_i^{(n)}, u_{i'}^{(n)})) \\
&\qquad +\sum_j \Gamma_n(u_j^{(n-1)};\boldsymbol{u}^{(1\ldots n)}\backslash(u_j^{(n-1)}, u_i^{(n)}, u_{i'}^{(n)})) \\
&\qquad\qquad \times\sum_{1\le k<n}\left(\frac{f^+(u_j^{(n-1)},\tilde{v})}{u_j^{(n-1)}-v}E_{k+2}^{(\hat{n})}\otimes E_2^{(\hat{n})}+\frac{1}{u_j^{(n-1)}-\tilde{v}}E_2^{(\hat{n})}\otimes E_{k+2}^{(\hat{n})}\right)\otimes(E_k^{(n-1)})^*_{a_j^{n-1}} \\
&\qquad\qquad\qquad \times\Psi(\boldsymbol{u}^{(1\ldots n)}\backslash(u_j^{(n-1)}, u_i^{(n)}, u_{i'}^{(n)})\,|\,u_j^{(n-1)}) \\
&\qquad +\sum_{j<j'}\Gamma_n((u_j^{(n-1)}, u_{j'}^{(n-1)});\boldsymbol{u}^{(1\ldots n)}\backslash(u_j^{(n-1)}, u_{j'}^{(n-1)}, u_i^{(n)}, u_{i'}^{(n)})) \\
&\qquad\qquad \times\sum_{1\le k,l<n}\left(\beta_1^{(11)}E_{k+2}^{(\hat{n})}\otimes E_{l+2}^{(\hat{n})}+\beta_2^{(11)}E_{l+2}^{(\hat{n})}\otimes E_{k+2}^{(\hat{n})}\right)\otimes(E_k^{(n-1)})^*_{a_j^{n-1}}(E_l^{(n-1)})^*_{a_{j'}^{n-1}} \\
&\qquad\qquad\qquad \times\Psi(\boldsymbol{u}^{(1\ldots n)}\backslash(u_j^{(n-1)}, u_{j'}^{(n-1)}, u_i^{(n)}, u_{i'}^{(n)})\,|\,u_j^{(n-1)}, u_{j'}^{(n-1)}, \boldsymbol{u}^{(n)}),\quad (143)
\end{aligned}
$$

where $\beta_1^{(21)}$, $\beta_2^{(21)}$ and $\gamma$ are given by (129) and (118) except $u_{m_n}^{(n)}$ should be replaced by $v$, and

$$
\begin{aligned}
\beta_1^{(12)} &:= \frac{u_{j'}^{(n-1)}-v}{u_{j'}^{(n-1)}-u_{i'}^{(n)}}\left(f^+(u_j^{(n-1)}, u_{i'}^{(n)})+\frac{(u_{j'}^{(n-1)}-u_{i'}^{(n)})(u_j^{(n-1)}-\tilde{v})}{u_j^{(n-1)}-u_{i'}^{(n)}}\right), \\
\beta_2^{(12)} &:= \frac{u_j^{(n-1)}-\tilde{v}}{u_{j'}^{(n-1)}-u_{i'}^{(n)}}f^+(u_{j'}^{(n-1)}, u_{i'}^{(n)})f^+(u_j^{(n-1)}, u_{j'}^{(n-1)}) \\
&\qquad +\frac{u_{j'}^{(n-1)}-\tilde{v}}{u_{j'}^{(n-1)}-u_{i'}^{(n)}}f^+(u_j^{(n-1)}, u_{i'}^{(n)})\left(u_j^{(n-1)}-v-\frac{1}{u_j^{(n-1)}-u_{j'}^{(n-1)}}\right), \\
\beta_1^{(11)} &:= \frac{f^+(u_j^{(n-1)},\tilde{v})}{(u_j^{(n-1)}-v)(u_{j'}^{(n-1)}-\tilde{v})},\qquad \beta_2^{(11)} := \frac{1}{u_{j'}^{(n-1)}-v}\left(\beta_1^{(11)}+\frac{1}{u_j^{(n-1)}-\tilde{v}}\right).
\end{aligned}
\tag{144}
$$

Adapting (140) and (143) to the relevant products in (136) allows us to rewrite the latter as

$$
\begin{aligned}
&\Gamma_{\hat{n}}(u_i^{(n)}, \boldsymbol{u}^{(1\ldots n)}\backslash u_i^{(n)})\,E_2^{(\hat{n})}\otimes E_1^{(\hat{n})}\otimes\Psi(\boldsymbol{u}^{(1\ldots n)}\backslash u_i^{(n)}) \\
&+\sum_j\frac{\Gamma_{\hat{n}}(u_i^{(n)}, \boldsymbol{u}^{(1\ldots n)}\backslash u_i^{(n)})\,\Gamma_n(u_j^{(n-1)};\boldsymbol{u}^{(1\ldots n)}\backslash(u_j^{(n-1)}, u_i^{(n)}))}{u_j^{(n-1)}-u_i^{(n)}} \\
&\qquad \times\sum_{1\le k<n}E_{k+2}^{(\hat{n})}\otimes E_1^{(\hat{n})}\otimes(E_k^{(n-1)})^*_{a_j^{n-1}}\Psi(\boldsymbol{u}^{(1\ldots n)}\backslash(u_j^{(n-1)}, u_i^{(n)})\,|\,u_j^{(n-1)}) \\
&+\sum_{i'\ne i}\Gamma_{\hat{n}}((u_i^{(n)}, u_{i'}^{(n)});\boldsymbol{u}^{(1\ldots n)}\backslash(u_i^{(n)}, u_{i'}^{(n)}))\frac{f^-(u_i^{(n)}, u_{i'}^{(n)})f^+(u_i^{(n)},\tilde{u}_{i'}^{(n)})}{u_{i'}^{(n)}-\tilde{u}_i^{(n)}} \\
&\qquad \times\left(E_2^{(\hat{n})}\otimes E_2^{(\hat{n})}\otimes\Psi(\boldsymbol{u}^{(1\ldots n)}\backslash(u_i^{(n)}, u_{i'}^{(n)}))+A\right),\quad (145)
\end{aligned}
$$

where

$$A := \sum_j \Gamma_n(u_j^{(n-1)}; \boldsymbol{u}^{(1...n)} \backslash (u_j^{(n-1)}, u_i^{(n)}, u_{i'}^{(n)}))$$

$$\times \sum_{1 \le k < n} \left( \frac{f^+(u_j^{(n-1)}, \tilde{u}_{i'}^{(n)})}{u_j^{(n-1)} - u_i^{(n)}} E_{k+2}^{(\hat{n})} \otimes E_2^{(\hat{n})} + \frac{1}{u_j^{(n-1)} - u_{i'}^{(n)}} E_2^{(\hat{n})} \otimes E_{k+2}^{(\hat{n})} \right) \otimes (E_k^{(n-1)})^*_{a_j^{n-1}}$$

$$\times \Psi^{(n-1)}(\boldsymbol{u}^{(1...n)} \backslash (u_j^{(n-1)}, u_i^{(n)}, u_{i'}^{(n)}) | u_j^{(n-1)})$$

$$+ \sum_{j < j'} \frac{\Gamma_n((u_j^{(n-1)}, u_{j'}^{(n-1)}); \boldsymbol{u}^{(1...n)} \backslash (u_j^{(n-1)}, u_{j'}^{(n-1)}, u_i^{(n)}, u_{i'}^{(n)}))}{(u_j^{(n-1)} - u_i^{(n)})(u_{j'}^{(n-1)} - u_{i'}^{(n)})}$$

$$\times \sum_{1 \le k, l < n} \left( f^+(u_j^{(n-1)}, u_i^{(n)}) E_{k+2}^{(\hat{n})} \otimes E_{l+2}^{(\hat{n})} + \theta \, E_{l+2}^{(\hat{n})} \otimes E_{k+2}^{(\hat{n})} \right) \otimes (E_k^{(n-1)})^*_{a_j^{n-1}} (E_l^{(n-1)})^*_{a_{j'}^{n-1}}$$

$$\times \Psi(\boldsymbol{u}^{(1...n)} \backslash (u_j^{(n-1)}, u_{j'}^{(n-1)}, u_i^{(n)}, u_{i'}^{(n)}) | u_j^{(n-1)}, u_{j'}^{(n-1)}),$$

and

$$\theta := \frac{(u_j^{(n-1)} - u_i^{(n)})(u_{j'}^{(n-1)} - u_{i'}^{(n)}) + u_j^{(n-1)} - u_{i'}^{(n)} + 1}{(u_j^{(n-1)} - u_{i'}^{(n)})(u_{j'}^{(n-1)} - u_i^{(n)})}.$$

The final step is to substitute (140)–(143) into the difference of (139) and (137), and (145) into (136), and equate the resulting expressions.

## 5 Conclusions

This paper is a continuation of [12], where twisted Yangian based models, known as one-dimensional "soliton non-preserving" open spin chains, were studied by means of algebraic Bethe Ansatz. The present paper extends the results of [12] to the odd case, when the bulk symmetry is $\mathfrak{gl}_{2n+1}$ and the boundary symmetry is $\mathfrak{so}_{2n+1}$. Theorem 3.9 states that Bethe vectors, defined by formula (66), are eigenvectors of the transfer matrix, defined by formula (68), provided Bethe equations (77) and (78) hold. It is important to note that Bethe equations for $Y^{\pm}(\mathfrak{gl}_N)$-based models were first considered in [1,9]. However, the completeness of solutions of such Bethe equations is still an open question. Investigation of higher-order transfer matrices and $Q$-operators might help to shed more light on this problem.

In Proposition 3.12 we presented a more symmetric form of the trace formula for Bethe vectors than the one found in [12]. This formula can be used to obtain Bethe vectors when the number of excitations is not large since the complexity of the "master" creation operator grows rapidly when the total excitation number increases. This is a well-known issue of trace formulas for both closed and open spin chains. Low rank examples of the "master" creation operator are given in Example 3.11.

We also obtained recurrence relations for twisted Yangian based Bethe vectors. They are given in Propositions 4.4 and 4.6 for even and odd cases, respectively. Repeated application of these relations allow us to express $Y^{\pm}(\mathfrak{gl}_N)$-based Bethe vectors in terms of $Y(\mathfrak{gl}_n)$-based Bethe vectors obeying recurrence relations found in [20] and recalled in Appendix A.3. The recurrence relations found in this paper provide elegant expressions when the rank is small, see Examples 4.5 and 4.7. The $n = 2$ even case in Example 4.5 may help investigating the open fishchain studied in [17]. However, recurrence relations become rather complex when the rank is not small, especially in the odd case. This raises a natural question, if there exists an alternative (simpler) method of constructing Bethe vectors for open spin chains. For closed spin chains the current ("Drinfeld New") presentation of Yangians and quantum loop

algebras [10] has played a significant role in obtaining not only recurrence relations, but also action relations, scalar products and norms of Bethe vectors, see [19–23]. Thus, it is natural to expect that a current presentation of twisted Yangians could pave a fruitful path for open spin chains analysis.

A current presentation of twisted Yangian $Y^+(\mathfrak{gl}_N)$ was recently obtained in [25]. (The rank 2 case was considered earlier in [6].) However, in [25] a different, the so-called non-split, presentation of twisted Yangian is considered (see Chapter 2 in [26]), which is based on the Chevalley involution of $\mathfrak{gl}_N$ and is not compatible (at least in a natural way) with the Bethe vacuum state. Nonetheless, we believe that the presentation obtained in [25] may have applications in open spin chain analysis and deserves attention. For example, integrable overlaps for twisted boundary states are constructed using the non-split presentation of twisted Yangians [15].

Overall, the approach presented in this paper does open a door to an exploration of scalar products and norms of Bethe vectors for twisted Yangian based models. However, developing Bethe Ansatz techniques in the current presentation of twisted Yangians might open a broader path to open spin chain analysis. An alternative path could be a development of separation of variable techniques along the lines of e.g. [18,31].

## Acknowledgments

The author thanks Allan Gerrard for collaboration at an early stage of this paper and for his contribution to Section 3.7. The author also thanks Andrii Liashyk for explaining $Y(\mathfrak{gl}_n)$-type recurrence relations and Paul Ryan for helpful discussions on applications of twisted Yangian based models.

## A  Appendix

### A.1  Weight grading of $Y^\pm(\mathfrak{gl}_N)$

Define an $n$-tuple $\omega_i \in \mathbb{Z}^n$ by $(\omega_i)_j := \delta_{ij}$ and recall the notation $\bar{j} = N - j + 1$. Then define weights of the elements $s_{ij}[r]$ using the following rule

$$\mathrm{wt}(s_{ij}[r]) := \sum_{i \le k < j} \omega_k + \sum_{\bar{j} \le k < \hat{n}} \omega_k, \quad \text{when} \quad i < j, \ i + j \le N + 1, \tag{A.1}$$

and require

$$\mathrm{wt}(s_{\bar{j}\bar{i}}[r]) = \mathrm{wt}(s_{ij}[r]), \quad \mathrm{wt}(s_{ji}[r]) = -\mathrm{wt}(s_{ij}[r]), \tag{A.2}$$

for all $1 \le i, j \le N$. Note that $\mathrm{wt}(s_{ii}[r]) = (0, \ldots, 0) \in \mathbb{Z}^n$. Extending linearly on all monomials this defines a weight grading on $Y^\pm(\mathfrak{gl}_N)$.

The recurrence relations (89) and (93) are compatible with this grading. The master creation operator (79) has the weight

$$\boldsymbol{\omega} := \mathrm{wt}\big(\mathscr{B}_N(\boldsymbol{u}^{(1\ldots n)})\big) = \begin{cases} (m_1, \ldots, m_{n-1}, m_n), & \text{when} \quad \hat{n} = n, \\ (m_1, \ldots, m_{n-1}, 2m_n), & \text{when} \quad \hat{n} = n + 1, \end{cases} \tag{A.3}$$

which we assign to the corresponding Bethe vector. Then (89) and (93) can be schematically written as

$$\Psi^{\boldsymbol{\omega}} = \sum_{\boldsymbol{\omega}' \in W} s_{\boldsymbol{\omega}'} \Psi^{\boldsymbol{\omega} - \boldsymbol{\omega}'}, \tag{A.4}$$

where $W$ is the set of weights of $s_{i,n+j}[r]$ with $1 \le i \le n$ and $1 \le j \le \hat{n}$, the $s_{\omega'}$ is a generating series of $Y^{\pm}(\mathfrak{gl}_N)$ of weight $\omega'$, and all scalar factors and spectral parameter dependencies are omitted, as in (1) and (2).

## A.2 Commutativity of transfer matrices

**Lemma A.1.** *Transfer matrices* $\tau(u)$ *defined by* (68) *form a commuting family of operators.*

*Proof.* We follow arguments in the Proof of Theorem 2.4 in [33]. In this proof, we will write $S_a(u)$ instead of $S_a^{(N)}(u)$ and $R_{ab}(u)$ instead of $R_{ab}^{(N,N)}(u)$. Then

$$\tau(u)\tau(v) = \mathrm{tr}_a\, M_a^{t_a}(u)\, S_a^{t_a}(u)\, \mathrm{tr}_b\, M_b(v)\, S_b(v) = \mathrm{tr}_{ab}\, M_a^{t_a}(u)\, M_b(v)\, S_a^{t_a}(u)\, S_b(v), \qquad \text{(A.5)}$$

where $t_a$ denotes the usual matrix transposition in the space labelled $a$. Upon inserting a resolution of identity in terms of $\widehat{R}$-matrices and using properties of matrix transposition and the trace (see Appendix A in [33]) we rewrite the right hand side of (A.5) as

$$\mathrm{tr}_{ab}\, M_a^{t_a}(u)\, M_b(v)\,(\widehat{R}_{ab}^{t_a}(\tilde{v}-u))^{-1}\,\widehat{R}_{ab}^{t_a}(\tilde{v}-u)\, S_a^{t_a}(u)\, S_b(v)$$
$$= \mathrm{tr}_{ab}\left(M_a^{t_a}(u)\,((\widehat{R}_{ab}^{t_a}(\tilde{v}-u))^{-1})^{t_b}\, M_b^{t_b}(v)\right)^{t_b}\left(S_a(u)\,\widehat{R}_{ab}(\tilde{v}-u)\, S_b(v)\right)^{t_a}$$
$$= \mathrm{tr}_{ab}\left(M_a^{t_a}(u)\,((\widehat{R}_{ab}^{t_a}(\tilde{v}-u))^{-1})^{t_b}\, M_b^{t_b}(v)\right)^{t_b t_a}\, S_a(u)\,\widehat{R}_{ab}(\tilde{v}-u)\, S_b(v). \qquad \text{(A.6)}$$

We insert a resolution identity in terms of $R$-matrices and use properties of matrix transposition and the trace once again. This gives

$$\mathrm{tr}_{ab}\left(M_a^{t_a}(u)\,((\widehat{R}_{ab}^{t_a}(\tilde{v}-u))^{-1})^{t_b}\, M_b^{t_b}(v)\right)^{t_b t_a}$$
$$\times (R_{ab}(u-v))^{-1}\, R_{ab}(u-v)\, S_a(u)\,\widehat{R}_{ab}(\tilde{v}-u)\, S_b(v)$$
$$= \mathrm{tr}_{ab}\left(((R_{ab}(u-v))^{-1})^{t_a t_b}\, M_a^{t_a}(u)\,((\widehat{R}_{ab}^{t_a}(\tilde{v}-u))^{-1})^{t_b}\, M_b^{t_b}(v)\right)^{t_b t_a}$$
$$\times R_{ab}(u-v)\, S_a(u)\,\widehat{R}_{ab}(\tilde{v}-u)\, S_b(v). \qquad \text{(A.7)}$$

The $R$-matrix (9) satisfies

$$((R_{ab}(u))^{-1})^{t_a t_b} = r(u)\, R_{ab}(-u), \qquad \left((\widehat{R}_{ab}^{t_a}(u))^{-1}\right)^{t_b} = r(u)\,\widehat{R}_{ab}(-u), \qquad \text{(A.8)}$$

where $r(u) := u^2/(u^2-1)$. Relations (A.8) and the dual twisted reflection equation (69) imply

$$\left(((R_{ab}(u-v))^{-1})^{t_a t_b}\, M_a^{t_a}(u)\,((\widehat{R}_{ab}^{t_a}(\tilde{v}-u))^{-1})^{t_b}\, M_b^{t_b}(v)\right)^{t_b t_a}$$
$$= r(u-v)\, r(\tilde{v}-u)\left(R_{ab}(v-u)\, M_a^{t_a}(u)\,\widehat{R}_{ab}(u-\tilde{v})\, M_b^{t_b}(v)\right)^{t_b t_a}$$
$$= r(u-v)\, r(\tilde{v}-u)\left(M_b^{t_b}(v)\,\widehat{R}_{ab}(u-\tilde{v})\, M_a^{t_a}(u)\, R_{ab}(v-u)\right)^{t_b t_a}$$
$$= \left(M_b^{t_b}(v)\,((\widehat{R}_{ab}^{t_a}(\tilde{v}-u))^{-1})^{t_b}\, M_a^{t_a}(u)\,((R_{ab}(u-v))^{-1})^{t_a t_b}\right)^{t_b t_a}. \qquad \text{(A.9)}$$

Applying (A.9) to the right hand side of (A.7) gives

$$\mathrm{tr}_{ab}\left(M_b^{t_b}(v)\,((R_{ab}^{t_a}(\tilde{v}-u))^{-1})^{t_b}\, M_a^{t_a}(u)\,((R_{ab}(u-v))^{-1})^{t_a t_b}\right)^{t_b t_a}\, S_b(v)\,\widehat{R}_{ab}(\tilde{v}-u)\, S_a(u)\, R_{ab}(u-v). \qquad \text{(A.10)}$$

It remains to repeat similar steps as above in reversed order and use cyclicity of the trace. The (A.10) then becomes

$$\mathrm{tr}_{ab}(R_{ab}(u-v))^{-1}\left(M_b^{t_b}(v)\,((R_{ab}^{t_a}(\tilde{v}-u))^{-1})^{t_b}\, M_a^{t_a}(u)\right)^{t_b t_a}\, S_b(v)\,\widehat{R}_{ab}(\tilde{v}-u)\, S_a(u)\, R_{ab}(u-v)$$
$$= \mathrm{tr}_{ab}\left(M_b^{t_b}(v)\,((R_{ab}^{t_a}(\tilde{v}-u))^{-1})^{t_b}\, M_a^{t_a}(u)\right)^{t_b t_a}\, S_b(v)\,\widehat{R}_{ab}(\tilde{v}-u)\, S_a(u)$$
$$= \mathrm{tr}_{ab}\left(M_b^{t_b}(v)\,((R_{ab}^{t_a}(\tilde{v}-u))^{-1})^{t_b}\, M_a^{t_a}(u)\right)^{t_b}\left(S_b(v)\,\widehat{R}_{ab}(\tilde{v}-u)\, S_a(u)\right)^{t_a}$$
$$= \mathrm{tr}_{ab}(R_{ab}^{t_a}(\tilde{v}-u))^{-1}\, M_b^{t_b}(v)\, M_a^{t_a}(u)\, S_b(v)\, S_a(u)^{t_a}\,\widehat{R}_{ab}^{t_a}(\tilde{v}-u)$$
$$= \mathrm{tr}_b\, M_b^{t_b}(v)\, S_b(v)\,\mathrm{tr}_a\, M_a^{t_a}(u)\, S_a(u)^{t_a}\,\widehat{R}_{ab}^{t_a}(\tilde{v}-u) = \tau(v)\,\tau(u), \qquad \text{(A.11)}$$

as required. $\qquad\square$

## A.3 A recurrence relation for $Y(\mathfrak{gl}_n)$-based models

The Proposition below is a restatement of Proposition 4.2 in [20] in terms of notation introduced in Section 4.1 and Proposition 4.1. Recall (82):

$$\Lambda_k(z; \boldsymbol{v}^{(1\ldots n-1)}) = f^-(z, \boldsymbol{v}^{(k-1)}) f^+(z, \boldsymbol{v}^{(k)}) \lambda_k(z).$$

Let $t_{ij}(z)$ denote the standard generating series of $Y(\mathfrak{gl}_n)$.

**Proposition A.2.** $Y(\mathfrak{gl}_n)$-based Bethe vectors satisfy the recurrence relation

$$\Phi(\boldsymbol{v}^{(1\ldots n-1)}) = \sum_{1 \leq i < n} \sum_{\substack{|\boldsymbol{v}_{\mathrm{II}}^{(r)}|=1 \\ i \leq r < n-1}} \prod_{i < k < n} \frac{\Lambda_k(\boldsymbol{v}_{\mathrm{II}}^{(k-1)}; \boldsymbol{v}_{\mathrm{I}}^{(1\ldots n-1)})}{\boldsymbol{v}_{\mathrm{II}}^{(k-1)} - \boldsymbol{v}_{\mathrm{II}}^{(k)}} \, t_{in}(\boldsymbol{v}_{\mathrm{II}}^{(n-1)}) \Phi(\boldsymbol{v}_{\mathrm{I}}^{(1\ldots n-1)}), \qquad (A.12)$$

where $\boldsymbol{v}_{\mathrm{II}}^{(n-1)} = v_j^{(n-1)}$ for any $1 \leq j \leq m_{n-1}$ and $\boldsymbol{v}_{\mathrm{II}}^{(s)} = \emptyset$ for all $1 \leq s < i$ so that

$$\boldsymbol{v}_{\mathrm{I}}^{(1\ldots n-1)} = (\boldsymbol{v}^{(1)}, \ldots, \boldsymbol{v}^{(i-1)}, \boldsymbol{v}_{\mathrm{I}}^{(i)}, \ldots, \boldsymbol{v}_{\mathrm{I}}^{(n-1)}). \qquad (A.13)$$

*Example* A.3. When $n = 4$, the recurrence relation (A.12) gives

$$\begin{aligned}
\Phi(\boldsymbol{v}^{(1,2,3)}) = \; & t_{34}(\boldsymbol{v}_{\mathrm{II}}^{(3)}) \Phi((\boldsymbol{v}^{(1)}, \boldsymbol{v}^{(2)}, \boldsymbol{v}_{\mathrm{I}}^{(3)})) \\
& + \sum_{|\boldsymbol{v}_{\mathrm{II}}^{(2)}|=1} t_{24}(\boldsymbol{v}_{\mathrm{II}}^{(3)}) \Phi((\boldsymbol{v}^{(1)}, \boldsymbol{v}_{\mathrm{I}}^{(2)}, \boldsymbol{v}_{\mathrm{I}}^{(3)})) \frac{f^-(\boldsymbol{v}_{\mathrm{II}}^{(2)}, \boldsymbol{v}_{\mathrm{I}}^{(2)}) f^+(\boldsymbol{v}_{\mathrm{II}}^{(2)}, \boldsymbol{v}_{\mathrm{I}}^{(3)}) \lambda_3(\boldsymbol{v}_{\mathrm{II}}^{(2)})}{\boldsymbol{v}_{\mathrm{II}}^{(2)} - \boldsymbol{v}_{\mathrm{II}}^{(3)}} \\
& + \sum_{\substack{|\boldsymbol{v}_{\mathrm{II}}^{(r)}|=1 \\ r=1,2}} t_{14}(\boldsymbol{v}_{\mathrm{II}}^{(3)}) \Phi((\boldsymbol{v}_{\mathrm{I}}^{(1)}, \boldsymbol{v}_{\mathrm{I}}^{(2)}, \boldsymbol{v}_{\mathrm{I}}^{(3)})) \frac{f^-(\boldsymbol{v}_{\mathrm{II}}^{(1)}, \boldsymbol{v}_{\mathrm{I}}^{(1)}) f^+(\boldsymbol{v}_{\mathrm{II}}^{(1)}, \boldsymbol{v}_{\mathrm{I}}^{(2)}) \lambda_2(\boldsymbol{v}_{\mathrm{II}}^{(1)})}{\boldsymbol{v}_{\mathrm{II}}^{(1)} - \boldsymbol{v}_{\mathrm{II}}^{(2)}} \\
& \qquad\qquad\qquad \times \frac{f^-(\boldsymbol{v}_{\mathrm{II}}^{(2)}, \boldsymbol{v}_{\mathrm{I}}^{(2)}) f^+(\boldsymbol{v}_{\mathrm{II}}^{(2)}, \boldsymbol{v}_{\mathrm{I}}^{(3)}) \lambda_3(\boldsymbol{v}_{\mathrm{II}}^{(2)})}{\boldsymbol{v}_{\mathrm{II}}^{(2)} - \boldsymbol{v}_{\mathrm{II}}^{(3)}}, \qquad (A.14)
\end{aligned}$$

where $\boldsymbol{v}_{\mathrm{II}}^{(3)} = v_j^{(3)}$ for any $1 \leq j \leq m_3$.

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
