# Peer review of "Bethe vectors and recurrence relations for twisted Yangian based models"

_SciPost Physics, doi:SciPost Phys. 17, 126 (2024)_

## Round 2 · Referee Report · Anonymous (Referee 2) · 2024-8-31

Strengths
Weaknesses
Report
The results obtained can be used to study scalar products of Bethe vectors in models with twisted Yangian. In turn, scalar products are needed to calculate correlation functions. Thus, the results listed above are certainly important.
The paper is very technical and quite difficult to read. However, I do not think that it can be written much simpler, since the topic itself is very technical.
I have read the author's responses to previous reviews and believe that the author has done a great job. In my opinion, the paper does not need further improvement. I suggest publishing the paper in its current form.
Requested changes
No changes.
Recommendation
Publish (easily meets expectations and criteria for this Journal; among top 50%)

Vidas Regelskis on 2024-08-09 [id 4685]
Markdown corrections to Author comments upon resubmission:
I'd like to thank the referees for their valuable comments and suggestions. Responses to the queries raised by the referees are given below.
Referee 1: Thank you for spotting the misprints and notation issues. These have been corrected. The notation $\tilde{v} = -v-\rho$ has been recalled before formula (3.35) (previously (43)).
Referee 2: Both Introduction and Conclusions sections were extended and a summary of the main results is now included.
Referee 3: 1. I agree with the referee that the parameter $\rho$ does not play a significant role in the paper, but it is important to applications of the results of this paper. This is now explained at the beginning of Section 2.3. Please see lines 140-145. 2. I have simplified Sections 2.3 and 2.4. I have removed the confusing part about the presentation of $Y^+(gl_{2n+1})$ in terms of the $(2n+2)\times (2n+2)$-dimensional generating matrix. Instead, I have defined "overlapping" matrix operators A, B, C and D in (2.15). Their crucial property is that in the $N=2n+1$ case they satisfy exchange relations of the same form as their non-overlapping counterparts in the $N=2n+2$ case. Taking matrix coefficients of (2.16)–(2.20) one obtains relations among generating series that coincide with those given by the defining relations (2.10) and (2.11). Please see lines 161-162. 3. It is not possible to construct $Y^-(gl_{2n+1})$ because it is not possible to anti-symmetrise the middle index $i=n+1$ giving $\theta_{n+1}=0$ and thus trivialising the series $s_{ij}(u)$ with $i=n+1$ or $j=n+1$. 4. The new Remark 2.1 explains the issues emerging in the $N=2n+1$ case that can not be deduced from the results of [GMR19]. 5. The Introduction has now a summary of the main results of the paper. 6. I have clarified this point in Section 3.1. Please see lines 183-184. 7. I have rewritten the second half of Section 3.2 that introduces the nested quantum spaces. I added an explanation of the tuple labelling. Please see lines 207-210. 8. I agree with the referee that some statements in the paper could be omitted by referring to [GMR19] instead. However, the current paper uses a slightly different notation and a row-nesting instead of a column-nesting after the first (top) level of nesting. The two approaches give the same result, but the row-nesting is more natural for the purposes of the present paper. This is why many things are recalled (and reformulated to fit the row-nesting approach) from [GMR19]. 9. The misprint has been corrected. 10. I have updated Section 4.1 discussing partitions of tuples. Please see lines 399-403. 11. This has been clarified now. The new Appendix A.3 recalls the relevant recurrence relation from [HL+17b] in terms of the notation used in this paper. Please see Proposition A.2 and Example A.3. 12. This notation is now included in (3.9).
Referee 4: 1. I agree with the referee that it is possible to choose a different full quantum space than the one defined in (3.1). For the nesting to work it is crucial that the individual quantum spaces can be restricted to irreducible representations of the reduced algebra. A tensor product of evaluation modules does have the wanted property (see lines 189-196). The paper is already technical, thus we do not consider more general constructions of the full quantum space. 2. This has been clarified. Please see lines 330-333 and Appendix A.2. 3. It is possible to consider situations when both boundary spaces are not one-dimensional. In such cases one would have to find a way to diagonalised one of the boundary spaces (see Remark 4.1 in [GMR19]) for the (usual) nested Bethe ansatz to work, or to employ off-diagonal Bethe ansatz techniques (see the book "Off-diagonal Bethe ansatz for exactly solvable models" by Y. Wang, W.-L. Yang, J. Cao and K. Shi). Alternatively, one could try employing separation of variables techniques (see new references [GLMS17] and [RV21]). 4. This is correct. The index $j$ does appear in both places. I have now included a derivation of formula (4.3) (previously (61)) where this property originates due to a repeated application of formula (4.2). Please see Proof of Corollary 4.2. 5. I thank the referee for this suggestion. I have updated the Introduction accordingly. Please see lines 85-99. 6. I thank the referee for this suggestion. The new Appendix A.1 explains this property.

---

## Round 2 · Author Response

I'd like to thank the referees for their valuable comments and suggestions. Responses to the queries raised by the referees are given below.
Referee 1: Thank you for spotting the misprints and notation issues. These have been corrected. The notation $\tilde{v} = -v-\rho$ has been recalled before formula (3.35) (previously (43)).
Referee 2: Both Introduction and Conclusions sections were extended and a summary of the main results is now included.
Referee 3:
-
I agree with the referee that the parameter $\rho$ does not play a significant role in the paper, but it is important to applications of the results of this paper. This is now explained at the beginning of Section 2.3. Please see lines 140-145.
-
I have simplified Sections 2.3 and 2.4. I have removed the confusing part about the presentation of $Y^+(\mathfrak{gl}_{2n+1})$ in terms of the $(2n+2)\times (2n+2)$-dimensional generating matrix. Instead, I have defined "overlapping" matrix operators A, B, C and D in (2.15). Their crucial property is that in the $N=2n+1$ case they satisfy exchange relations of the same form as their non-overlapping counterparts in the $N=2n+2$ case. Taking matrix coefficients of (2.16)–(2.20) one obtains relations among generating series that coincide with those given by the defining relations (2.10) and (2.11). Please see lines 161-162.
-
It is not possible to construct $Y^-(\mathfrak{gl}_{2n+1})$ because it is not possible to anti-symmetrise the middle index $i=n+1$ giving $\theta_{n+1}=0$ and thus trivialising the series $s_{ij}(u)$ with $i=n+1$ or $j=n+1$.
-
The new Remark 2.1 explains the issues emerging in the $N=2n+1$ case that can not be deduced from the results of [GMR19].
-
The Introduction has now a summary of the main results of the paper.
-
I have clarified this point in Section 3.1. Please see lines 183-184.
-
I have rewritten the second half of Section 3.2 that introduces the nested quantum spaces. I added an explanation of the tuple labelling. Please see lines 207-210.
-
I agree with the referee that some statements in the paper could be omitted by referring to [GMR19] instead. However, the current paper uses a slightly different notation and a row-nesting instead of a column-nesting after the first (top) level of nesting. The two approaches give the same result, but the row-nesting is more natural for the purposes of the present paper. This is why many things are recalled (and reformulated to fit the row-nesting approach) from [GMR19].
-
The misprint has been corrected.
-
I have updated Section 4.1 discussing partitions of tuples. Please see lines 399-403.
-
This has been clarified now. The new Appendix A.3 recalls the relevant recurrence relation from [HL+17b] in terms of the notation used in this paper. Please see Proposition A.2 and Example A.3.
-
This notation is now included in (3.9).
Referee 4:
-
I agree with the referee that it is possible to choose a different full quantum space than the one defined in (3.1). For the nesting to work it is crucial that the individual quantum spaces can be restricted to irreducible representations of the reduced algebra. A tensor product of evaluation modules does have the wanted property (see lines 189-196). The paper is already technical, thus we do not consider more general constructions of the full quantum space.
-
This has been clarified. See lines 330-333 and Appendix A.2.
-
It is possible to consider situations when both boundary spaces are not one-dimensional. In such cases one would have to find a way to diagonalised one of the boundary spaces (see Remark 4.1 in [GMR19]) for the (usual) nested Bethe ansatz to work, or to employ off-diagonal Bethe ansatz techniques (see the book "Off-diagonal Bethe ansatz for exactly solvable models" by Y. Wang, W.-L. Yang, J. Cao and K. Shi). Alternatively, one could try employing separation of variables techniques (see new references [GLMS17] and [RV21]).
-
This is correct. The index $j$ does appear in both places. I have now included a derivation of formula (4.3) (previously (61)) where this property originates due to a repeated application of formula (4.2). Please see Proof of Corollary 4.2.
-
I thank the referee for this suggestion. I have updated the Introduction accordingly. Please see lines 85-99.
-
I thank the referee for this suggestion. The new Appendix A.1 explains this property.

---

## Round 2 · List of Changes

- I have extended Introduction and Conclusions sections with additional references, a summary of the main results, and some open problems.
- I have simplified Section 2.3 and updated Section 2.4. A new Remark 2.1 was added.
- I have updated the discussion of quantum spaces in Section 3.1 and clarified Sections 3.2 and 3.3 by emphasizing the specificities of the N=2n+1 case.
- I have updated the explanation of partitions in Section 4.1.
- The previous expansion formula (61) is now presented in the new Corollary 4.2. A proof of this formula and an example of such an expansion when N=3 was added.
- An appendix was added. Appendix A.1 shows compatibility of recurrence relations with the weight grading of twisted Yangians. Appendix A.2 shows commutativity of transfer matrices. Appendix A.3 recalls a Y(n)-type recurrence relation used in this paper.

---

## Round 3 · Author Response

I'd like to thank the referees for reviewing the manuscript and for the last minor suggestions. Below are my replies to the points raised by the first referee.

  1. I thank the referee for noticing the overlap of transposition symbol $w$ appearing in Sections 2 with the complex parameter $w$ appearing in Section 3. The transposition symbol $w$ was replaced with $\omega$; letter $t$ is already used in Section 3.6 and Appendix A.2 to denote the usual transposition. (The symbol $\omega$ is also used in Appendix A.1 to denote weights of elements $s_{ij}[r]$; this notation appears in Appendix A.1 only and should not bring any confusion with transposition $\omega$ in Section 2.)

  2. It is noted above equation (2.15) that $A$, $B$, $C$ and $D$ are $\hat{n}\times {\hat{n}$ dimensional matrix operators and so $i$ and $j$ run from $1$ to $\hat{n}$.

  3. The notation of "overlapping" matrix operators is now illustrated in Remark 2.1 with an explicit example of the $N=3$ case; the text below the new example was also clarified. I hope this resolves the issue raised by the referee.

  4. I agree with the referee that the empty set notation is delicate to employ. I have now explicitly indicated the empty sets in all relevant statements and examples in Section 4 and Appendix A.3. I hope this clarifies this delicate notation.

  5. I thank the referee for spotting a typo in Corollary 4.2; it has been corrected. A similar typo was in Example 4.3; it was also corrected.

---

## Round 3 · List of Changes

• The transposition symbol $w$ was replaced with $\omega$ in lines 132, 137, 160 and equations (2.8), (2.14).

  • Remark 2.1 was updated with an explicit example of the $A$, $B$, $C$ and $D$ operators for the $N=3$ case; the text below the example was clarified.

  • The empty subsets were explicitly indicated in all statements and examples in Section 4 and Appendix A.3.

  • Typos in Corollary 4.2 and Example 4.3 were fixed.

  • Reference [Gom24] was updated with journal entry and doi.

---

## Editorial Decision

published